# Insertion Based Sequence Generation with Learnable Order Dynamics

**Dhruvesh Patel** [* 1]  **Benjamin Rozonoyer** [* 1]  **Gaurav Pandey** [2]  **Tahira Naseem** [2]  **Ramón Fernandez Astudillo** [2]
**Andrew McCallum** [1]

## Abstract

Existing insertion-based masked diffusion models that generate sequences by interleaving token insertion with unmasking use fixed schedules that are not dependent on the data. For structured sequences like graphs and molecules, learning data-dependent generation orders can improve generation quality by reducing uncertainty over the action space. We propose **Lo**FlexMDM, an insertion-based masked diffusion model with learnable order dynamics that learns data-dependent insertion and unmasking rates. We generalize the discrete flow matching framework to work with variable-length sequences, propose a tractable schedule parameterization and a training objective for joint training of the generator and the target order dynamics. On *De Novo* and fragment-constrained molecule generation, **Lo**FlexMDM improves sample quality over FlexMDM by up to 17.5% and 6.7%, respectively. These results show that learning the target generation order can improve insertion-based diffusion models without giving up tractable training. We open source the code at https://github.com/dhruvdcoder/LoFlexMDM.

## 1. Introduction

Masked diffusion models (MDMs) (Sahoo et al., 2024; Shi et al., 2024) generate sequences by iteratively unmasking tokens whose positions are expressed as absolute indices in a fixed-length string. However, many structured sequences like string representations of graphs and molecules are naturally variable-length with dependencies that are better expressed through relative positions. For example, consider a star-shaped graph with one junction node, and a path that goes through the junction node. All the edges on this path are easy to predict locally, except the edge going outward from the junction, which requires looking ahead to determine the correct arm to choose (see Appendix B for a concrete example). Therefore, a natural way to generate such a path is to start from both the endpoints and insert edges one by one inward toward the junction such that by the end the outward edge from the junction also becomes trivially predictable (Patel et al., 2025b).

Insertion-based diffusion models (Patel et al., 2026b; Ding et al., 2026) that generate by inserting tokens at any available gap between existing tokens offer a principled approach to generating variable-length sequences. However, by limiting insertions to one token per gap, these models stop short of realizing the full benefits of parallel generation. More importantly, single-token insertions force greedy local decisions and reduce the model's ability to capture correlations among token groups within the same gap. A model that inserts multiple mask tokens into a gap and unmasks them in parallel avoids both limitations.

Kim et al. (2025a) propose FlexMDM, a variable-length MDM that grows a sequence by inserting multiple mask tokens in any available gap while simultaneously unmasking available masked tokens. In FlexMDM, each token is first inserted as a mask and later unmasked into a symbol. This two-stage process expands the set of possible generation orders. Since the target insertion and unmasking rates are fixed (independent of the data sample), the model must learn to generate every ground truth sequence in all possible orders in this expanded space. However, when some orders are more suitable than others, learning to generate in all possible orders, which spreads the probability across many potentially suboptimal generation orders, is wasteful.

We propose **Lo**FlexMDM, an insertion-based masked diffusion model with **L**earnable **o**rder dynamics. **Lo**FlexMDM keeps FlexMDM's two-step insertion and unmasking process, but replaces fixed target schedules with learnable, example-dependent rates, which implicitly define the *order dynamics* as a distribution over generation orders (Fig. 1 left). To make these learned schedules trainable, we introduce three technical contributions. First, we extend the Discrete Flow Matching (DFM) framework (Lipman et al., 2024) to projected partial sequences (Section 2) and derive

---

[*]Equal contribution  [1]University of Massachusetts Amherst  [2]IBM Research. Correspondence to: Dhruvesh Patel <dhruveshpate@umass.edu>.

*Proceedings of the 43rd International Conference on Machine Learning*, Seoul, South Korea. PMLR 306, 2026. Copyright 2026 by the author(s).

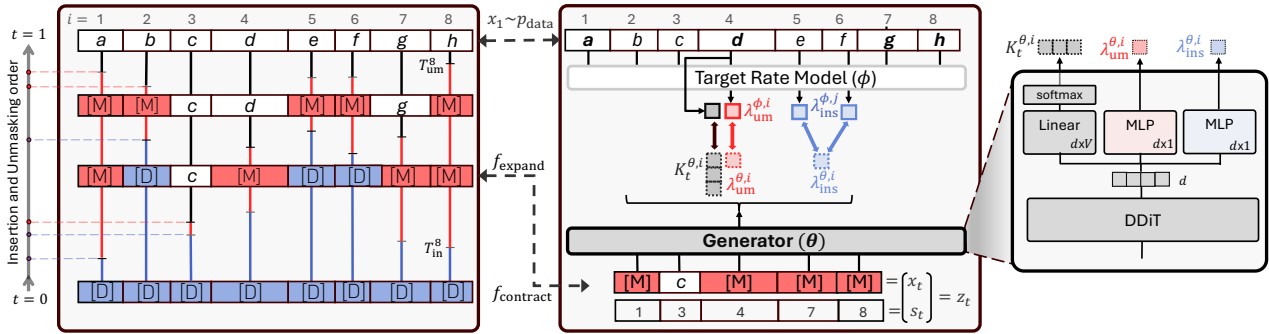

*Figure 1.* **Left:** The left subfigure illustrates how the endpoint-dependent target process grows the final sequence $x_1$ from the empty sequence (all `[D]` tokens) by sampling the insertion time $T_{\text{in}}{}^i \sim f^\phi_{T_{\text{in}}{}^i}(t; x_1)$ for the insertion transition `[D]` $\rightarrow$ `[M]`, followed by the unmasking time $T_{\text{um}}{}^i \sim f^\phi_{T_{\text{um}}{}^i|T_{\text{in}}{}^i}(t; x_1)$ for the unmasking transition `[M]` $\rightarrow x_1^i$, for each position $i$. The target process induces a target generation order: $j$ is generated after $i$ if $T_{\text{um}}{}^j > T_{\text{um}}{}^i$. **Middle:** The middle subfigure (top), shows the auxiliary neural network $\phi$ that takes in a clean sequence $x_1$ and outputs per-token target insertion rates $\lambda^\phi_{\text{in}}$ and unmasking rates $\lambda^\phi_{\text{um}}$, which induce the densities $f^\phi$. The map $f_{\text{contract}}$ removes the `[D]` tokens from the intermediate $z_t$ to obtain a partial sequence $x_t$. The generator network (bottom right) takes $x_t$ and produces insertion $\lambda^\theta_{\text{in}}$ and unmasking $\lambda^\theta_{\text{um}} \cdot K^\theta$ rates which are trained to match the endpoint-dependent target rates $\lambda^\phi_{\text{in}}$ and $\lambda^\phi_{\text{um}}$ using Eq. (11) in Section 3.2. **Right:** The rightmost subfigure shows the generator network which consists of a transformer backbone, a linear LM head that produces token probabilities, and two rate heads that produce insertion and unmasking rates.

the ELBO for joint training of the generator and the target schedules (Section 3.2). Second, we propose a tractable parameterization of the target schedules using Kumaraswamy CDFs (Kumaraswamy, 1980), which allows us to sample the event times in parallel while also admitting a closed form likelihood (Section 3.1.1). This parameterization allows joint training of the generator and target schedules with a REINFORCE leave-one-out estimator, without simulating full trajectories (Section 3.2.1).

Across graph traversal and molecule generation, **Lo**FlexMDM learns generation orders that match the structure of the task. These learned orders improve sample quality over FlexMDM by up to 17.5 percentage points on *de novo* and 6.7 points on fragment-constrained molecule generation.

## 2. Preliminaries: Discrete Flow Matching

Let $\mathbb{X}$ be a countable space in which our data lives (e.g., the space of sequences of length $L$). The aim is to construct a generative process that samples from a desired distribution $p_1 = p_{\text{data}}$ over this space. Discrete Flow Matching (Campbell et al., 2024; Gat et al., 2024) constructs a continuous time Markov chain $\{X_t\}$ taking values in $\mathbb{X}$, with time marginals $p_t(x) := \mathbb{P}(X_t = x)$, which transforms an initial sample $X_0 \sim p_0$ to a final sample $X_1 \sim p_1 = p_{\text{data}}$, where $p_0$ is easy to sample from. For small time steps $h$, the CTMC transitions follow $x_{t+h} \sim p_{t+h|t}(\cdot \mid x_t) := \mathbb{P}(X_{t+h} = \cdot | X_t = x_t)$, where the transition probability is given by

$$p_{t+h|t}(y \mid x) = \delta_x(y) + h\, R_t(x, y) + o(h),$$

and $\delta_x(y)$ is the indicator function. The initial distribution $p_0$ and the rate $R_t$ of the CTMC determine the gen-

erative process completely. [1] Therefore, to obtain the desired CTMC, one first constructs a *probability flow* $p_t$ such that the desired terminal distribution condition is satisfied: $p_{t=1} = p_{\text{data}}$, and then constructs a (non-unique) generator rate $R_t$ that satisfies the Kolmogorov Forward Equation:

$$\partial_t p_t(x) = \sum_{y \in \mathbb{X}} R_t(y, x)\, p_t(y). \tag{1}$$

Such a rate is said to *generate* the marginal probability path $p_t$. The general template is to choose a family of densities $p_{t|C}$, where $C \in \mathbb{C}$ is an indexing or conditioning variable, typically an endpoint index $C = (X_0, X_1)$, and then to match the unknown rates $R_t^\theta(x, \cdot)$ with the target conditional rates $R_t(x, \cdot \mid C)$ that generate $p_{t|C}$ (in the sense of Eq. (1)) by minimizing the Bregman divergence:

$$\mathcal{L}(\theta) = \mathop{\mathbb{E}}_{X_t, C} D\big(R_t(X_t, \cdot \mid C), R_t^\theta(X_t, \cdot)\big),$$

where the expectation is over $C \sim p(C)$, $X_t \sim p_{t|C}$. In the case of $C = (X_0, X_1)$, where $X_1 \sim p_{\text{data}}$, this maximizes a lower bound on the data log-likelihood (Shaul et al., 2025).

**Projected DFM:** The framework of DFM can be generalized to construct a conditional rate on an auxiliary space $\mathbb{Z}$, where it may be easier to define the dynamics, and then *project* the rate onto the space of interest $\mathbb{X}$ creating weakly intertwined CTMCs as shown in the schematic in Fig. 2. Let $\{Z_t\}$ be a CTMC taking values in $\mathbb{Z}$, and define $\pi_{\text{enc}}(x \mid z)$ to be an *encoder kernel* from $\mathbb{Z}$ to $\mathbb{X}$, then the auxiliary dynamics

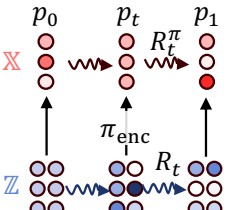

*Figure 2.* Projected Discrete Flow Matching

---

[1]Please refer to Appendix C for an extended discussion.

induce a (law-dependent) projected CTMC on $\mathbb{X}$ with rate

$$R_t^\pi(x, y) = \sum_{z,w \in \mathbb{Z}} \pi_{\text{enc}}(y \mid w)\, \pi_{\text{dec},t}(z \mid x)\, R_t(z, w),$$

where $z, w \in \mathbb{Z}$, $R_t$ is the rate for the CTMC $\{Z_t\}$, and $\pi_{\text{dec},t}(z \mid x) \propto p_t(z)\pi_{\text{enc}}(x \mid z)$ is the Bayes decoder. The same construction applies to conditional rates $R_t(z, w \mid C)$, yielding projected conditional rates $R_t^\pi(x, y \mid C)$. Even though the target path is defined through the auxiliary process, we can still learn the generator $R_t^\theta$ over $\mathbb{X}$ alone by minimizing the Bregman divergence: [2]

$$\mathcal{L}(\theta) = \mathop{\mathbb{E}}_{X_t, C} D\big(R_t^\pi(X_t, \cdot \mid C),\, R_t^\theta(X_t, \cdot)\big). \quad (2)$$

**Positional Independence Assumption.** For product spaces $\mathbb{X}^L$ like the space of sequences, the size of the general rate matrix grows exponentially with the length $L$. To avoid this, sparsity is introduced into the rate matrix by assuming per-position independence of the target conditional probability $p_{t|C}(x_t|c)$, i.e., $p_{t|C}(x_t|c) = \prod_{i=1}^L p_{t|C}^i(x_t^i|c)$, where the superscript $i$ denotes the $i$-th position, which results in the corresponding rate matrix being a mixture of per-position rates (Campbell et al., 2022). We will make a similar assumption for projected rate matching for variable length sequences.

# 3. Variable Length Sequence Generation

Let $\mathbb{X} := \bigcup_{k=1}^L \mathbb{V}^k$ be the space of variable length sequences. Since there are multiple possible alignments of a partial sequence $x_t$ to a clean sequence $x_1$, it is not easy to construct conditional marginals $p_{t|C}(x_t|x_1)$ as is typically done in masked DFM. However, one can use an auxiliary space that includes the alignment information and perform projected rate matching. Similarly to Kim et al. (2025a), we define the auxiliary space of partial sequences as $\mathbb{Z} := \bigcup_{k \in \llbracket L \rrbracket} \mathbb{V}^k \times \mathbb{S}_k$, where $\mathbb{S}_k := \{(s^1, \ldots, s^k) \in \llbracket L \rrbracket^k \mid s^1 < \cdots < s^k\}$ is the space of ordered tuples of positions and $L$ is the maximum length. An example element of $\mathbb{Z}$ is shown on the right of Fig. 1 as the input $(x_t, s_t)$ to the generator. Additionally, we note the following isomorphism. Let $\texttt{[D]}$ be a special token outside the vocabulary $\mathbb{V}$ that is used to indicate a dropped position. Then $\bar{\mathbb{Z}} := (\mathbb{V} \cup \{\texttt{[D]}\})^L$ is isomorphic to $\mathbb{Z}$ through the mapping $f_{\text{contract}} : \bar{\mathbb{Z}} \to \mathbb{Z}$ that removes the $\texttt{[D]}$ tokens from $\bar{z}$ to form $x$, and moves the exact position information to $s$ to produce $z = (x, s)$. For example, as shown in Fig. 1, for $L = 8$ and $\bar{z} = (\texttt{[M]}, \texttt{[D]}, c, \texttt{[M]}, \texttt{[D]}, f, \texttt{[M]}, \texttt{[M]})$, $f_{\text{contract}}(\bar{z}) = ((\texttt{[M]}, c, \texttt{[M]}, f, \texttt{[M]}, \texttt{[M]}), (1, 3, 4, 6, 7, 8))$. From this point on, we will use $z$ to mean an element of $\mathbb{Z}$ or $\bar{\mathbb{Z}}$ interchangeably. We also define $f_{\text{rm-drop}} : \bar{\mathbb{Z}} \to \mathbb{X}$ by deleting each $\texttt{[D]}$ slot; for the example above, $f_{\text{rm-drop}}(\bar{z}) = (\texttt{[M]}, c, \texttt{[M]}, f, \texttt{[M]}, \texttt{[M]})$. Note that the generator oper-

---

[2]See Appendix D for a detailed discussion.

ates only on $\mathbb{X}$, and the $\texttt{[D]}$ token is not part of the input or the output of the generator.

**Mixture paths in the auxiliary space.** We construct mixture paths on $\bar{\mathbb{Z}}$ by setting the conditioning variable as $(z_1, z_0)$, where $z_1 = \left(x_1 \odot \texttt{[D]}^{L - \text{len}(x_1)} \overset{\ldots}{\texttt{[D]}}, \llbracket L \rrbracket\right)$ and $z_0 = \texttt{[D]}.\overset{L}{\ldots}.\texttt{[D]}$, with $\odot$ denoting the sequence concatenation operation. The goal is to define per-position target rates such that the transition $\texttt{[D]} \to \texttt{[M]} \to z_1^i$ occurs for each position $i$ where $z_1^i \neq \texttt{[D]}$ so that we can express a tractable distribution over the orderings of insertion and unmasking events for the entire sequence, which can be learned along with the generator parameters. For the positions $i$ where $z_1^i = \texttt{[D]}$, the target exit rate is identically zero leaving the position pinned at $\texttt{[D]}$.

## 3.1. Learnable Order Dynamics

Any regular CTMC over a countable state space can be represented using a Poisson process and an embedded state transition kernel. Therefore, we define the per-position target rates as

$$R_t^i(z, w^i \mid (z_1, z_0)) = \lambda_t^i(z; (z_1, z_0)) \cdot K_t^i(z, w^i \mid (z_1, z_0)),$$

where $\lambda_t^i(z; (z_1, z_0))$ is the state and time dependent arrival rate of the Poisson process, and $K_t^i(z, w^i \mid (z_1, z_0))$ is the state transition kernel—i.e, when an arrival occurs at the $i$-th position, the state changes according to the categorical distribution $K_t^i$. In order for $R_t(z, w)$ to be a valid rate, $\lambda_t^i > 0$ and $K_t^i(z, z^i) = 0$. This characterization allows us to express the order dynamics by controlling the hazard rates, without changing the terminal state distributions of the process. For our case of two events per position, insertion and unmasking, we denote the respective event times as $T_{\text{in}}^{~i} := T_1^i$ and $T_{\text{um}}^{~i} := T_2^i$. The left side of Figure 1 provides an intuitive illustration of the induced order dynamics. Proposition 1 shows that the target rates that generate the desired conditional probability flow can be expressed using two increasing right continuous functions. Note that since $z_0 = \texttt{[D]}.\overset{L}{\ldots}.\texttt{[D]}$ is fixed, we will hide the dependence on $z_0$ from the notation for brevity.

**Proposition 1** (Target conditional rates). *Let* $\mathbb{P}(T_{\text{in}}^{~i} \leq t) = F_{\text{in}}^i(t)$ *and* $\mathbb{P}(T_{\text{um}}^{~i} \leq t \mid T_{\text{in}}^{~i} = s) = \delta(t \geq s)\frac{F_{\text{um}}^i(t) - F_{\text{um}}^i(s)}{1 - F_{\text{um}}^i(s)}$, *where* $F_{\text{in}}^i(t)$ *and* $F_{\text{um}}^i(t)$ *are right continuous and non-decreasing functions on* $[0, 1]$ *with* $F_{\text{in}}^i(0), F_{\text{um}}^i(0) = (0, 0)$, *and* $F_{\text{in}}^i(1), F_{\text{um}}^i(1) = (1, 1)$. *Then the rate*

$$\lambda_t^i(z; z_1) = \begin{cases} \dfrac{\dot{F}_{\text{in}}^i(t; z_1)}{1 - F_{\text{in}}^i(t; z_1)} & \text{if } z^i = \texttt{[D]} \neq z_1^i, \\[3mm] \dfrac{\dot{F}_{\text{um}}^i(t; z_1)}{1 - F_{\text{um}}^i(t; z_1)} & \text{if } z^i = \texttt{[M]}, \\[2mm] 0 & \text{otherwise.} \end{cases}$$

$$K_t^i(z, w^i \mid z_1) = \begin{cases} \delta_{[\mathtt{M}]}(w^i) & \text{if } w^i = [\mathtt{D}], \\ \delta_{z_1^i}(w^i) & \text{if } w^i = [\mathtt{M}], \\ \text{undefined} & \text{otherwise.} \end{cases}$$

generates the marginals (i.e., satisfies the KFE):

$$p_t^i(z^i|z_1)$$

$$= \begin{cases} \mathbb{P}(T_{\text{in}}{}^i > t)\,\delta_{[\mathtt{D}]}(z^i) \\ \quad + \mathbb{P}(T_{\text{in}}{}^i \le t < T_{\text{um}}{}^i)\,\delta_{[\mathtt{M}]}(z^i) & \text{if } z_1^i \neq [\mathtt{D}], \\ \quad + \mathbb{P}(T_{\text{um}}{}^i \le t)\,\delta_{z_1^i}(z^i) \\ \delta_{[\mathtt{D}]}(z^i) & \text{if } z_1^i = [\mathtt{D}], \end{cases}$$

and these marginals satisfy the boundary conditions $p_1^i(z^i|z_1) = \delta_{z_1^i}(z^i)$ and $p_0^i(z^i|z_1) = \delta_{[\mathtt{D}]}(z^i)$.

### 3.1.1. PARAMETERIZATION

By learning the parameters $\phi$ governing $F_*^{\phi,i} : [0,1] \times \mathbb{Z} \to [0,1]$ of Proposition 1 we can learn the target order dynamics. The choice of the functional form of $F_*$ depends on the analytical tractability of the resulting $p_t^\phi(z|z_1)$. Specifically, we need to be able to sample $z_t \sim p_t^\phi(z|z_1)$ (ideally using inverse CDF sampling) in order to compute the projected rate matching loss (Eq. (2)) and also be able

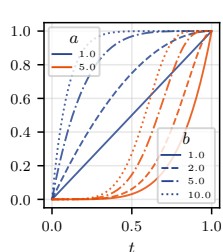

*Figure 3.* Kumaraswamy CDF shapes for different parameter values.

to compute the likelihood $p_t^i(z^i|z_1)$ for updating $\phi$, ideally in closed form to avoid numerical integration. Note that since the likelihood is independent per-position, if we satisfy the desiderata stated above, we obtain an efficient training procedure as discussed in Section 3.2. As discussed in Appendix E.2, the likelihood $p_t^\phi(z|z_1)$ requires computing the following integral:

$$\mathbb{P}(T_{\text{in}}{}^i \le t < T_{\text{um}}{}^i) = (1 - F_{\text{um}}^i(t)) \int_0^t \frac{\dot{F}_{\text{in}}^i(s)}{1 - F_{\text{um}}^i(s)}\, ds.$$

The Kumaraswamy CDF (Kumaraswamy, 1980) (Fig. 3):

$$F_*^{\phi,i}(t; z_1) = 1 - (1 - t^{a_*})^{b_*}, \quad * \in \{\text{in}, \text{um}\} \quad (3)$$

is an ideal candidate for the functional form of $F_*$ as it allows inverse CDF sampling, can produce internal modes, and for $a_{\text{in}}^i = a_{\text{um}}^i = a$, allows us to obtain closed form expressions for the likelihood (see Appendix E.5.3 for more details). Therefore, we parameterize $b_{\text{in}}^{\phi,i}(z_1)$ and $b_{\text{um}}^{\phi,i}(z_1)$ by a transformer with weights represented by $\phi$, and set $a_{\text{in}}^i = a_{\text{um}}^i = a$ to be a constant, which yields the following closed form expressions for the hazard rates:

$$\lambda_{\text{in}}^i(t) = b_{\text{in}} \frac{a t^{a-1}}{1 - t^a}, \qquad \lambda_{\text{um}}^i(t) = b_{\text{um}} \frac{a t^{a-1}}{1 - t^a}. \quad (4)$$

Interestingly, in this final form, we can see that the shape function $\frac{a t^{a-1}}{1 - t^a}$ is the same for both insertion and unmasking, and only the hazard rate multipliers $b_{\text{in}}$ and $b_{\text{um}}$ control the overall rate. Under the time-change $\tau = -\log(1 -$

$t^a)$, it becomes and exponential race with the parameters $b_{\text{in}}$ and $b_{\text{um}}$ with precedence constraints (insertion before unmasking).

### 3.2. Training

The training objective, illustrated on the right side of Figure 1 can be derived using projected rate matching (Appendix D.0.1), and is given by the unmasking and insertion loss as follows:

$$\mathcal{L}_{\text{um}}^{\phi,\theta} = \mathbb{E}_\phi \left[ \sum_{i=1}^{\text{len}(x_t)} \delta_{[\mathtt{M}]}(x_t^i) \Big( -\lambda_{\text{um},t}^{\phi,s_t^i}(z_1) \log K_t^{\theta,i}(z_1^{s_t^i}|x_t) \right.$$

$$\left. + D_\psi(\lambda_{\text{um},t}^{\phi,s_t^i}(z_1), \lambda_{\text{um},t}^{\theta,i}(x_t)) \Big) \right], \quad (5)$$

$$\mathcal{L}_{\text{in}}^{\phi,\theta} = \mathbb{E}_\phi \left[ \sum_{i=0}^{\text{len}(x_t)} D_\psi \left( \sum_{j \in \text{gap}_i(s_t)} \lambda_{\text{in},t}^{\phi,j}(z_1), \lambda_{\text{in},t}^{\theta,i}(x_t) \right) \right],$$

where the expectation is over $t \sim \mathcal{U}(0,1)$, $z_1 \sim p_{\text{data}}$, $z_t \sim p_{t|z_1}^\phi$, and $x_t = f_{\text{rm-drop}}(z_t)$. The Bregman divergence $D_\psi$ is computed using the negative entropy function $\psi(r) = \sum_k r_k \log r_k$. We provide a step-by-step derivation of the training objective in Appendix E.4.

---

**Algorithm 1** Training for **LoFlexMDM**

**Require:** Dataset $\mathcal{D}$, generator parameters $\theta$, target rate parameters $\phi$, learning rate $\eta$
1: **while** not converged **do**
2:     Sample $z_1 \sim p_{\text{data}}$ and $t \sim \mathcal{U}(0,1)$
3:     Compute $(a^{\phi,i}, b^{\phi,i}) \leftarrow \phi(z_1)$ for all $i \in [\![L]\!]$
4:     **for** $k = 1$ to 2 **do**
5:         Sample $T_{\text{in}}{}^{i,(k)} \sim F_{\text{in}}^{\phi,i}$,
6:         $T_{\text{um}}{}^{i,(k)} \leftarrow \text{TRUNCSAMPLE}(T_{\text{in}}{}^{i,(k)}, F_{\text{um}}^{\phi,i})$
7:         $z_t^{i,(k)} \leftarrow \begin{cases} [\mathtt{D}] & T_{\text{in}}{}^{i,(k)} > t \\ [\mathtt{M}] & T_{\text{in}}{}^{i,(k)} \le t < T_{\text{um}}{}^{i,(k)} \\ z_1^i & \text{otherwise} \end{cases}$
8:         $x_t^{(k)}, s_t^{(k)} \leftarrow f_{\text{contract}}(z_t^{(k)})$
9:         $\mathcal{L}_k \leftarrow \mathcal{L}^{\phi,\theta}(x_t^{(k)}, s_t^{(k)}, z_1)$
10:     **end for**
11:     Update $(\theta, \phi) \leftarrow (\theta, \phi) - \eta \nabla_{(\theta,\phi)}^{\text{auto-grad}} \mathcal{L}$     ▷ Eq. (6)
12: **end while**

---

Furthermore, Proposition 5 in Appendix D.0.1 shows that using the standard argument of using KL-divergence between CTMC path measures, followed by the data processing inequality, this loss maximizes a lower bound on the data log-likelihood under the generator.

*Remark* 1. For the case of fixed $\phi$ such that $a_*^{i,\phi} = b_*^{i,\phi} = 1$ for all $i$, the Bregman divergence $D(\lambda_{\text{um}}^\phi, \lambda_{\text{um}}^\theta)$ becomes zero, and we obtain the loss equivalent to FlexMDM (Kim et al., 2025a) written in terms of rates instead of posterior

probabilities. Therefore, we call our method **Lo**FlexMDM, where L stands for learnable order dynamics.

*Remark* 2 (Schedule regularization). Since both the generator and target rates are trainable, they can take extreme values creating numerical degeneracy in the training. To discourage such degenerate schedules, we add a schedule regularization term as described in Appendix E.5.2.

---

**Algorithm 2** Tau-Leaping Sampling for **Lo**FlexMDM

---

1: **Input:** Time steps $0 = t_0 < t_1 < \cdots < t_N = 1$, nucleus $p$, confidence function $c$
2: Initialize $x_0 = \emptyset$ (empty sequence)
3: **for** $n = 0$ to $N - 1$ **do**
4:    $\tau \leftarrow t_{n+1} - t_n$ and $\quad x_{n+1} \leftarrow x_n$
5:    Compute $\lambda^{\theta,i}_{\text{in},t_n}(x_n)$, $\lambda^{\theta,i}_{\text{um},t_n}(x_n)$, and $K^{\theta,i}(\cdot|x_n)$
6:    **for** each gap position $i = 0$ to $\text{len}(x_n)$ **do**
7:       $N^i_{\text{in}} \sim \text{Poisson}(\lambda^{\theta,i}_{\text{in},t_n}(x_n) \cdot \tau)$
8:       **if** $N^i_{\text{in}} > 0$ **then**
9:          Insert $N^i_{\text{in}}$ masks at gap position $i$ in $x_{n+1}$
10:       **end if**
11:    **end for**
12:    **for** each masked position $i$ in $x_{n+1}$ **do**
13:       **for** each token $y \in \mathbb{V}$ **do**
14:          $N^{i,y}_{\text{um}} \sim \text{Poisson}(\lambda^{\theta,i}_{\text{um},t_n}(x_n) \cdot K^{\theta,i}(y|x_n) \cdot \tau)$
15:       **end for**
16:       $u_i \leftarrow \delta(\sum_{y \in \mathbb{V}} N^{i,y}_{\text{um}} > 0)$    ▷ candidate unmask locations
17:    **end for**
18:    **if** `confidence` enabled **then**
19:       $u \leftarrow \text{SELECT}(\sum_i u_i; c(K^\theta, \lambda^\theta))$   ▷ select $|\{i : u_i = 1\}|$ locations with largest confidence $c_i$
20:    **end if**
21:    **for** each masked position $i$ in $x_{n+1}$ with $u_i = 1$ **do**
22:       $x^i_{n+1} \overset{\text{nucleus}(p)}{\sim} K^{\theta,i}(\cdot|x_n)$
23:    **end for**
24: **end for**
25: **return** $x_N$

---

### 3.2.1. GRADIENT COMPUTATION

Since the expectation depends on the target rate network parameters $\phi$, we resort to score based gradient estimation for $\phi$ and use REINFORCE leave-one-out (Kool et al., 2019) to reduce variance with minimal impact on training efficiency. The estimator for $\nabla_{(\phi,\theta)} \mathbb{E}\left[\mathcal{L}^{\phi,\theta}\right]$ is

$$\frac{1}{2}\left[\mathcal{L}^{\phi,\theta}(z^{(1)}) - \mathcal{L}^{\phi,\theta}(z^{(2)})\right]\left[\nabla_\phi \log \frac{p^\phi_{t|z_1}(z^{(1)})}{p^\phi_{t|z_1}(z^{(2)})}\right]$$

$$+ \frac{1}{2}\left[\nabla_\phi \mathcal{L}^{\phi,\theta}(z^{(1)}) + \nabla_\phi \mathcal{L}^{\phi,\theta}(z^{(2)})\right]$$

$$\frac{1}{2}\left[\nabla_\theta \mathcal{L}^{\phi,\theta}(z^{(1)}) + \nabla_\theta \mathcal{L}^{\phi,\theta}(z^{(2)})\right], \quad (6)$$

where $z^{(1)}, z^{(2)} \sim p^\phi_{t|z_1}$ and $z_1 \sim p_{\text{data}}$ (see Appendix E.5 for the details). Intuitively, for fixed $z_1$, the two partial states $z^{(1)}_t$ and $z^{(2)}_t$ differ only in event timing. The first term in Algorithm 1 uses $\mathcal{L}_1 - \mathcal{L}_2$ as an advantage signal from the generator. This term upweights the partial state that $\theta$ can complete better, thereby increasing the log-likelihood of that state (order) under $\phi$. We perform the gradient updates using a single backward pass of autograd using the stop-gradient (sg) operator as shown in Algorithm 1.

### 3.3. Sampling

As shown in Fig. 1 on the bottom right, the generator network predicts the rates $\lambda^{\theta,i}_{\text{in},t}(x_t)$ for insertion at gap position $i$ and $\lambda^{\theta,i}_{\text{um},t}(x_t) \, K^{\theta,i}(\cdot|x_t)$ for unmasking at masked position $i$. We use a simple $\tau$-leaping sampler (Gillespie, 2001) that allows us to sample multiple events simultaneously. Additionally, the guarantee for the correctness of adaptive sampling from FlexMDM (Kim et al., 2025a) extends to **Lo**FlexMDM. The complete sampling algorithm is shown in Algorithm 2.

## 4. Experiments

**Model.** For all our experiments, we use DDiT (Lou et al., 2023) with RoPE, and adaptive layer-norm (AdaLN) for conditioning on the time variable (Peebles & Xie, 2023). We implement the target and generator rates with MLP scalar heads on top of the transformer. For FlexMDM (baseline), we follow the setup in Kim et al. (2025a), which also uses DDiT architecture, but has a length prediction head for insertion instead of a rate prediction head. We experiment with a "separate" setting where the target rates are predicted by a separate auxiliary transformer, and with a memory-efficient "shared" setting where all heads share the same backbone transformer, which is updated by both the $\nabla_\theta$ and $\nabla_\phi$ components of the Eq. (6) gradient estimator. [3]

### 4.1. Graph Traversal

**Setup.** We use the star graphs dataset from Patel et al. (2025b). It consists of star shaped graphs with one junction node and multiple arms coming in and out of the junction node. The task is to produce a path from the given starting and terminal nodes. The input graph is stringified as edge pairs arranged in a random order (the order is random but remains fixed for a given graph). The input string is this stringified graph followed by the starting and terminal nodes, and the model needs to generate the path (edges in the correct order) connecting the starting and terminal

---

[3]We use xLM library (https://github.com/dhruvdcoder/xlm-core) for implementing the training and evaluation harness. See Appendix F.1 for more details about the experiment setup.

*Table 1.* Exact Match (%) results for star graph traversal tasks. ❄/🔥 =frozen/trainable; ✓/× = conf. enabled/disabled.

| Model | $b_{um}$ | $b_{in}$ | conf. | **medium** | **hard** |
|---|---|---|---|---|---|
| 1 ARM | | | N/A | 75.0 | 23.0 |
| 2 MDM | | | ✓ | 36.5 | 21.0 |
| 3 FlexMDM | ❄ | ❄ | × | 89.6 | 6.0 |
| 4 | ❄ | ❄ | ✓ | 91.3 | 7.4 |
| 5 **Lo**FlexMDM | 🔥 | 🔥 | × | 92.3 | 38.0 |
| 6 (Separate) | ❄ | 🔥 | × | 93.2 | 87.9 |
| 7 | ❄ | 🔥 | ✓ | 93.0 | 88.1 |
| 8 **Lo**FlexMDM | ❄ | 🔥 | × | 93.6 | 34.2 |
| 9 (Shared) | ❄ | 🔥 | ✓ | 93.2 | 69.7 |

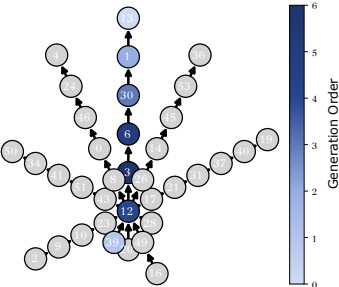
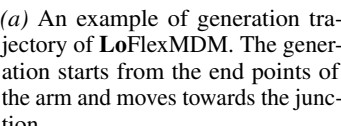

*(a)* An example of generation trajectory of **Lo**FlexMDM. The generation starts from the end points of the arm and moves towards the junction.

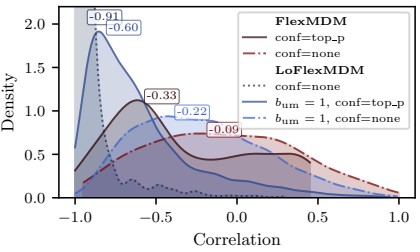

*(b)* Correlation between generation order and distance from the junction node, both normalized by path length. Only examples that achieve 100% exact match are considered.

nodes. We use the medium and hard difficulty settings of the dataset as both of these use graphs with variable arm lengths. The graphs in the hard setting vary significantly in the number of arms as well as the arm lengths, making it a challenging task. However, if the edges of the output path are generated starting from the end points moving towards the junction node, then the task becomes trivial even for a local model, i.e., a model that does not plan ahead.

### 4.1.1. DESIGN CHOICES

We use the graph traversal to explore the design choices for parameterizing the target hazard rates $\lambda_t^{\phi,i}(z; z_1)$.

**Flexibility *vs* Stability**: As noted in Section 3.1.1, the hazard rates have have two free parameters for each position, $b_{in}^i$ and $b_{um}^i$. We first examine the effect of flexibility of the target rate model $\lambda_t^{\phi,i}(z; z_1)$. Interestingly, the performance of the generator improves when the *unmasking* target rate $b_{um}$ is fixed (❄) as opposed to trainable (🔥), as seen from the slight improvement in medium difficulty and a significant improvement in hard difficulty of **Lo**FlexMDM (Table 1, line 5 *vs* line 6). We find that the additional flexibility in learning $b_{um}$ destabilizes training. Appendix G.1.1 discusses this observation in more detail. Note that fixing $b_{um}$ still leaves enough room for learning of insertion order, i.e., if $h_1 < h_2$, then for any $t$, $\mathbb{P}(T_{um} \le t \mid T_{in} = h_1) < \mathbb{P}(T_{um} \le t \mid T_{in} = h_2)$. The preference for unmasking orders will show up in $K_t^{\theta,i}$ values, which can be used for confidence-based position selection during decoding. Based on this observation, we fix $b_{um} = 1$ in the subsequent experiments.

**Shared *vs* Separate Backbone**: Additionally, we explore the use of a shared DDiT backbone for the target rate model and generator. For the hard version of the star graph traversal task, using a shared backbone leads to considerable drop in performance.

### 4.1.2. ANALYSIS: GENERATION ORDER

We investigate whether the introduction of learnable target rates leads to generations in near optimal order. The optimal *local* order for star graphs starts from the end points of the arm and moves towards the junction. We analyze examples of the complete history of the generation under **Lo**FlexMDM and FlexMDM. As shown in the example in Figure 4a, **Lo**FlexMDM indeed learns to generate in the optimal order. To inspect this further, we compute the correlation between the generation order and distance from the junction node. Since the arm lengths vary significantly for the hard version of the task, we normalize both the generation order and distance from the junction node by the path length. Figure 4b shows the correlation between the generation order (normalized by path length) and normalized distance from the junction node. The ideal correlation for optimal local order should be -1. Interestingly, we observe that **Lo**FlexMDM with trainable $b_{um}$ (line 5 in Table 1) achieves mean correlation of $-0.91$, when compared to $-0.22$ for the case when we fix $b_{um} = 1$ (line 6) and do not use confidence-based position selection during inference (line 7 in Table 1). Using confidence-based position selection for the fixed $b_{um} = 1$ case achieves a mean correlation of $-0.60$. This supports our initial observation in Section 4.1.1 that fixing $b_{um} = 1$ stabilizes the training while still providing enough room for learning an ideal insertion order, which can be elicited even in the steps which use confidence-based position selection for unmasking. Interestingly, using confidence-based position selection for FlexMDM changes the correlation from $-0.09$ (light red) to $-0.33$ (dark red), lower than the respective correlation for **Lo**FlexMDM with fixed $b_{um} = 1$ (light blue to dark blue). In Appendix G.1.2, we repeat this analysis without the exact-match filter to show that filtering on exact match does not alter the conclusion of the order analysis: correlations are slightly weaker without the filter, but the relative ordering across settings is unchanged.

## 4.2. Small Molecule Generation

**Setup.** We train all the models on the SAFE dataset (Noutahi et al., 2024), which consists of molecules from ZINC (Irwin et al., 2012) and UniChem (Chambers et al.,

*Table 2.* Results for *de novo* small molecule generation. For both FlexMDM and **Lo**FlexMDM, we use 1024 sampling steps. ❄ indicates frozen, 🔥 indicates trainable; ✓/✗ indicates whether confidence-based position selection for unmasking is enabled, and $p$ indicates the truncation threshold for nucleus sampling (Holtzman et al., 2020). † Noutahi et al. (2024) ‡ Lee et al. (2025)

| Model | $b_{um}$ | $b_{in}$ | conf. | $p$ | Validity (%) | Diversity | Uniqueness (%) | Quality (%) |
|---|---|---|---|---|---|---|---|---|
| SAFE-GPT† | | | | | 94.0 ±0.4 | 0.879 ±0.001 | **100.0** ±0.0 | 54.7 ±0.3 |
| MDM‡ | | | | | 96.7 ±0.3 | 0.896 ±0.001 | 99.3 ±0.2 | 53.8 ±1.7 |
| FlexMDM | ❄ | ❄ | ✗ | | 98.9 ±0.1 | 0.890 ±0.000 | 99.6 ±0.1 | 62.0 ±0.7 |
| | ❄ | ❄ | ✓ | | 67.8 ±0.3 | **0.940** ±0.000 | 61.7 ±0.7 | 5.5 ±0.4 |
| **Lo**FlexMDM (Aux. Size=medium) | ❄ | 🔥 | ✓ | 1.0 | 99.0 ±0.0 | 0.900 ±0.000 | 99.3 ±0.2 | 55.4 ±0.3 |
| | ❄ | 🔥 | ✓ | 0.5 | **99.9** ±0.0 | 0.830 ±0.000 | 93.2 ±0.6 | 69.3 ±0.7 |
| | ❄ | 🔥 | ✗ | 1.0 | 99.1 ±0.1 | 0.900 ±0.000 | 99.6 ±0.1 | 55.1 ±0.6 |
| | ❄ | 🔥 | ✗ | 0.5 | 99.7 ±0.1 | 0.850 ±0.000 | 99.5 ±0.1 | **79.5** ±0.7 |
| **Lo**FlexMDM (Aux. Size=small) | ❄ | 🔥 | ✓ | 1.0 | 99.1 ±0.1 | 0.900 ±0.000 | 99.4 ±0.1 | 55.5 ±0.7 |
| | ❄ | 🔥 | ✓ | 0.5 | 99.7 ±0.1 | 0.820 ±0.000 | 93.6 ±0.5 | 73.3 ±0.6 |
| | ❄ | 🔥 | ✗ | 1.0 | 99.2 ±0.1 | 0.900 ±0.000 | 99.6 ±0.2 | 55.4 ±0.7 |
| | ❄ | 🔥 | ✗ | 0.5 | 99.7 ±0.1 | 0.850 ±0.000 | 99.4 ±0.1 | 79.3 ±0.5 |
| **Lo**FlexMDM (Aux. Size=xtiny) | ❄ | 🔥 | ✓ | 1.0 | 99.0 ±0.1 | 0.900 ±0.000 | 99.7 ±0.0 | 55.8 ±0.7 |
| | ❄ | 🔥 | ✓ | 0.5 | 99.7 ±0.1 | 0.830 ±0.000 | 93.4 ±0.3 | 72.2 ±0.6 |
| | ❄ | 🔥 | ✗ | 1.0 | 99.0 ±0.1 | 0.900 ±0.000 | 99.8 ±0.1 | 55.1 ±0.7 |
| | ❄ | 🔥 | ✗ | 0.5 | 99.7 ±0.1 | 0.850 ±0.000 | 99.6 ±0.2 | 79.4 ±0.6 |

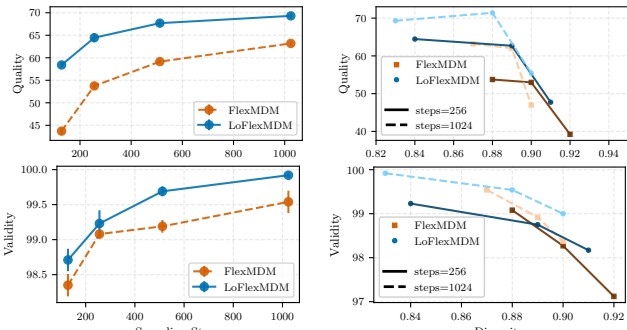

*Figure 5.* **Left**: Quality *vs* sampling steps for *de novo* generation for **Lo**FlexMDM and FlexMDM. **Right**: Quality *vs* diversity for *de novo* generation for **Lo**FlexMDM and FlexMDM.

2013a). Following Lee et al. (2025), we train for 50k steps with a global batch size of 2048. Based on the results of the graph traversal experiments, we fix the unmasking rate $b_{um}^\phi = b_{um}^\theta = 1$ and only keep the target insertion rate $b_{in}^\phi$ trainable. We evaluate on *de novo* generation by sampling 1000 molecules unconditionally from the trained models with sampling budget of 1024 steps and repeat the evaluation 5 times with different random seeds. We compute the metrics of **validity**, **uniqueness**, **diversity**, and **quality**. Appendix F.2.4 provides more details about the metrics, sampling protocol and baseline settings. Table 2 shows the results, and we provide a detailed analysis below.

**Fixed window vs. variable length**: For MDM, following Lee et al. (2025) we sample the lengths of the masked chunks from the ZINC250k distribution even for *de novo* generation. However, for FlexMDM and **Lo**FlexMDM, we perform completely unconstrained generation starting from a blank canvas. This highlights the advantage of working with a variable-length model.

**Diversity-quality trade-off**: As seen in Table 2, **Lo**FlexMDM improves quality over FlexMDM (79.5% vs. 62.0% under each model's best decoding settings), with a slight drop in diversity (0.850 vs. 0.890). Validity is also slightly higher (99.7% vs. 98.9%). This trade-off is expected: learning structurally good generation orders reduces randomness. We analyze the learned ordering explicitly in Section 4.2.1. Figure 5 (right) plots quality and validity against diversity as we vary the nucleus-sampling cutoff $p$ (Algorithm 2, line 23). **Lo**FlexMDM quality rises from ~55% at $p$=1.0 to ~80% at $p$=0.5 as randomness decreases. FlexMDM stays near ~62–63% over the same range. In the same vein, uniqueness drops slightly but validity improves when we use confidence-based position selection during decoding (lines 19–21) rather than sampling uniformly from eligible positions (line 17), making decoding more greedy.

**Effect of inference budget**: Next we fix the sampling parameters to their respective best values and vary the sampling steps. As seen in Figure 5 (left), **Lo**FlexMDM shows consistent and significant improvement over FlexMDM in quality and validity at all sampling budgets. Results with additional nucleus-sampling $p$ values, as well as a shared-backbone implementation, are provided in Table 10 in Appendix G.

### 4.2.1. ANALYSIS: GENERATION ORDER

Unlike star graphs (Section 4.1.2), where all the datapoints share a global ordering heuristic, which is independent of the token identities, molecular strings do not have such a global ordering heuristic. We therefore perform a different

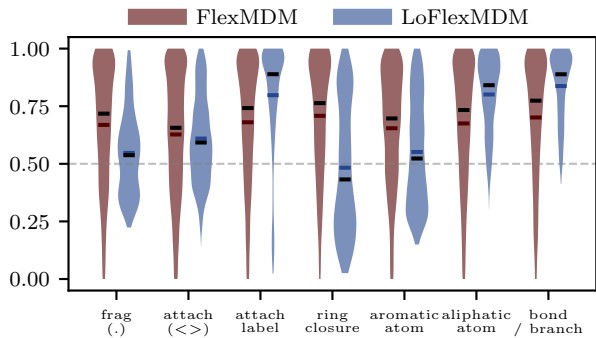

*Figure 6.* Normalised generation time by SAFE token category for FlexMDM (left) and **Lo**FlexMDM (right). For each violin, the coloured horizontal line marks the mean and the black line marks the median.

analysis for molecules: classify each token in the final string into one of seven semantic categories and measure when each category tends to appear during generation.

We analyze the 1000 unconditional *de novo* samples used for Table 2. Each token in the final Bracket SAFE string is assigned to one of seven categories: fragment separator ( . ), attachment bracket (</>), attachment label (the digit inside <>), ring closure (digits outside <>), aromatic atom, aliphatic atom, or bond/branch token (=, #, (, ) ). For each token we record the generation step at which it is finalized and min–max normalize that step within each molecule to $[0, 1]$, so that 0 denotes the earliest tokens and 1 the latest.

As seen in Figure 6, FlexMDM shows roughly uniform timing across categories, with mean generation times for different categories between $0.65$ and $0.71$. **Lo**FlexMDM instead generates the structure first (e.g., ring closures and fragment separators), and then fills in the chemistry (e.g., aliphatic atoms, attachment labels, and bond/branch tokens). **Lo**FlexMDM also separates the existence of an attachment point ($\mu \approx 0.61$) from its identity ($\mu \approx 0.80$), committing to where fragments connect before deciding which fragments connect. These observations are consistent with the qualitative examples shown in Figure 7, where we provide generation trajectories for FlexMDM and **Lo**FlexMDM with token type marked by different colors (see caption for the details). Appendix G.2.1 shows an additional example.

### 4.3. Constrained Generation

Following Lee et al. (2025), we evaluate on four fragment-constrained benchmarks from the SAFE-GPT protocol: linker design, motif extension, scaffold decoration, and superstructure generation. In each task, the model completes a bracket SAFE string from a fixed fragment prefix that specifies part of the target molecule. We use the same *de novo* checkpoints from Section 4.2, without any task-specific finetuning, and use the best sampling parameters for each model. Additional details about the evaluation protocol are

*Table 3.* **Fragment-constrained molecule generation results.** Means over 5 runs (Std. in Appendix 7). Tasks are abbreviated as LD (linker design), ME (motif extension), SD (scaffold decoration), and SG (superstructure generation). The best results are highlighted in bold and the second-best mean in each column is underlined. († Noutahi et al. (2024) ‡; Lee et al. (2025))

| Model | Task | Val. | Div. | Uniq. | Qual. |
|---|---|---|---|---|---|
| ARM[†] | LD | 76.6 | 0.545 | 82.5 | 21.7 |
| | ME | 96.1 | 0.562 | 66.8 | 18.6 |
| | SD | 97.7 | 0.575 | 74.7 | 10.0 |
| | SG | 95.7 | 0.573 | 83.0 | 14.3 |
| MDM[‡] | LD | **100.0** | 0.547 | **83.7** | 21.9 |
| | ME | 82.9 | 0.617 | 77.5 | 30.1 |
| | SD | 96.6 | 0.591 | 82.7 | 31.8 |
| | SG | 97.5 | 0.599 | **83.6** | 34.8 |
| FlexMDM | LD | 99.7 | **0.599** | 63.7 | 50.8 |
| | ME | 99.7 | **0.623** | 79.9 | 46.9 |
| | SD | 99.6 | **0.615** | 84.3 | 39.0 |
| | SG | 99.7 | **0.616** | 74.5 | 35.8 |
| **Lo**FlexMDM | LD | 99.6 | 0.576 | 64.4 | **51.7** |
| | ME | **99.9** | 0.608 | 79.2 | **53.6** |
| | SD | 99.8 | 0.601 | 82.6 | **40.5** |
| | SG | **100.0** | 0.593 | 72.6 | **37.0** |

provided in Appendix F.2.4. As seen in Table 3, validity is near saturation for FlexMDM and **Lo**FlexMDM on all four tasks, so quality is again the most informative metric. **Lo**FlexMDM achieves the best quality on every task. The gain over FlexMDM is largest on motif extension (53.6% vs. 46.9%). The same pattern holds on linker design (51.7% vs. 50.8%), scaffold decoration (40.5% vs. 39.0%), and superstructure generation (37.0% vs. 35.8%). As in *de novo* generation, **Lo**FlexMDM trades a small amount of diversity for higher quality. These results show that learned order dynamics also help under fragment constraints.

## 5. Discussion

**Summary of results.** Our experiments conclusively show that learned order dynamics improve the variable-length unmasking model in structured generation tasks. On star graph traversal, **Lo**FlexMDM learns to generate paths from the endpoints toward the junction, and this order leads to large gains on the hard setting. On molecules, the learned order is less global but still structured: **Lo**FlexMDM tends to resolve ring closures, fragment separators, and attachment structure before filling in labels, aliphatic atoms, and bond or branch tokens. These learned schedules significantly improve the quality of generated samples in both *de novo* and fragment-constrained generation.

**Limitations and outlook.** Using a shared backbone is more attractive for scaling to larger models for domains like text; however, this setting in our experiments shows comparatively smaller gains. Bootstrapping orders from the generator alone remains a promising direction for future work. Narrowing to a small number of orders reduces the uncertainty for the generator over the action space, which

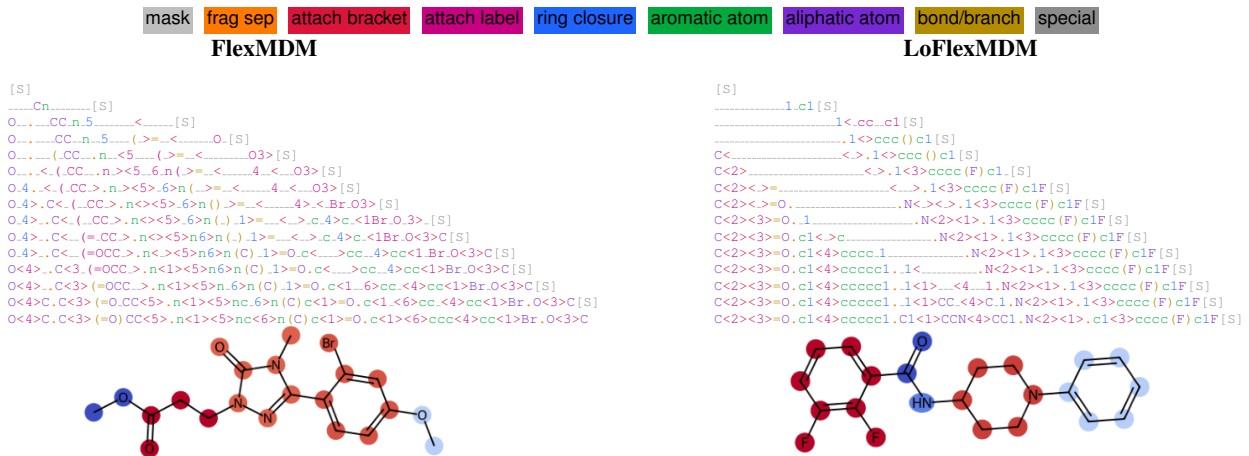

*Figure 7.* **Example *de novo* generation trajectory.** FlexMDM (left) and **Lo**FlexMDM (right) on the same molecule. Every line is a subsampled generation step; grey _ marks a still-masked position. Tokens are coloured by semantic category (legend above). The molecule graph below each trajectory colours atoms by normalised generation time (blue = early, red = late). **Lo**FlexMDM first resolves ring closures and fragment separators, then places attachment brackets and aromatic atoms, and fills in aliphatic atoms, attachment labels, and bond/branch tokens last. FlexMDM shows no such systematic pattern. An additional example is in Appendix G.2.1.

leads to faster generation. However, we still observe idle steps in the generation process (time progresses but the sequence does not change), suggesting significant room for speed improvement in variable-length unmasking models. For molecules specifically, SAFE representation, due to its permutation invariance in the fragment ordering, allows large equivalence classes of strings to represent the same molecule, which exacerbates the problem of idle steps. Using more compact string representations for molecules could show even greater gains in speed and quality.

## 6. Related Work

For discrete diffusion, there exist different *fixed* noising processes like uniform noising, masking, or a mixture of both (Sahoo et al., 2024; Schiff et al.; Zheng et al., 2025). For masked diffusion, Kim et al. (2025b) show that the order of denoising is crucial for the generative abilities of diffusion models, and the typical noising process that uses a uniform masking schedule may not be able to exploit the structure present in discrete sequences, thereby limiting the generator's ability. Wang et al. (2025) looks at learning of generation order in any-order models (Hoogeboom et al., 2022; Uria et al., 2013) where the position information is absolute and a single token is generated at a time.

The useful desiderata of *i)* modeling distributions over variable-length sequences and *ii)* the ability to insert tokens have inspired a number of recent works to move away from fixed length text diffusion towards insertion and substitution based models in which positions during the generation are treated as relative rather than absolute. Patel et al. (2025b) propose Insertion Language Model (ILM) that generates sequences by jointly modeling the distribution over the insertion location and the vocabulary. Havasi et al. (2025) pro-

pose EditFlows that uses the projected DFM to construct a generator CTMC that models the rates of insertion, deletion, and substitution operations. Both these models can only insert one token per gap at a time. FlexMDM (Kim et al., 2025a) can insert multiple mask tokens per gap. But the flexibility comes at a cost. The action space of insertion-based generation is larger than that of unmasking based models, and there have been limited attempts to explore learning the generation order in insertion-based models. Most of the exiting works focus on models that insert one token at a time and require simulating the generation trajectory for learning the generation order (Gu et al., 2019; Brantley et al., 2019). Our approach is general enough to be applied to all three settings: fixed-length unmasking, variable-length unmasking, and insertion only models, and combines well with multi-token prediction. We provide additional discussion in Appendix H.

## 7. Conclusion

We introduce **Lo**FlexMDM, a insertion-based masked diffusion model, that learns data-dependent insertion and unmasking rates. We demonstrate that the model successfully learns meaningful task-specific and data-dependent generation orders for graph traversal and molecule generation. The experiments on *de novo* and fragment-constrained molecule generation demonstrate that learned order dynamics lead to significant improvements in the validity and quality of generated samples compared to the baseline that does not use learned order dynamics.

## Acknowledgements

Dhruvesh and Benjamin acknowledge support from IBM under IBM Research Collaboration Agreement No. W1668553

and from the National Science Foundation under grant IIS-2106391. All the authors are grateful to the anonymous reviewers for their helpful suggestions to improve the paper.

## Impact Statement

This paper presents work whose goal is to advance the field of Machine Learning. There are many potential societal consequences of our work, none which we feel must be specifically highlighted here.

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

# Appendix for Insertion Based Sequence Generation with Learnable Order Dynamics

# A. Summary of Notation

Table 4 summarizes the notation used throughout the paper.

| Notation | Description |
|---|---|
| **General** | |
| $\mathbb{X}, \mathbb{Z}$ | Sets (blackboard font) |
| $[\![n]\!]$ | Set of natural numbers $\{1, 2, \ldots, n\}$ |
| $x_t$ | Particular state at time $t$ of a stochastic process $X_t$ |
| $x_t^i, s^i, w^i$, etc. | $i$-th component of $x, s, w$ respectively, i.e. superscript $i$ denotes the $i$-th component in a sequence |
| $\delta_x(y) = \delta_{\{x\}}(y)$ | Indicator function that is 1 if $x = y$ and 0 otherwise |
| $p_t(x)$ | The marginal law of CTMC $X_t$ at time $t$, i.e., $p_t(x) = \mathbb{P}(X_t = x)$ |
| $p_{t\|s}(x\|y)$ | The conditional law of CTMC $X_t$ at time $t$ given the value at time $s$, i.e., $p_{t\|s}(x\|y) = \mathbb{P}(X_t = x\|X_s = y)$ |
| $p_{t\|C}(x\|c)$ | The conditional law of CTMC $X_t$ at time $t$ given the conditioning variable $C = c$, i.e., $p_{t\|C}(x\|c) = \mathbb{P}(X_t = x\|C = c)$ |
| $R_t(x, y)$ | The rate of the transition from $x$ to $y$ at time $t$, i.e., $R_t(x, y) = \lim_{h \to 0} \frac{\mathbb{P}(X_{t+h} = y\|X_t = x) - \delta_x(y)}{h}$ |
| $R_t(x, y\|c)$ | The conditional rate of the transition from $x$ to $y$ at time $t$ given the conditioning variable $C = c$. $R_t(x, y\|c)$ *generates* $p_{t\|C}(x\|c)$ in the sense of Definition 1. |
| **Specific variables** | |
| `[D]` | Drop token |
| `[M]` | Mask token |
| $\mathbb{V}$ | Vocabulary of tokens including the `[M]` token but excluding the `[D]` token |
| $\mathbb{V}^n$ | Set of token sequences of length $n$, i.e., $\mathbb{V}^n = \mathbb{V} \times \overset{n \text{ times}}{\cdots} \times \mathbb{V}$ |
| $\mathbb{X}$ | Space of variable length sequences, i.e., $\mathbb{X} = \bigcup_{k=1}^{L} \mathbb{V}^k$ |
| $\mathbb{Z}$ | Auxiliary space of partial sequences, i.e., $\mathbb{Z} = \bigcup_{k \in [\![L]\!]} \mathbb{V}^k \times \mathbb{S}_k$, where $\mathbb{S}_k := \{(s^1, \ldots, s^k) \in [\![L]\!]^k \mid s^1 < \cdots < s^k\}$ is the space of ordered tuples of positions |
| $\bar{\mathbb{Z}}$ | Auxiliary space of partial sequences with the `[D]` token added to the vocabulary, i.e., $\bar{\mathbb{Z}} = (\mathbb{V} \cup \{\texttt{[D]}\})^L$, with isomorphism $f_{\text{contract}} : \bar{\mathbb{Z}} \to \mathbb{Z}$ that removes the `[D]` tokens from the $\bar{z}$ to form $x$, and moves the exact position information to $s$ to produce $z = (x, s)$ |
| $R_t^i(z, w^i \mid (z_1, z_0))$ | The per-position target rate of the transition from $z$ to $w^i$ at time $t$ given the conditioning variable $(z_1, z_0)$, i.e., $R_t^i(z, w^i \mid (z_1, z_0)) = \lambda_t^i(z; (z_1, z_0)) \cdot K_t^i(z, w^i \mid (z_1, z_0))$ |
| $\lambda_t^i(z; (z_1, z_0))$ | The state and time dependent arrival rate of the Poisson process at the $i$-th position at time $t$ given the conditioning variable $(z_1, z_0)$ |
| $K_t^i(z, w^i \mid (z_1, z_0))$ | The state transition kernel at the $i$-th position at time $t$ given the conditioning variable $(z_1, z_0)$, i.e., when an arrival occurs at the $i$-th position, the state changes according to the categorical distribution $K_t^i$ |
| $T_{\text{in}}^i := T_1^i$ | The event time of the insertion event at the $i$-th position |
| $T_{\text{um}}^i := T_2^i$ | The event time of the unmasking event at the $i$-th position |

*Table 4.* Summary of notation used throughout the paper.

# B. Motivation for Insertion-based Generation

In this section, we provide a motivating example for the type of sequence generation tasks where viewing the sequence elements in a relative order and generating them through insertions with appropriate insertion order is more natural than using a left-to-right autoregressive generator or an arbitrary order generator using absolute positions. Figure 8 shows an example of a star graph with variable arm lengths. A left-to-right autoregressive model is poorly matched to this task. Most edges along each arm are easy to predict locally, so teacher forcing provides little signal at the junction, where the model must choose the correct continuation without explicit lookahead. An arbitrary-order model that generates using absolute positions faces a different limitation. The path is most naturally built from each arm's endpoint inward toward the junction, but unmasking at fixed indices in that order requires predicting the path length precisely in advance, another challenging problem in itself.

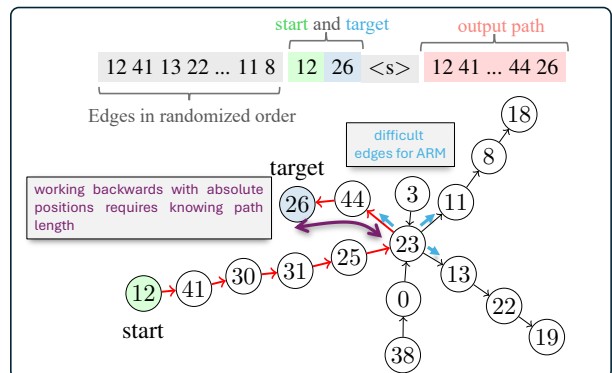

*Figure 8.* An example of a star graph with variable arm lengths from the hard subset of the star graph traversal task from Patel et al. (2025b). The top section shows the string representation of the graph, where the input sequence consists of edges in a random order, followed by the starting and terminal nodes. The output part is the path (edges in the correct order) connecting the starting and terminal nodes. *The figure is adapted from Patel et al. (2025b)* with permission.

## C. Background: Discrete Flow Matching

### C.1. Markov Processes on Discrete Spaces

Let $\mathbb{X}$ be countable set (e.g. the vocabulary $\mathbb{V}$ or the space of sequences of length $L$ with tokens from $\mathbb{V}$), and $\{X_t\}$ be a continuous time stochastic process on $\mathbb{X}$.

$$p_{t+h|t}(x_{t+h}|x_t) := \mathbb{P}(X_{t+h} = x_{t+h}|X_t = x_t) = \delta_{x_t}(x_{t+h}) + h\, R_t(x_t, x_{t+h}) + o(h), \tag{7}$$

where $R_t(x, y)$ is the *rate* of the transition from $x$ to $y$ at time $t$, such that $R_t(x, x) = -\sum_{y \neq x} R_t(x, y)$. Equivalently, $R_t(x, y) = \lim_{h \to 0} \frac{\mathbb{P}(X_{t+h} = y | X_t = x) - \delta_x(y)}{h}$.

The Kolmogorov Forward Equation (KFE) is the central continuity equation for CTMCs. It connects the marginal probabilities to the rate:

$$\partial_t p_t(x) = \sum_y R_t(y, x)\, p_t(y). \tag{8}$$

**Definition 1** (Time marginals generated by CTMC). We note that the following statements are equivalent (Campbell et al., 2024):

- $R_t(\cdot, \cdot)$ **generates** $p_t$

- $R_t(\cdot, \cdot)$ and $p_t$ satisfy the KFE in the marginal form (Eq. (8))

- A Markov process $X_t$ with rates $R_t(\cdot, \cdot)$ and initial distribution $p_0$ has $p_t$ as its marginal distribution at time $t$

### C.1.1. INDEXED COLLECTIONS OF CTMCS

**Definition 2** (Conditional Rates). For a general variable $C \in \mathcal{C}$ with distribution $\pi$,

- denote $C$ dependent rates as $R_t(x, y|C)$

- denote by $\{\{X_t^c\}\}_{c \in \mathcal{C}}$ a collection of CTMCs on $\mathbb{X}$ with rates $R_t(x, y|C)$ and initial distribution $p_0(x|C)$

- $p_{t|C}(x|c) := \mathbb{P}(X_t^c = x)$ denote the marginals of the CTMC $\{X_t^c\}$

In the special case of endpoint indexed collection, we have $C = (X_0, X_1)$, $X_1 \sim p_{\text{data}}$ and $X_0 \sim \pi(x_0|x_1)$, where $\pi$ is the coupling distribution (Gat et al., 2024).

**Proposition 2** (Mixture of CTMCs). *The mixture rate $R_t(x, \cdot) = \mathbb{E}_{C|X_t=x}[R_t(x, \cdot|C)]$ generates the marginals $p_t(x) = \mathbb{E}_C[p_{t|C}(x|C)]$.*

The proof assumes mild regularity conditions and follows from a straightforward application of the KFE (see Campbell et al. (2024) or Lipman et al. (2024) Thm. 14 for details). This result shows there exists a rate matrix $R_t(\cdot, \cdot) = \mathbb{E}_C[R_t(\cdot, \cdot|C)]$ and the corresponding CTMC $\{X_t\}$ such that $R_t$ *generates* $p_t(x) = \mathbb{E}_C[p_{t|C}(x|C)]$. Therefore, in order to sample from the data distribution $p_1$, we can follow:

1. Pick the indexing set $C = (X_0, X_1)$

2. Use $p_1 = p_{\text{data}}$ and pick the coupling distribution $\pi(x_0|x_1)$, typically independent i.e. $\pi(x_0|x_1) = p_0(x_0)$ for some easy to sample $p_0$.

3. Construct a collection of CTMCs $\{X_t^c\}_{c \in \mathcal{C}}$ with rates $R_t(x, y|C)$ such that it *generates* $p_{t|C}$, which satisfies $p_{1|C}(x|x_1, x_0) = \delta_{x_1}(x)$ and $p_{0|C}(x|x_0, x_0) = \delta_{x_0}(x)$.

4. The marginal $p_t(x) = \mathbb{E}_C[p_{t|C}(x|C)]$ satisfies $p_1(x) = p_{\text{data}}(x)$ and Proposition 2 ensures that the CTMC with the rate $R_t(\cdot, \cdot) = \mathbb{E}_C[R_t(\cdot, \cdot|C)]$ *generates* the marginals $p_t(x) = \mathbb{E}_C[p_{t|C}(x|C)]$.

5. Construct an optimization problem to learn parametric $R_t^\theta(\cdot, \cdot)$.

Next we look at the optimization objective for learning the rates $R_t$.

### C.2. Discrete Flow Matching

We want to learn the rates $R_t(x, y)$ using a parametric model $R_t^\theta(x, y)$. Since we can easily write down the conditional rates $R_t(x, y|z)$ we can use appropriate loss function to match $R_t^\theta(x, \cdot)$ with $R_t(x, \cdot) = \mathbb{E}_{z \sim p_{Z|t}(\cdot|x)} R_t(x, \cdot|z)$. That is for some distance measure $D$, we want to minimize:

$$\mathcal{L}(\theta) = \mathop{\mathbb{E}}_{X_t \sim p_t} D\big( \mathop{\mathbb{E}}_{C \sim p_{C|t}(\cdot|X_t)} R_t(X_t, \cdot|C),\ R_t^\theta(X_t, \cdot) \big),$$

where the expectations are over $x_t \sim p_t$ and $C \sim \pi(\cdot|x_t)$.

In most cases, this objective is not useful in this form, but for Bregman divergence, we have Holderrieth et al. (2025) convexity in the first argument and linearity of the gradient w.r.t convex combinations in the first argument. Recall the definition of Bregman divergence $D_F(b, a)$ defined for any convex function $F$ is given by:

$$D_F(b, a) = F(b) - F(a) - \nabla F(a) \cdot (b - a), \tag{9}$$

where $\nabla F(a)$ is the gradient of $F$ at $a$.

**Convexity in the first argument** It is convex in the first argument. Meaning that $\lambda D_F(b_1, a) + (1 - \lambda) D_F(b_2, a) \geq D_F(\lambda b_1 + (1 - \lambda) b_2, a)$ for all $\lambda \in [0, 1]$. This is easy to see:

$$
\begin{aligned}
D_F(\lambda b_1 + (1 - \lambda)b_2, a) &= F(\lambda b_1 + (1 - \lambda)b_2) - F(a) - \nabla F(a) \cdot (\lambda b_1 + (1 - \lambda)b_2 - a) \\
&\leq \lambda F(b_1) + (1 - \lambda)F(b_2) - F(a) - \nabla F(a) \cdot (\lambda b_1 + (1 - \lambda)b_2 - a) \\
&= \lambda[F(b_1) - F(a) - \nabla F(a) \cdot (b_1 - a)] \\
&\quad + (1 - \lambda)[F(b_2) - F(a) - \nabla F(a) \cdot (b_2 - a)] \\
&= \lambda D_F(b_1, a) + (1 - \lambda)D_F(b_2, a)
\end{aligned}
\tag{10}
$$

**Gradient is linear for convex combinations in the first argument.** Its gradient with respect to the second argument is **linear** w.r.t affine combinations in the first argument (Holderrieth et al., 2025). This is a consequence of the expression for the gradient w.r.t the second argument, which is given by

$$\nabla_a D_F(b, a) = -\nabla^2 F(a)(b - a), \tag{11}$$

where $\nabla^2 F(a)$ is the Hessian of $F$ at $a$. Clearly, $f(b) = \nabla_a D_F(b, a)$ is linear in $b$. Therefore, under convex combinations, which can be extended to expectations, we get:

$$\nabla_a D_F(\lambda b_1 + (1 - \lambda)b_2, a) = \lambda \nabla_a D_F(b_1, a) + (1 - \lambda)\nabla_a D_F(b_2, a)$$
$$\nabla_a[D_F(\mathbb{E}\,Y, a)] = \mathbb{E}[\nabla_a D_F(Y, a)]. \tag{12}$$

Therefore for $C = X_1$ we can use the following objective to train the parameters of $R_t^\theta(x, y)$:

$$\mathcal{L}(\theta) = \mathbb{E}_{X_1 \sim p_{\text{data}},\, X_0 \sim p_{t|X_1}} [D_F(R_t(X, \cdot | X_1),\ R_t^\theta(X, \cdot))]. \tag{13}$$

## D. Projected Discrete Flow Matching

In this section, we will discuss a type of *weak* intertwining of two Markov processes that depends on the law of the main process. This is a generalization of the flow matching with auxiliary variables (Havasi et al., 2025) and the joint interpolants (Kim et al., 2025a), in that we allow probabilistic mappings between the two spaces.

Let $\mathbb{X}$ and $\mathbb{Z}$ be two countable state spaces, and let $\pi_{\text{enc}}(x|z)$ be probability kernel from $\mathbb{Z}$ to $\mathbb{X}$.

**Definition 3.** Let $\{Z_t\}$ be a CTMC on $\mathbb{Z}$ with rate matrix $R_t$ and initial distribution $p_0(z)$. Then the CTMC $\{X_t\}$ on $\mathbb{X}$ with rate matrix $R_t^\pi$ is said to be a *weakly intertwined projection* of $Z_t$ under the encoder $\pi_{\text{enc}}$ if

$$R_t^\pi(x, y) = \sum_{z \in \mathbb{Z}} \sum_{w \in \mathbb{Z}} \pi_{\text{enc}}(y|w)\pi_{\text{dec,t}}(z|x)R_t(z, w), \tag{14}$$

where $\pi_{\text{dec,t}}(z|x)$ is the time and law dependent Bayes' decoder given by:

$$\pi_{\text{dec,t}}(z|x) = \frac{p_t(z)\pi_{\text{enc}}(x|z)}{p_t(x)}, \quad p_t(x) > 0, \tag{15}$$

where $p_t(x) := \mathbb{P}(X_t = x)$ and $p_t(z) := \mathbb{P}(Z_t = z)$ are the marginal laws of $\{X_t\}$ and $\{Z_t\}$ respectively.

**Proposition 3** ($R_t^\pi$ generates $p_t(x)$)**.** *Let $\{X_t\}$ be a weakly intertwined projection of $\{Z_t\}$ under the encoder $\pi_{enc}$. Then $R_t^\pi$ as defined in Eq. (14) generates $p_t(x)$ in the sense of Definition 1.*

*Proof.* We can easily see that $R_t^\pi$ generates $p_t(x) = \mathbb{E}_{z \sim p_t}[\pi_{\text{enc}}(x|z)]$ by checking the KFE:

$$\partial_t p_t(x) = \partial_t \sum_{z \in \mathbb{Z}} p_t(z)\pi_{\text{enc}}(x|z)$$

$$= \sum_{z \in \mathbb{Z}} \partial_t p_t(z)\pi_{\text{enc}}(x|z)$$

$$= \sum_{z \in \mathbb{Z}} \left[\sum_{w \in \mathbb{Z}} p_t(w)R_t(w, z)\right] \pi_{\text{enc}}(x|z)$$

$$= \sum_{z \in \mathbb{Z}} \left[\sum_{w \in \mathbb{Z}} \sum_{y \in \mathbb{X}} \pi_{\text{dec,t}}(w|y)p_t(y)R_t(w, z)\right] \pi_{\text{enc}}(x|z) \quad \text{(by Eq. (15))}$$

$$= \sum_{y \in \mathbb{X}} R_t^\pi(y, x)\, p_t(y) \quad \text{(by Eq. (14))} \tag{16}$$

$\square$

Similarly, for any conditioning variable $C$, we can define the $C$ dependent rates analogous to Definition 2 as:

$$R_t^\pi(x, y|C) = \sum_{z \in \mathbb{Z}} \sum_{w \in \mathbb{Z}} \pi_{\text{enc}}(y|w) \pi_{\text{dec,t}}(z|x, C) R_t(z, w|C), \tag{17}$$

where the Bayes decoder is given by:

$$\pi_{\text{dec,t}}(z|x, C) = \frac{p_{t|C}(z|C) \pi_{\text{enc}}(x|z)}{p_{t|C}(x|C)}, \quad p_{t|C}(x|C) > 0. \tag{18}$$

We can show that with $p_{t|C}(x|C) := \sum_{z \in \mathbb{Z}} p_{t|C}(z|C) \pi_{\text{enc}}(x|z)$,

$$\partial_t p_{t|C}(x|C) = \sum_y R_t^\pi(y, x|C)\, p_{t|C}(y|C). \tag{19}$$

That is, $R_t^\pi(x, y|C)$ *generates* $p_{t|C}(x|C)$:

$$\begin{aligned}
\partial_t p_{t|C}(x|C) &= \partial_t \sum_{z \in \mathbb{Z}} p_{t|C}(z|C) \pi_{\text{enc}}(x|z) \\
&= \sum_{z \in \mathbb{Z}} \partial_t p_{t|C}(z|C) \pi_{\text{enc}}(x|z) \\
&= \sum_{z \in \mathbb{Z}} \left[ \sum_{w \in \mathbb{Z}} p_{t|C}(w|C) R_t(w, z|C) \right] \pi_{\text{enc}}(x|z) \\
&= \sum_{z \in \mathbb{Z}} \left[ \sum_{w \in \mathbb{Z}} \sum_{y \in \mathbb{X}} \pi_{\text{dec,t}}(w|y, C) p_{t|C}(y|C) R_t(w, z|C) \right] \pi_{\text{enc}}(x|z) \quad \text{(by Eq. (18))} \\
&= \sum_{y \in \mathbb{X}} R_t^\pi(y, x|C)\, p_{t|C}(y|C) \quad \text{(by Eq. (17))}.
\end{aligned} \tag{20}$$

*Remark* 3 (Degenerate (Dirac) decoders preserve the KFE). The Bayes decoder $\pi_{\text{dec,t}}(z|x, C)$ in Eq. (17) averages over all latent states $z$ compatible with the observed state $x$ (and condition $C$). As we will see later that we can additionally condition on *side information* $H$ (e.g., an alignment/position tuple) such that the latent state can be selected deterministically from $(x, H)$. In this case we can use a degenerate decoder

$$\pi_{\text{dec,t}}(z|x, C, H) = \delta_z(\varphi(x, H)). \tag{21}$$

Defining projected rates using Eq. (17) with the enlarged condition $(C, H)$, the same calculation as above shows that the KFE remains satisfied:

$$\partial_t p_{t|C,H}(x|C, H) = \sum_y R_t^\pi(y, x|C, H)\, p_{t|C,H}(y|C, H). \tag{22}$$

If desired, one can marginalize out $H$ via the mixture identity $R_t^\pi(x, y|C) = \mathbb{E}_{H \sim p_{H|t}(\cdot|x,C)}[R_t^\pi(x, y|C, H)]$.

Following is the key result stating that the projected rates generate the desired time marginals just like vanilla DFM without auxiliary variables.

**Proposition 4** ($R_t^\pi$ generates $p_t$). *Let $\{X_t\}$ be a weakly intertwined projection of $\{Z_t\}$ under the encoder $\pi_{\text{enc}}$. Then $R_t^\pi$ as defined in Eq. (14) generates $p_t$ in the sense of Definition 1.*

*Proof.* We show that $R_t^\pi(x, y) := \mathbb{E}_{C \sim p_{C|t}(\cdot|x)} R_t^\pi(x, y|C)$ *generates* $p_t(x) = \mathbb{E}_C\, p_{t|C}(x|C)$ by checking the KFE. (In particular, this argument applies verbatim if we enlarge the conditioning variable from $C$ to $(C, H)$ to incorporate auxiliary

side information $H$.)

$$
\begin{aligned}
\partial_t p_t(x) &= \sum_C p(C) \partial_t p_{t|C}(x|C) \\
&= \sum_C p(C) \sum_y R_t^\pi(y, x|C) \, p_{t|C}(y|C) \quad \text{(by Eq. (19))} \\
&= \sum_y \sum_C R_t^\pi(y, x|C) \, p_{C|t}(C|y) p_t(y) \\
&= \sum_y R_t^\pi(y, x) \, p_t(y).
\end{aligned}
\tag{23}
$$

$\square$

Proposition 4 provides us a way to construct the desired probability path on $\mathbb{X}$ that satisfies the boundary conditions $p_0 = p$ and $p_1 = p_{\text{data}}$, by constructing conditional rates $R_t(z, w|C)$ in the auxiliary space $\mathbb{Z}$ and selecting appropriate encoder $\pi_{\text{enc}}$ to obtain $R_t^\pi(x, y|C)$. This approach comes in handy when it is easier to construct probability path on the auxiliary space $\mathbb{Z}$ than the original space $\mathbb{X}$.

### D.0.1. PROJECTED RATE MATCHING

Even though the probability paths are constructed in the auxiliary space, we still want to learn the rates in the original space $\mathbb{X}$.

$$
\mathbb{E}_{X_t, C} D_F(R_t^\pi(X_t, \cdot|C), R_t^\theta(X_t, \cdot))
$$

The question, however, is whether this is the same as

$$
\mathbb{E}_{X_t} D_F(R_t^\pi(X_t, \cdot), R_t^\theta(X_t, \cdot)).
$$

Due to the convexity in the first argument of Bregman divergence, we see that the two expressions are not equal in general: the first one is an upper bound for the second one.

Recall from earlier that $R_t^\pi(x, y) = \mathbb{E}_{C \sim p_{C|t}(\cdot|x)} R_t^\pi(x, y|C)$. Using this, we can rewrite the second expression as:

$$
\mathbb{E}_{X_t} D_F(R_t^\pi(X_t, \cdot), R_t^\theta(X_t, \cdot)) = \mathbb{E}_{X_t} D_F\left( \mathbb{E}_{C|X_t} R_t^\pi(X_t, \cdot|C), R_t^\theta(X_t, \cdot) \right).
$$

Since $D_F(b, a)$ is convex in the first argument $b$, by Jensen's inequality we have:

$$
D_F\left( \mathbb{E}_{C|X_t} R_t^\pi(X_t, \cdot|C), R_t^\theta(X_t, \cdot) \right) \le \mathbb{E}_{C|X_t} \left[ D_F(R_t^\pi(X_t, \cdot|C), R_t^\theta(X_t, \cdot)) \right].
$$

Taking expectations over $X_t$ on both sides:

$$
\begin{aligned}
\mathbb{E}_{X_t} D_F\left( \mathbb{E}_{C|X_t} R_t^\pi(X_t, \cdot|C), R_t^\theta(X_t, \cdot) \right) &\le \mathbb{E}_{X_t} \left[ \mathbb{E}_{C|X_t} D_F(R_t^\pi(X_t, \cdot|C), R_t^\theta(X_t, \cdot)) \right] \\
&= \mathbb{E}_{X_t, C} D_F(R_t^\pi(X_t, \cdot|C), R_t^\theta(X_t, \cdot)).
\end{aligned}
$$

Therefore:

$$
\mathbb{E}_{X_t} D_F(R_t^\pi(X_t, \cdot), R_t^\theta(X_t, \cdot)) \le \mathbb{E}_{X_t, C} D_F(R_t^\pi(X_t, \cdot|C), R_t^\theta(X_t, \cdot)).
\tag{24}
$$

**Proposition 5** (Projected rate matching upper bounds terminal KL). *Let $\{X_t\}_{t\in[0,1]}$ be a time-inhomogeneous CTMC on $\mathbb{X}$ with initial distribution $p_0$ and (marginal) projected rates $R_t^\pi(\cdot,\cdot)$. Let $\{\hat{X}_t\}$ be another CTMC on $\mathbb{X}$ with the same initial distribution $p_0$ and model rates $R_t^\theta(\cdot,\cdot)$. Assume absolute continuity of the jump intensities: for all $t \in [0,1]$ and $x \neq y$, $R_t^\pi(x,y) > 0$ implies $R_t^\theta(x,y) > 0$, and assume the expectations below are finite.*

*Let $\mathbb{P}^\pi$ and $\mathbb{P}^\theta$ denote the corresponding path measures on $[0,1]$, and let $p_1^\pi$ and $p_1^\theta$ be the terminal marginals of $X_1$ and $\hat{X}_1$. Define $D_F$ to be the Bregman divergence generated by $F(a) = \sum_{y\neq x} a_y \log a_y$ applied to the off-diagonal rate vectors $R_t(x,\cdot) \coloneqq (R_t(x,y))_{y\neq x}$. Then:*

$$\mathcal{D}_{KL}\big(p_1^\pi \,\big\|\, p_1^\theta\big) \leq \mathcal{D}_{KL}\big(\mathbb{P}^\pi \,\big\|\, \mathbb{P}^\theta\big) = \int_0^1 \mathop{\mathbb{E}}_{\mathbb{P}^\pi}\Big[D_F\big(R_t^\pi(X_t,\cdot),\ R_t^\theta(X_t,\cdot)\big)\Big]\, \mathrm{d}t. \tag{25}$$

*Moreover, if the projected rates arise as a conditional mixture $R_t^\pi(x,\cdot) = \mathbb{E}_{C\sim p_{C|t}(\cdot|x)}[R_t^\pi(x,\cdot|C)]$, then for every $t$,*

$$\mathop{\mathbb{E}}_{\mathbb{P}^\pi}\Big[D_F\big(R_t^\pi(X_t,\cdot),\ R_t^\theta(X_t,\cdot)\big)\Big] \leq \mathop{\mathbb{E}}_{(X_t,C)}\Big[D_F\big(R_t^\pi(X_t,\cdot|C),\ R_t^\theta(X_t,\cdot)\big)\Big], \tag{26}$$

*where $C \sim p_{C|t}(\cdot|X_t)$ in the right-hand side. Consequently,*

$$\mathcal{D}_{KL}\big(p_1^\pi \,\big\|\, p_1^\theta\big) \leq \int_0^1 \mathop{\mathbb{E}}_{(X_t,C)}\Big[D_F\big(R_t^\pi(X_t,\cdot|C),\ R_t^\theta(X_t,\cdot)\big)\Big]\, \mathrm{d}t. \tag{27}$$

*Proof.* The inequality $\mathcal{D}_{\mathrm{KL}}(p_1^\pi\|p_1^\theta) \leq \mathcal{D}_{\mathrm{KL}}(\mathbb{P}^\pi\|\mathbb{P}^\theta)$ follows by data processing inequality, since $X_1$ is a measurable function of the full path. The path-space identity is the standard Radon–Nikodym derivative formula for time-inhomogeneous CTMCs, which yields $\mathcal{D}_{\mathrm{KL}}(\mathbb{P}^\pi\|\mathbb{P}^\theta) = \int_0^1 \mathbb{E}_{\mathbb{P}^\pi}[D_F(R_t^\pi(X_t,\cdot), R_t^\theta(X_t,\cdot))]\, \mathrm{d}t$ when $F(a) = \sum a \log a$ is applied to the off-diagonal jump intensities. Finally, the conditional-mixture upper bound is exactly Eq. (24) applied pointwise in $t$, and integrating over $t$ gives the final inequality. For a simple proof using discretization arguments, see Shaul et al. (2025). □

# E. Insertion-based masked diffusion with learnable order dynamics

### E.1. Target rates (Proof for Proposition 1)

In order to construct the target probability paths, we will set $h = (z_0, z_1)$, where $z_1 \sim p_1(z_1)$, where $p_1(z_1)$ is the distribution over $\bar{\mathbb{Z}}$ obtained by padding the ground truth sequences $x_1 \sim p_{\mathrm{data}}(x_1)$ with [D] tokens to length $L$, and $z_0 \sim p_0(z_0)$, where $p_0(z_0)$ places all mass on the sequence of length $L$ with all [D] tokens.

Now we have two times $T_{\mathrm{in}}$ and $T_{\mathrm{um}}$ that need to be sampled ensuring that $T_{\mathrm{in}} < T_{\mathrm{um}}$, i.e., insertion happens before unmasking.

**Notation:** In the following, the CDFs $F_{\mathrm{in}}^{\phi,i}(t; z_1)$ and $F_{\mathrm{um}}^{\phi,i}(t; z_1)$ depend on the learnable parameters $\phi$ and the target sequence $z_1$; for brevity we suppress this dependence in the notation and write $F_{\mathrm{in}}^i(t)$ and $F_{\mathrm{um}}^i(t)$ throughout this proof.

We first choose an insertion time distribution by specifying its CDF:

$$\mathbb{P}(T_{\mathrm{in}}{}^i \leq t) = F_{\mathrm{in}}^i(t), \qquad f_{T_{\mathrm{in}}{}^i}(t) = \frac{\mathrm{d}}{\mathrm{d}t} F_{\mathrm{in}}^i(t). \tag{28}$$

such that $F_{\mathrm{in}}^i(0) = 0$ and $F_{\mathrm{in}}^i(1) = 1$. We then define the conditional distribution of $T_{\mathrm{um}}{}^i$ given $T_{\mathrm{in}}{}^i = s$ by **truncating** a base distribution of a random variable $U^i$ on $[0,1]$ to $[s,1]$. Let the base distribution have CDF $F_{\mathrm{um}}^i(u)$, again such that $F_{\mathrm{um}}^i(0) = 0$ and $F_{\mathrm{um}}^i(1) = 1$. Throughout this proof, we use the density notation $f_{T_{\mathrm{in}}{}^i}(t) \equiv \dot{F}_{\mathrm{in}}^i(t)$ and $f_{U^i}(t) \equiv \dot{F}_{\mathrm{um}}^i(t)$ (or equivalently $\frac{d}{dt}F_*^i(t)$), consistent with the proposition's dot notation. Then

$$\mathbb{P}(T_{\mathrm{um}}{}^i \leq t \mid T_{\mathrm{in}}{}^i = s) = \delta(t \geq s)\left[\frac{\mathbb{P}(U^i \leq t) - \mathbb{P}(U^i \leq s)}{1 - \mathbb{P}(U^i \leq s)}\right], \tag{29}$$

$$f_{T_{\mathrm{um}}{}^i | T_{\mathrm{in}}{}^i = s}(t) = \delta(t \geq s)\frac{f_{U^i}(t)}{\mathbb{P}(U^i > s)} = \delta(t \geq s)\frac{\dot{F}_{\mathrm{um}}^i(t)}{1 - F_{\mathrm{um}}^i(s)} \tag{30}$$

for $t \in [0,1]$. This construction guarantees $T_{\mathrm{in}}{}^i < T_{\mathrm{um}}{}^i$ almost surely.

**Conditional probability paths:** Viewing $z$ and $z_1$ as elements of $\bar{\mathbb{Z}}$, per-token conditional probability paths in the augmented space $\bar{\mathbb{Z}}$ are given by:

$$p_t^{\phi,i}(z^i|z_1,z_0) = \underbrace{\mathbb{P}(T_{\text{in}}{}^i > t)\,\delta_{[\text{D}]}(z^i)}_{p_t^{\phi,i}([\text{D}]|z_1,z_0)}$$
$$+ \underbrace{\mathbb{P}(T_{\text{in}}{}^i \le t < T_{\text{um}}{}^i)\,\delta_{[\text{M}]}(z^i)(1 - \delta_{z_1^i}([\text{D}]))}_{p_t^{\phi,i}([\text{M}]|z_1,z_0)}$$
$$+ \underbrace{\mathbb{P}(T_{\text{um}}{}^i \le t)\,\delta_{z_1^i}(z^i)}_{p_t^{\phi,i}(z_1^i|z_1,z_0)}, \tag{31}$$

where $T_{\text{in}}, T_{\text{um}}$ depend on $z_1$ and $\phi$ through Eqs. (28) and (29), and $z_0 = [\text{D}]$ always. Note the additional indicator function $(1 - \delta_{z_1^i}([\text{D}]))$ ensures that if $z_1^i = [\text{D}]$ (i.e. a pad to the right of the clean sequence), then no change happens.

**Target conditional rates** Fix a token position $i$ and assume $z_1^i \ne [\text{D}]$ (otherwise the process stays at $[\text{D}]$ and all rates are 0). At this position, the process has three states $\{[\text{D}], [\text{M}], z_1^i\}$ with transitions

$$[\text{D}] \xrightarrow{T_{\text{in}}{}^i} [\text{M}] \xrightarrow{T_{\text{um}}{}^i} z_1^i,$$

where the first jump happens at time $T_{\text{in}}{}^i$ and the second at time $T_{\text{um}}{}^i$. We will take the time derivative of Eq. (31) to obtain the LHS of the KFE Eq. (8). Going term by term, we have the following

**Derivative 1:**

$$\partial_t \mathbb{P}(T_{\text{in}}{}^i > t) = \partial_t(1 - F_{\text{in}}^i(t)) = -f_{T_{\text{in}}{}^i}(t) \tag{32}$$
$$= -\frac{f_{T_{\text{in}}{}^i}(t)}{\mathbb{P}(T_{\text{in}}{}^i > t)}\,\mathbb{P}(T_{\text{in}}{}^i > t) \tag{33}$$
$$:= -\lambda_{\text{in}}^i(t)\,\mathbb{P}(T_{\text{in}}{}^i > t) \tag{34}$$

**Derivative 2:** Due to the partition $\{T_{\text{in}}{}^i \le t\} = \{T_{\text{in}}{}^i \le t < T_{\text{um}}{}^i\} \sqcup \{T_{\text{um}}{}^i \le t\}$, we have

$$\partial_t \mathbb{P}(T_{\text{in}}{}^i \le t < T_{\text{um}}{}^i) = \partial_t \mathbb{P}(T_{\text{in}}{}^i \le t) - \partial_t \mathbb{P}(T_{\text{um}}{}^i \le t) \tag{35}$$
$$= -\partial_t \mathbb{P}(T_{\text{in}}{}^i > t) - \partial_t \mathbb{P}(T_{\text{um}}{}^i \le t) \tag{36}$$
$$= \lambda_{\text{in}}^i(t)\,\mathbb{P}(T_{\text{in}}{}^i > t) - f_{T_{\text{um}}{}^i}(t) \tag{37}$$

**Derivative 3:**

$$\partial_t \mathbb{P}(T_{\text{um}}{}^i \le t) = \frac{\mathrm{d}}{\mathrm{d}t}\int_0^t \mathbb{P}(T_{\text{um}}{}^i \le t \mid T_{\text{in}}{}^i = s)f_{T_{\text{in}}{}^i}(s)\mathrm{d}s \tag{38}$$

$$= \mathbb{P}(T_{\text{um}}{}^i \le t \mid T_{\text{in}}{}^i = t)f_{T_{\text{in}}{}^i}(t) + \int_0^t \frac{\partial}{\partial t}\mathbb{P}(T_{\text{um}}{}^i \le t \mid T_{\text{in}}{}^i = s)f_{T_{\text{in}}{}^i}(s)\mathrm{d}s \tag{39}$$

$$= 0 \cdot f_{T_{\text{in}}{}^i}(t) + \int_0^t f_{T_{\text{um}}{}^i|T_{\text{in}}{}^i=s}(t)\,f_{T_{\text{in}}{}^i}(s)\mathrm{d}s \tag{40}$$

$$= f_{T_{\text{um}}{}^i}(t) \tag{41}$$

$$= \frac{f_{T_{\text{um}}{}^i}(t)}{\mathbb{P}(T_{\text{in}}{}^i \le t < T_{\text{um}}{}^i)}\,\mathbb{P}(T_{\text{in}}{}^i \le t < T_{\text{um}}{}^i) \tag{42}$$

$$:= \lambda_{\text{um}}^i(t)\,\mathbb{P}(T_{\text{in}}{}^i \le t < T_{\text{um}}{}^i) \tag{43}$$

where we define the unmasking rate as

$$\lambda_{\text{um}}^i(t) := \frac{f_{T_{\text{um}}{}^i}(t)}{\mathbb{P}(T_{\text{in}}{}^i \le t < T_{\text{um}}{}^i)}. \tag{44}$$

Substituting back into Derivative 2:

$$\partial_t \mathbb{P}(T_{\text{in}}{}^i \leq t < T_{\text{um}}{}^i) = \lambda_{\text{in}}^i(t)\,\mathbb{P}(T_{\text{in}}{}^i > t) - \lambda_{\text{um}}^i(t)\,\mathbb{P}(T_{\text{in}}{}^i \leq t < T_{\text{um}}{}^i) \tag{45}$$

Putting all three derivatives together, from Eq. (31) we have:

$$\partial_t p_t^{\phi,i}(z^i|z_1, z_0) = -\lambda_{\text{in}}^i(t)\,\mathbb{P}(T_{\text{in}}{}^i > t)\delta_{[\text{D}]}(z^i) \tag{46}$$

$$+ \left[\lambda_{\text{in}}^i(t)\,\mathbb{P}(T_{\text{in}}{}^i > t) - \lambda_{\text{um}}^i(t)\,\mathbb{P}(T_{\text{in}}{}^i \leq t < T_{\text{um}}{}^i)\right]\delta_{[\text{M}]}(z^i) \cdot \bar{\delta}_{z_1^i}([\text{D}]) \tag{47}$$

$$+ \lambda_{\text{um}}^i(t)\,\mathbb{P}(T_{\text{in}}{}^i \leq t < T_{\text{um}}{}^i)\delta_{z_1^i}(z^i) \tag{48}$$

$$= \lambda_{\text{in}}^i(t)\,\mathbb{P}(T_{\text{in}}{}^i > t)\left[\delta_{[\text{M}]}(z^i)\bar{\delta}_{z_1^i}([\text{D}]) - \delta_{[\text{D}]}(z^i)\right] \tag{49}$$

$$+ \lambda_{\text{um}}^i(t)\,\mathbb{P}(T_{\text{in}}{}^i \leq t < T_{\text{um}}{}^i)\left[\delta_{z_1^i}(z^i) - \delta_{[\text{M}]}(z^i)\bar{\delta}_{z_1^i}([\text{D}])\right] \tag{50}$$

For the sake of compactness, from here on, we will not write $\bar{\delta}_{z_1^i}([\text{D}])$ explicitly, as it is only present to handle the case of $z_1^i = [\text{D}]$.

$$\partial_t p_t^{\phi,i}(z^i|z_1, z_0) = \lambda_{\text{in}}^i(t)p_t^{\phi,i}([\text{D}]|z_1, z_0)\left[\delta_{[\text{M}]}(z^i) - \delta_{[\text{D}]}(z^i)\right] \tag{51}$$

$$+ \lambda_{\text{um}}^i(t)p_t^{\phi,i}([\text{M}]|z_1, z_0)\left[\delta_{z_1^i}(z^i) - \delta_{[\text{M}]}(z^i)\right]$$

$$= \sum_{y^i \in \{[\text{D}],[\text{M}],z_1^i\}}\left[\lambda_{\text{in}}^i(t)\left(\delta_{[\text{M}]}(z^i) - \delta_{y^i}(z^i)\right)\delta_{[\text{D}]}(y^i)\right.$$

$$\left. + \lambda_{\text{um}}^i(t)\left(\delta_{z_1^i}(z^i) - \delta_{y^i}(z^i)\right)\delta_{[\text{M}]}(y^i)\right]p_t^{\phi,i}(y^i|z_1, z_0)$$

$$= \sum_{y^i \in \mathbb{V} \cup \{[\text{D}],[\text{M}]\}}\left[\lambda_{\text{in}}^i(t)\left(\delta_{[\text{M}]}(z^i) - \delta_{y^i}(z^i)\right)\delta_{[\text{D}]}(y^i)\right.$$

$$\left. + \lambda_{\text{um}}^i(t)\left(\delta_{z_1^i}(z^i) - \delta_{y^i}(z^i)\right)\delta_{[\text{M}]}(y^i)\right]p_t^{\phi,i}(y^i|z_1, z_0) \tag{52}$$

Comparing with the KFE, we get data-conditional per-token rates:

$$R_t^{\phi,i}(y^i, z^i|z_1) = \lambda_{\text{in}}^i(t)\left[\delta_{[\text{M}]}(z^i) - \delta_{y^i}(z^i)\right]\delta_{[\text{D}]}(y^i)$$

$$+ \lambda_{\text{um}}^i(t)\left[\delta_{z_1^i}(z^i) - \delta_{y^i}(z^i)\right]\delta_{[\text{M}]}(y^i), \tag{53}$$

which describes $[\text{D}] \to [\text{M}]$ at rate $\lambda_{\text{in}}^i(t)$ (insertion) and $[\text{M}] \to z_1^i$ at rate $\lambda_{\text{um}}^i(t)$ (unmasking).

The only thing left before we can use this target rate for training is to compute the rates $\lambda_{\text{in}}^i(t)$ and $\lambda_{\text{um}}^i(t)$ in closed form if possible. The rates $\lambda_{\text{in}}^i(t)$ and $\lambda_{\text{um}}^i(t)$ can be written in closed form in terms of the insertion density $f_{T_{\text{in}}^i}$ and the base unmasking density $\dot{F}_{\text{um}}^i$ (and its CDF $F_{\text{um}}^i$), without committing to any particular functional family.

**Insertion rate** By definition,

$$\lambda_{\text{in}}^i(t) := \frac{f_{T_{\text{in}}^i}(t)}{\mathbb{P}(T_{\text{in}}{}^i > t)} = \frac{f_{T_{\text{in}}^i}(t)}{1 - F_{\text{in}}^i(t)}. \tag{54}$$

**Unmasking rate** Under the truncation construction Eq. (29), we have for $t \geq s$:

$$\mathbb{P}(T_{\text{um}}{}^i > t \mid T_{\text{in}}{}^i = s) = 1 - \mathbb{P}(T_{\text{um}}{}^i \leq t \mid T_{\text{in}}{}^i = s) \tag{55}$$

$$= 1 - \frac{F_{\text{um}}^i(t) - F_{\text{um}}^i(s)}{1 - F_{\text{um}}^i(s)} \tag{56}$$

$$= \frac{1 - F_{\text{um}}^i(t)}{1 - F_{\text{um}}^i(s)}. \tag{57}$$

The unconditional density of $T_{\text{um}}{}^i$ is

$$
\begin{aligned}
f_{T_{\text{um}}{}^i}(t) &= \int_0^t f_{T_{\text{um}}{}^i \mid T_{\text{in}}{}^i = s}(t)\, f_{T_{\text{in}}{}^i}(s)\, \mathrm{d}s \\
&= \int_0^t \frac{\dot{F}^i_{\text{um}}(t)}{1 - F^i_{\text{um}}(s)}\, f_{T_{\text{in}}{}^i}(s)\, \mathrm{d}s \\
&= \dot{F}^i_{\text{um}}(t) \int_0^t \frac{f_{T_{\text{in}}{}^i}(s)}{1 - F^i_{\text{um}}(s)}\, \mathrm{d}s.
\end{aligned}
\tag{58}
$$

The masked-state probability (the denominator of $\lambda^i_{\text{um}}$) can be written as

$$
\begin{aligned}
\mathbb{P}(T_{\text{in}}{}^i \leq t < T_{\text{um}}{}^i) &= \int_0^t \mathbb{P}(T_{\text{um}}{}^i > t \mid T_{\text{in}}{}^i = s)\, f_{T_{\text{in}}{}^i}(s)\, \mathrm{d}s \\
&= \int_0^t (1 - \mathbb{P}(T_{\text{um}}{}^i \leq t \mid T_{\text{in}}{}^i = s))\, f_{T_{\text{in}}{}^i}(s)\, \mathrm{d}s \\
&= \int_0^t \frac{1 - F^i_{\text{um}}(t)}{1 - F^i_{\text{um}}(s)}\, f_{T_{\text{in}}{}^i}(s)\, \mathrm{d}s \\
&= (1 - F^i_{\text{um}}(t)) \int_0^t \frac{f_{T_{\text{in}}{}^i}(s)}{1 - F^i_{\text{um}}(s)}\, \mathrm{d}s.
\end{aligned}
\tag{59}
$$

Therefore the integral cancels in the ratio, and

$$
\lambda^i_{\text{um}}(t) := \frac{f_{T_{\text{um}}{}^i}(t)}{\mathbb{P}(T_{\text{in}}{}^i \leq t < T_{\text{um}}{}^i)} = \frac{\dot{F}^i_{\text{um}}(t)}{1 - F^i_{\text{um}}(t)}.
\tag{60}
$$

### E.2. Likelihood under $\phi$

In order to implement the gradient estimator from Eq. (6), we will need to compute $p_t^{\phi,i}(z^i \mid z_1, z_0)$. From Eq. (31), the conditional probability path is:

$$
\begin{aligned}
p_t^{\phi,i}(z^i \mid z_1, z_0) &= \mathbb{P}(T_{\text{in}}{}^i > t)\, \delta_{[\text{D}]}(z^i) + \mathbb{P}(T_{\text{in}}{}^i \leq t < T_{\text{um}}{}^i)\, \delta_{[\text{M}]}(z^i)(1 - \delta_{z_1^i}([\text{D}])) \\
&\quad + \mathbb{P}(T_{\text{um}}{}^i \leq t)\, \delta_{z_1^i}(z^i).
\end{aligned}
\tag{61}
$$

For pad positions with $z_1^i = [\text{D}]$, the process stays at $[\text{D}]$ and thus $p_t^{\phi,i}(z^i \mid z_1, z_0) = \delta_{[\text{D}]}(z^i)$. For positions with $z_1^i \neq [\text{D}]$, under the truncated construction Eq. (29) we can write the state probabilities as:

$$
\mathbb{P}(T_{\text{in}}{}^i > t) = 1 - F^i_{\text{in}}(t),
\tag{62}
$$

$$
\mathbb{P}(T_{\text{in}}{}^i \leq t < T_{\text{um}}{}^i) = (1 - F^i_{\text{um}}(t)) \int_0^t \frac{f_{T_{\text{in}}{}^i}(s)}{1 - F^i_{\text{um}}(s)}\, \mathrm{d}s,
\tag{63}
$$

$$
\mathbb{P}(T_{\text{um}}{}^i \leq t) = 1 - \mathbb{P}(T_{\text{in}}{}^i > t) - \mathbb{P}(T_{\text{in}}{}^i \leq t < T_{\text{um}}{}^i).
\tag{64}
$$

Substituting into the path definition yields:

$$
p_t^{\phi,i}(z^i \mid z_1, z_0) = \mathbb{P}(T_{\text{in}}{}^i > t)\, \delta_{[\text{D}]}(z^i) + \mathbb{P}(T_{\text{in}}{}^i \leq t < T_{\text{um}}{}^i)\, \delta_{[\text{M}]}(z^i)(1 - \delta_{z_1^i}([\text{D}])) + \mathbb{P}(T_{\text{um}}{}^i \leq t)\, \delta_{z_1^i}(z^i).
\tag{65}
$$

### E.3. Training

#### E.3.1. PROJECTED TARGET RATES

We will now apply the framework of projected rate matching from appendix D to the case of variable length masked diffusion (Kim et al., 2025a). Before proceeding, recall the key notation. For variable length masked diffusion, we have an augmented space $\bar{\mathbb{Z}}_L = (\mathbb{V} \cup \{[\text{D}], [\text{M}]\})^L$ with encoder $\pi_{\text{enc}}$ that removes $[\text{D}]$ tokens. The conditioning variable is

$z_1 = (x_1, s_1) \in \mathbb{Z}$, where $s_1^i = i$, $i \in [\text{len}(x_1)]$ are the positions in the ground truth sequence $x_1$. [4] Equivalently, for some intermediate time, $s_t \in \mathbb{S}_{\text{len}(x_t)}$ denotes the ordered tuple of non-[D] positions in $z_t$. That is, $s_t = (s_t^1, \ldots, s_t^{\text{len}(x_t)})$ where $s_t^1 < \cdots < s_t^{\text{len}(x_t)}$ are the positions in $z_t$ that contain non-[D] tokens. Then position $i$ in the original space $x_t$ corresponds to position $s_t^i$ in the augmented space $z_t$.

For $i \in \{0, 1, \ldots, \text{len}(x_t)\}$, we define the gap at position $i$ as the set of drop positions in the augmented space between original-space positions $i$ and $i+1$:

$$\text{gap}_i(s_t) := \{ j \in [L] : s_t^i < j < s_t^{i+1} \}, \tag{66}$$

where $s_t^0 := 0$ and $s_t^{\text{len}(x_t)+1} := L + 1$. These are the augmented positions where inserting a token would place it between positions $i$ and $i+1$ in the original space.

**Sequence-level rates:** For $x_t$ a subsequence of $x_1$, we condition on both $z_1$ and the position tuple $s_t \in \mathbb{S}_{\text{len}(x_t)}$ (Remark 3). The projected conditional rates (Eq. (17)) are:

$$R_t^\pi(x_t, y | z_1, s_t) = \sum_{w \in \bar{\mathbb{Z}}} \pi_{\text{enc}}(y | w) \, R_t((x_t, s_t), w | z_1), \tag{67}$$

where we use a point-mass decoder $\pi_{\text{dec},t}(z_t | x_t, z_1, s_t) = \delta_{z_t}((x_t, s_t))$ that fixes the augmented state to $z_t = (x_t, s_t)$. For any $w = (\tilde{y}, r)$, the encoder can be written explicitly as:

$$\pi_{\text{enc}}(y | w) = \delta_{f_{\text{rm-drop}}(w)}(y)$$
$$= \underbrace{\left( \prod_{i=1}^{\text{len}(\tilde{y})} \delta_{y^i}(\tilde{y}^{r^i}) \right)}_{\text{non-drop pos.}} \underbrace{\left( \prod_{i=1: i \notin r}^{L} \delta_{[D]}(\tilde{y}^{r^i}) \right)}_{\text{drop positions}}. \tag{68}$$

Using the assumption of per-position mixture, augmented space rates decompose as:

$$R_t((x_t, s_t), w | z_1) = \sum_{j=1}^{L} \delta_{(x_t, s_t)^{\neg j}}(w^{\neg j}) R_t^j(z_t^j, w^j | z_1), \tag{69}$$

where $z_t^j$ denotes the token at position $j$ in $z_t = (x_t, s_t)$. Substituting into Eq. (67):

$$R_t^\pi(x_t, y | z_1, s_t) = \sum_{w} \pi_{\text{enc}}(y | w) \sum_{j=1}^{L} \delta_{(x_t, s_t)^{\neg j}}(w^{\neg j}) R_t^j(z_t^j, w^j | z_1)$$
$$= \sum_{j=1}^{L} \sum_{w^j} \pi_{\text{enc}}(y | (x_t, s_t)^{\neg j}, w^j) \, R_t^j(z_t^j, w^j | z_1), \tag{70}$$

where we used $w = ((x_t, s_t)^{\neg j}, w^j)$, to mean that $w$ equals $(x_t, s_t)$ except at position $j$ where it contains $w^j$.

Splitting the sum over $j \in [L]$ into two parts based on whether position $j$ is a [D] position or not in $z_t = (x_t, s_t)$:

$$R_t^\pi(x_t, y | z_1, s_t) = \underbrace{\sum_{j \in \{s_t^1, \ldots, s_t^{\text{len}(x_t)}\}} \sum_{w^j} \pi_{\text{enc}}(y | (x_t, s_t)^{\neg j}, w^j) R_t^j(z_t^j, w^j | z_1)}_{\text{Non-[D] positions}}$$
$$+ \underbrace{\sum_{j \notin \{s_t^1, \ldots, s_t^{\text{len}(x_t)}\}} \sum_{w^j} \pi_{\text{enc}}(y | (x_t, s_t)^{\neg j}, w^j) R_t^j([D], w^j | z_1)}_{\text{[D] positions}}. \tag{71}$$

---

[4]Technically, the empty sequence $z_0 = \emptyset$ is also present in the condition but we will suppress it for brevity.

For non-$[D]$ positions, reindexing with $j = s_t^i$ for $i \in [\mathrm{len}(x_t)]$, and noting that $z_t^{s_t^i} = x_t^i$:

$$\text{Non-}[D] = \sum_{i=1}^{\mathrm{len}(x_t)} \sum_{w^{s_t^i}} \pi_{\mathrm{enc}}(y|(x_t, s_t)^{\neg s_t^i}, w^{s_t^i}) \, R_t^{s_t^i}(x_t^i, w^{s_t^i}|z_1)$$

$$\stackrel{(1)}{=} \sum_{i=1}^{\mathrm{len}(x_t)} \sum_{w^{s_t^i}} \delta_y((x_t^{\neg i}, w^{s_t^i})) \, R_t^{s_t^i}(x_t^i, w^{s_t^i}|z_1)$$

$$= \sum_{i=1}^{\mathrm{len}(x_t)} \delta_{y^{\neg i}}(x_t^{\neg i}) \, R_t^{s_t^i}(x_t^i, y^i|z_1) \tag{72}$$

$$\stackrel{(2)}{=} \sum_{i=1}^{\mathrm{len}(x_t)} \delta_{y^{\neg i}}(x_t^{\neg i}) \, \lambda_{\mathrm{um},t}^{s_t^i}(z_1) \left[ \delta_{z_1^{s_t^i}}(y^i) - \delta_{x_t^i}(y^i) \right] \delta_{[M]}(x_t^i) \tag{73}$$

where (1) uses the encoder definition Eq. (68), and (2) substitutes the target augmented rates from Eq. (53).

For the $[D]$ positions (where $j \notin \{s_t^1, \ldots, s_t^{\mathrm{len}(x_t)}\}$), we have $z_t^j = [D]$:

$$[D] \text{ pos.} = \sum_{j \notin s_t} \sum_{w^j} \pi_{\mathrm{enc}}(y|(x_t, s_t)^{\neg j}, w^j) \, R_t^j([D], w^j|z_1)$$

$$\stackrel{(1)}{=} \sum_{j \notin s_t} \pi_{\mathrm{enc}}(y|(x_t, s_t)^{\neg j}, [M]) \, \lambda_{\mathrm{in},t}^j(z_1)$$

$$\stackrel{(2)}{=} \sum_{i=0}^{\mathrm{len}(x_t)} \delta_y((x_t^{1:i}, [M], x_t^{i+1:\mathrm{len}(x_t)})) \sum_{j \in \mathrm{gap}_i(s_t)} \lambda_{\mathrm{in},t}^j(z_1), \tag{74}$$

where (1) uses the augmented target rates (Eq. (53)): $R_t^j([D], w^j|z_1) = \lambda_{\mathrm{in},t}^j(z_1)\delta_{[M]}(w^j)$, so only $w^j = [M]$ contributes (insertion). For (2), when $w^j = [M]$ at position $j \in \mathrm{gap}_i(s_t)$, the encoder requires $y = f_{\mathrm{rm\text{-}drop}}((x_t, s_t)^{\neg j}, [M])$ to have the form $(x_t^{1:i}, [M], x_t^{i+1:\mathrm{len}(x_t)})$, inserting $[M]$ at position $i$ in the original space.

We can express the non-drop and drop positions rates compactly by defining per-position rate components. For unmasking at position $i \in [\mathrm{len}(x_t)]$:

$$R_{\mathrm{um}}^{\pi,i}(x_t^i, y^i|z_1, s_t) := \lambda_{\mathrm{um},t}^{s_t^i}(z_1) \left[ \delta_{z_1^{s_t^i}}(y^i) - \delta_{x_t^i}(y^i) \right] \delta_{[M]}(x_t^i), \tag{75}$$

and for insertion at gap $i \in \{0, 1, \ldots, \mathrm{len}(x_t)\}$:

$$R_{\mathrm{in}}^{\pi,i}([M]|z_1, s_t) := \sum_{j \in \mathrm{gap}_i(s_t)} \lambda_{\mathrm{in},t}^j(z_1). \tag{76}$$

Therefore, the projected rates can be written as per-position mixture:

$$R_t^\pi(x_t, y|z_1, s_t) = \sum_{i=1}^{\mathrm{len}(x_t)} \delta_{y^{\neg i}}(x_t^{\neg i}) \, R_{\mathrm{um}}^{\pi,i}(x_t^i, y^i|z_1, s_t)$$

$$+ \sum_{i=0}^{\mathrm{len}(x_t)} \delta_y((x_t^{1:i}, [M], x_t^{i+1:\mathrm{len}(x_t)})) \, R_{\mathrm{in}}^{\pi,i}([M]|z_1, s_t). \tag{77}$$

*Remark* 4 (Relationship to marginalized rates:). The projected rate $R_t^\pi(x_t, y|z_1, s_t)$ conditioned on both $z_1$ and $s_t$ is related to the marginalized version by:

$$R_t^\pi(x_t, y|z_1) = \sum_{s_t \in \mathbb{S}_{\mathrm{len}(x_t)}} p_{t|z_1}(s_t|x_t, z_1) \, R_t^\pi(x_t, y|z_1, s_t). \tag{78}$$

Using Proposition 5, we can learn both $\phi$ and $\theta$ by minimizing

$$\mathcal{L}^{\theta,\phi} = \mathop{\mathbb{E}}_{X_t, Z_1, S_t} D_F(R_t^{\pi,\phi}(X_t, \cdot | Z_1, S_t), R_t^\theta(X_t, \cdot)), \tag{79}$$

where $Z_1 \sim p_{\text{data}}$ and $(X_t, S_t) \sim p_{t|Z_1}^\phi$.

### E.4. Loss function

We parameterize the generator rates $R_t^\theta(x_t, y)$ using per-position mixture

$$R^\theta(x_t, y) = \sum_{i=1}^{\text{len}(x_t)} \delta_{x_t^{\neg i}}(y^{\neg i}) \left[ R_{\text{um},t}^{\theta,i}(x_t, y^i) + R_{\text{in},t}^{\theta,i}(x_t, [\text{M}]) \right], \tag{80}$$

where $R_{\text{um},t}^{\theta,i}(x, y^i) = \lambda_{\text{um},t}^{\theta,i}(x) K_t^{\theta,i}(y^i|x_t)$ and $R_{\text{in},t}^{\theta,i}(x, [\text{M}]) = \lambda_{\text{in},t}^{\theta,i}(x)$.

We can now substitute the parametric rates into Eq. (79) to get the training objective.

$$\mathcal{L}^{\theta,\phi}(x_t, z_1, s_t) = \sum_{y \in \mathcal{Y}(x_t)} \left[ R_t^{\pi,\phi}(x_t, y|z_1, s_t) \log \frac{R_t^{\pi,\phi}(x_t, y|z_1, s_t)}{R_t^\theta(x_t, y)} - R_t^{\pi,\phi}(x_t, y|z_1, s_t) + R_t^\theta(x_t, y) \right], \tag{81}$$

where $\mathcal{Y}(x_t)$ is the set of sequences $y$ that differ from $x_t$ at exactly one position. Note that both terms depend on learnable parameters ($\phi$ and $\theta$ respectively), so there are no constant terms to drop.

Substituting the two-sum form Eq. (77), we obtain:

$$
\begin{aligned}
\mathcal{L}^{\theta,\phi}(x_t, z_1, s_t) = \mathop{\mathbb{E}}_{t, z_1, z_t \sim p_{t|z_1}^\phi} \Bigg[ & \sum_{i=1}^{\text{len}(x_t)} \delta_{[\text{M}]}(x_t^i) \sum_{y^i \in \mathbb{V}} \bigg( R_{\text{um}}^{\pi,\phi,i}(x_t^i, y^i|z_1, s_t) \log \frac{R_{\text{um}}^{\pi,\phi,i}(x_t^i, y^i|z_1, s_t)}{R_{\text{um},t}^{\theta,i}(x_t, y^i)} \\
& - R_{\text{um}}^{\pi,\phi,i}(x_t^i, y^i|z_1, s_t) + R_{\text{um},t}^{\theta,i}(x_t, y^i) \bigg) \\
& + \sum_{i=0}^{\text{len}(x_t)} \bigg( R_{\text{in}}^{\pi,\phi,i}([\text{M}]|z_1, s_t) \log \frac{R_{\text{in}}^{\pi,\phi,i}([\text{M}]|z_1, s_t)}{R_{\text{in},t}^{\theta,i}(x_t, [\text{M}])} \\
& - R_{\text{in}}^{\pi,\phi,i}([\text{M}]|z_1, s_t) + R_{\text{in},t}^{\theta,i}(x_t, [\text{M}]) \bigg) \Bigg].
\end{aligned}
$$

Expanding using the definitions of $R_{\text{um}}^{\pi,\phi,i}$, $R_{\text{in}}^{\pi,\phi,i}$, $R_{\text{um},t}^{\theta,i}$, $R_{\text{in},t}^{\theta,i}$ produces the final expression for the loss as stated in Eq. (5).

$$
\begin{aligned}
\mathcal{L}^{\theta,\phi} = \mathop{\mathbb{E}}_{t, z_1, z_t \sim p_{t|z_1}^\phi} \Bigg[ & \sum_{i=1}^{\text{len}(x_t)} \delta_{[\text{M}]}(x_t^i) \bigg( \lambda_{\text{um},t}^{\phi,s_t^i}(z_1) \log \frac{\lambda_{\text{um},t}^{\phi,s_t^i}(z_1)}{\lambda_{\text{um},t}^{\theta,i}(x_t) K_t^{\theta,i}(z_1^{s_t^i}|x_t)} \\
& - \lambda_{\text{um},t}^{\phi,s_t^i}(z_1) + \lambda_{\text{um},t}^{\theta,i}(x_t) \bigg) \\
& + \sum_{i=0}^{\text{len}(x_t)} \Bigg( \bigg( \sum_{j \in \text{gap}_i(s_t)} \lambda_{\text{in},t}^{\phi,j}(z_1) \bigg) \log \frac{\sum_{j \in \text{gap}_i(s_t)} \lambda_{\text{in},t}^{\phi,j}(z_1)}{\lambda_{\text{in},t}^{\theta,i}(x_t)} \\
& - \bigg( \sum_{j \in \text{gap}_i(s_t)} \lambda_{\text{in},t}^{\phi,j}(z_1) \bigg) + \lambda_{\text{in},t}^{\theta,i}(x_t) \Bigg) \Bigg]. \tag{82}
\end{aligned}
$$

### E.5. Gradient computation

In order to take gradient w.r.t $\phi$ present in the expectation, we can use REINFORCE leave one out as follows. We want to compute the following gradient:

$$\nabla_\phi \mathbb{E}_{z \sim p^\phi} f^\phi(z)$$

A simple REINFORCE style estimate will be:

$$\nabla_\phi \mathbb{E}_{z \sim p^\phi} f^\phi(z) = \mathbb{E}_{z \sim p^\phi} \left[ \nabla_\phi \log p^\phi(z) f^\phi(z) + \nabla_\phi f^\phi(z) \right]. \tag{83}$$

The first term will be high variance. We can introduce GRPO style baseline for the first term. Say we have $n$ samples $z_1, \ldots, z_n$ from $p^\phi$. Then we can define the baseline as:

$$b_i = \frac{1}{n-1} \sum_{j \neq i} f^\phi(z_j).$$

We can then do a multi-sample MC estimate with this baseline where each $z_i$ plays the role of $z$ once, giving us the following estimate for the first term:

$$\frac{1}{n} \sum_{i=1}^n \left[ f^\phi(z_i) - b_i \right] \nabla_\phi \log p^\phi(z_i)$$

$$= \frac{1}{n} \sum_{i=1}^n \left[ f^\phi(z_i) - \left( \frac{1}{n-1} \sum_{j \neq i} f^\phi(z_j) \right) \right] \nabla_\phi \log p^\phi(z_i)$$

Using $n = 2$ we get the following simple estimator:

$$\frac{1}{2} \sum_{i=1}^2 \left[ f^\phi(z_i) - \left( \frac{1}{2-1} \sum_{j \neq i} f^\phi(z_j) \right) \right] \nabla_\phi \log p^\phi(z_i)$$

$$= \frac{1}{2} \sum_{i=1}^2 \left[ f^\phi(z_i) - \sum_{j \neq i} f^\phi(z_j) \right] \nabla_\phi \log p^\phi(z_i)$$

$$= \frac{1}{2} \left\{ \left[ f^\phi(z_1) - f^\phi(z_2) \right] \nabla_\phi \log p^\phi(z_1) + \left[ f^\phi(z_2) - f^\phi(z_1) \right] \nabla_\phi \log p^\phi(z_2) \right\}$$

$$= \frac{1}{2} \left\{ \left[ f^\phi(z_1) - f^\phi(z_2) \right] \nabla_\phi \log p^\phi(z_1) - \left[ f^\phi(z_1) - f^\phi(z_2) \right] \nabla_\phi \log p^\phi(z_2) \right\}$$

$$= \frac{1}{2} \left[ f^\phi(z_1) - f^\phi(z_2) \right] \left[ \nabla_\phi \log p^\phi(z_1) - \nabla_\phi \log p^\phi(z_2) \right]$$

Plugging this back into Eq. (83) we get the following estimator for the gradient:

$$\nabla_\phi \mathbb{E}_{z \sim p^\phi} f^\phi(z) \approx \frac{1}{2} \left[ f^\phi(z_1) - f^\phi(z_2) \right] \left[ \nabla_\phi \log p^\phi(z_1) - \nabla_\phi \log p^\phi(z_2) \right] + \frac{1}{2} \left[ \nabla_\phi f^\phi(z_1) + \nabla_\phi f^\phi(z_2) \right]$$

#### E.5.1. FULL GRADIENT ESTIMATOR FOR TWO PARAMETER SETS

If we have $f^{\phi,\theta}(z)$ that depends on two sets of parameters $(\phi, \theta)$ where $z \sim p^\phi$, we need to compute:

$$\nabla_{(\phi,\theta)} \mathbb{E}_{z \sim p^\phi} f^{\phi,\theta}(z).$$

The key observation is that the distribution $p^\phi$ only depends on $\phi$ and not on $\theta$. Therefore:

**Gradient w.r.t. $\theta$:** Since $p^\phi$ doesn't depend on $\theta$, we can move the gradient inside the expectation:

$$\nabla_\theta \mathop{\mathbb{E}}_{z \sim p^\phi} f^{\phi,\theta}(z) = \mathop{\mathbb{E}}_{z \sim p^\phi} \nabla_\theta f^{\phi,\theta}(z) \approx \frac{1}{2} \left[ \nabla_\theta f^{\phi,\theta}(z_1) + \nabla_\theta f^{\phi,\theta}(z_2) \right].$$

This is a simple Monte Carlo average of the direct gradients.

**Gradient w.r.t. $\phi$:** Here we still need REINFORCE since $p^\phi$ depends on $\phi$:

$$\nabla_\phi \mathop{\mathbb{E}}_{z \sim p^\phi} f^{\phi,\theta}(z) = \mathop{\mathbb{E}}_{z \sim p^\phi} \left[ \nabla_\phi \log p^\phi(z) f^{\phi,\theta}(z) + \nabla_\phi f^{\phi,\theta}(z) \right]$$

$$\approx \frac{1}{2} \left[ f^{\phi,\theta}(z_1) - f^{\phi,\theta}(z_2) \right] \left[ \nabla_\phi \log p^\phi(z_1) - \nabla_\phi \log p^\phi(z_2) \right]$$

$$+ \frac{1}{2} \left[ \nabla_\phi f^{\phi,\theta}(z_1) + \nabla_\phi f^{\phi,\theta}(z_2) \right].$$

The full gradient estimator is then:

$$\nabla_{(\phi,\theta)} \mathop{\mathbb{E}}_{z \sim p^\phi} f^{\phi,\theta}(z) \approx \begin{pmatrix} \frac{1}{2} \left[ f^{\phi,\theta}(z_1) - f^{\phi,\theta}(z_2) \right] \left[ \nabla_\phi \log p^\phi(z_1) - \nabla_\phi \log p^\phi(z_2) \right] \\ + \frac{1}{2} \left[ \nabla_\phi f^{\phi,\theta}(z_1) + \nabla_\phi f^{\phi,\theta}(z_2) \right] \\ \frac{1}{2} \left[ \nabla_\theta f^{\phi,\theta}(z_1) + \nabla_\theta f^{\phi,\theta}(z_2) \right] \end{pmatrix}. \tag{84}$$

### E.5.2. SCHEDULE REGULARIZATION

If $\phi$ is optimized jointly with $\theta$, Eq. (5) admits degenerate solutions where the insertion/unmasking-time distributions place excessive probability mass near the endpoints $t = 0$ or $t = 1$ (e.g., delaying most events until $t \approx 1$). This suppresses the (rate-weighted) prediction term in Eq. (5) and allows the model hazards to match near-zero target hazards with negligible cost. To discourage such degenerate schedules, we add a sequence level schedule prior, which acts as a regularizer on the target schedule.

We add a loss term to regularize the *mixture CDF across positions* to be close to a straight line $F(t) = t$. Intuitively, this encourages the marginal event time of a uniformly sampled position to be approximately $\mathcal{U}[0,1]$, discouraging excessive mass near $t = 0$ or $t = 1$ without prescribing a particular per-position shape. The mixture CDFs over positions is defined as:

$$\bar{F}_{\text{in}}^\phi(t) := \frac{1}{L} \sum_{i=1}^L F_{\text{in}}^{\phi,i}(t), \qquad \bar{F}_{\text{um}}^\phi(t) := \frac{1}{L} \sum_{i=1}^L F_{\text{um}}^{\phi,i}(t), \tag{85}$$

where $L$ is the sequence length. We then add a penalty encouraging $\bar{F}_{\text{in}}^\phi(t) \approx t$ and $\bar{F}_{\text{um}}^\phi(t) \approx t$, on a grid $\{t_k\}_{k=1}^K \subset (0,1)$:

$$\mathcal{L}_{\text{reg}}(\phi) = \sum_{k=1}^K w_k \left( \bar{F}_{\text{in}}^\phi(t_k) - t_k \right)^2 + \sum_{k=1}^K w_k \left( \bar{F}_{\text{um}}^\phi(t_k) - t_k \right)^2, \tag{86}$$

which approximates the continuous-time objective $\int_0^1 \left( \bar{F}_{\text{in}}^\phi(t) - t \right)^2 + \left( \bar{F}_{\text{um}}^\phi(t) - t \right)^2 \, \mathrm{d}t$.

Additionally, we penalize each position separately to stay away from concentrating mass near the endpoints. For this, we fix a small time cutoff $t_\epsilon \in (0, 1/2)$ and tolerance $\delta \in (0,1)$, and enforce:

$$F_{\text{in}}^{\phi,i}(t_\epsilon) \leq \delta, \qquad 1 - F_{\text{in}}^{\phi,i}(1 - t_\epsilon) \leq \delta, \tag{87}$$

$$G_{\text{um}}^{\phi,i}(t_\epsilon) \leq \delta, \qquad 1 - G_{\text{um}}^{\phi,i}(1 - t_\epsilon) \leq \delta, \tag{88}$$

implemented via soft hinge penalties. In the implementation, we use $t_\epsilon = 0.01$ and $\delta = 0.01$.

### E.5.3. KUMARASWAMY SCHEDULES

**Choice of distribution family for the target schedule** We initially considered a Beta distribution for the event-time CDFs because it admits an interpretable location-scale parameterization. That form makes it convenient to regularize the

schedule parameters directly with a prior. Once we impose the precedence constraint $T_{\text{in}} < T_{\text{um}}$, however, we can no longer evaluate the log-likelihood of an intermediate state $z_t$ under $\phi$ in closed form for Beta. The truncated construction with Kumaraswamy distributions does yield closed-form expressions for $p^{\phi}_{t|z_1}(z_t)$, as we derive below. Kumaraswamy parameters do not separate cleanly into location and scale, so we instead regularize the schedule in time space (appendix E.5.2).

We use Kumaraswamy schedules for the insertion and unmasking times: $T_{\text{in}}{}^i \sim \text{Kumaraswamy}(a_{\text{in}}, b_{\text{in}})$ and $U^i \sim \text{Kumaraswamy}(a_{\text{um}}, b_{\text{um}})$ on $[0, 1]$. Then

$$F_{\text{in}}^i(t) = 1 - (1 - t^{a_{\text{in}}})^{b_{\text{in}}}, \qquad f_{T_{\text{in}}{}^i}(t) = a_{\text{in}} b_{\text{in}} t^{a_{\text{in}}-1}(1 - t^{a_{\text{in}}})^{b_{\text{in}}-1}, \qquad \mathbb{P}(T_{\text{in}}{}^i > t) = (1 - t^{a_{\text{in}}})^{b_{\text{in}}}, \tag{89}$$

$$F_{\text{um}}^i(t) = 1 - (1 - t^{a_{\text{um}}})^{b_{\text{um}}}, \qquad \dot{F}_{\text{um}}^i(t) = a_{\text{um}} b_{\text{um}} t^{a_{\text{um}}-1}(1 - t^{a_{\text{um}}})^{b_{\text{um}}-1}, \qquad 1 - F_{\text{um}}^i(t) = \mathbb{P}(U^i > t) = (1 - t^{a_{\text{um}}})^{b_{\text{um}}}. \tag{90}$$

By Eq. (29), the induced conditional distribution of $T_{\text{um}}{}^i$ on $[s, 1]$ is

$$\mathbb{P}(T_{\text{um}}{}^i \le t \mid T_{\text{in}}{}^i = s) = \delta(t \ge s)\left[1 - \frac{1 - F_{\text{um}}^i(t)}{1 - F_{\text{um}}^i(s)}\right] = \delta(t \ge s)\left[1 - \frac{(1 - t^{a_{\text{um}}})^{b_{\text{um}}}}{(1 - s^{a_{\text{um}}})^{b_{\text{um}}}}\right], \tag{91}$$

$$f_{T_{\text{um}}{}^i|T_{\text{in}}{}^i=s}(t) = \delta(t \ge s)\frac{\dot{F}_{\text{um}}^i(t)}{1 - F_{\text{um}}^i(s)} = \delta(t \ge s)\frac{a_{\text{um}} b_{\text{um}} t^{a_{\text{um}}-1}(1 - t^{a_{\text{um}}})^{b_{\text{um}}-1}}{(1 - s^{a_{\text{um}}})^{b_{\text{um}}}}. \tag{92}$$

Plugging into Eqs. (54) and (60) yields

$$\lambda_{\text{in}}^i(t) = \frac{a_{\text{in}} b_{\text{in}} t^{a_{\text{in}}-1}}{1 - t^{a_{\text{in}}}} \qquad \text{and} \qquad \lambda_{\text{um}}^i(t) = \frac{a_{\text{um}} b_{\text{um}} t^{a_{\text{um}}-1}}{1 - t^{a_{\text{um}}}}. \tag{93}$$

For $t \in [0, 1]$,

$$\mathbb{P}(T_{\text{in}}{}^i > t) = (1 - t^{a_{\text{in}}})^{b_{\text{in}}}, \tag{94}$$

$$\mathbb{P}(T_{\text{in}}{}^i \le t < T_{\text{um}}{}^i) = (1 - t^{a_{\text{um}}})^{b_{\text{um}}} \int_0^t \frac{a_{\text{in}} b_{\text{in}} s^{a_{\text{in}}-1}(1 - s^{a_{\text{in}}})^{b_{\text{in}}-1}}{(1 - s^{a_{\text{um}}})^{b_{\text{um}}}} \, ds, \tag{95}$$

$$\mathbb{P}(T_{\text{um}}{}^i \le t) = 1 - (1 - t^{a_{\text{in}}})^{b_{\text{in}}} - \mathbb{P}(T_{\text{in}}{}^i \le t < T_{\text{um}}{}^i). \tag{96}$$

To see that $\mathbb{P}(T_{\text{in}}{}^i \le t < T_{\text{um}}{}^i)$ is a valid probability, rewrite it using the law of total probability:

$$\mathbb{P}(T_{\text{in}}{}^i \le t < T_{\text{um}}{}^i) = \int_0^t f_{T_{\text{in}}{}^i}(s)\, \mathbb{P}(T_{\text{um}}{}^i > t \mid T_{\text{in}}{}^i = s)\, ds = \int_0^t f_{T_{\text{in}}{}^i}(s)\frac{1 - F_{\text{um}}^i(t)}{1 - F_{\text{um}}^i(s)}\, ds.$$

For $s \le t$, the survival function is monotone so $1 - F_{\text{um}}^i(t) \le 1 - F_{\text{um}}^i(s)$, hence the ratio is in $[0, 1]$ and the integral is in $[0, F_{\text{in}}^i(t)] \subseteq [0, 1]$. In general, the integral does not simplify to elementary functions, but it admits closed forms in certain special cases.

**Special case $a_{\text{in}} = a_{\text{um}} = a$**   If $a_{\text{in}} = a_{\text{um}} = a$, the integral $I(t) := \int_0^t \frac{a_{\text{in}} b_{\text{in}} s^{a_{\text{in}}-1}(1-s^{a_{\text{in}}})^{b_{\text{in}}-1}}{(1-s^{a_{\text{um}}})^{b_{\text{um}}}} \, ds$ simplifies to

$$
\begin{aligned}
I(t) &= \int_0^t a b_{\text{in}}\, s^{a-1}(1 - s^a)^{b_{\text{in}}-b_{\text{um}}-1}\, ds \\
&\overset{u=s^a}{=} b_{\text{in}} \int_0^{t^a} (1 - u)^{b_{\text{in}}-b_{\text{um}}-1}\, du \\
&= \begin{cases} \dfrac{b_{\text{in}}}{b_{\text{in}} - b_{\text{um}}}\left[1 - (1 - t^a)^{b_{\text{in}}-b_{\text{um}}}\right], & b_{\text{in}} \ne b_{\text{um}}, \\[2mm] -b_{\text{in}} \ln(1 - t^a), & b_{\text{in}} = b_{\text{um}}. \end{cases}
\end{aligned}
\tag{97}
$$

### E.5.4. PROBABILITY OF AN ORDER UNDER KUMARASWAMY SCHEDULE

For a sequence of length $L$, let $\{T^i\}_{i=1}^L$ denote the event times, one for each position, such that $T^i \le 1$ almost surely. Let $\pi : [L] \to [L]$ be a bijection such that $T^{\pi(1)} < T^{\pi(2)} < \cdots < T^{\pi(L)}$, then $\pi$ is an order over the positions induced by the event times.

Interestingly, we get a nice closed form expression for orders under the Kumaraswamy schedule if we fix $a$ for all positions (special case 2, mentioned in appendix E.5.3).

**Proposition 6** (Probability of an order under Kumaraswamy Schedule). *For $t \in [0, 1]$, let the CDF of the event time for the $i$-th position be given by the Kumaraswamy distribution with parameters $a$ and $b^i$, i.e., $F_{T^i}(t) = 1 - (1 - t^a)^{b^i}$, then the probability of an order $\pi$ is given by:*

$$\mathbb{P}(\pi) = \prod_{i=1}^{L} \frac{b^{\pi(i)}}{\sum_{j=i}^{L} b^{\pi(j)}} \tag{98}$$

*Proof.* The proof is straightforward and follows directly from the following two results.

1. If the event times $T^i$ are independent and have exponential distribution with parameter $b^i$, then the probability of an order $\pi$ is given by:

$$\mathbb{P}(\pi) = \prod_{i=1}^{L} \frac{b^{\pi(i)}}{\sum_{j=i}^{L} b^{\pi(j)}}$$

2. Let $N_t$ be a nonhomogeneous Poisson process with hazard rate of the form $\lambda(t) = b\beta(t)$. Then $\bar{N}_u$ is a homogeneous Poisson process with warped time $u = \int_0^t \beta(s)\mathrm{d}s$. Let $U_1 = \inf\{u : \bar{N}_u = 1\}$, be the first arrival time of the homogeneous Poisson process.

$\square$

---

**Algorithm 3** Training for **LoFlexMDM** (Detailed)

---

**Require:** Dataset $\mathcal{D}$, generator parameters $\theta$, target rate parameters $\phi$, learning rate $\eta$

1: **function** TRUNCSAMPLE$(s, F)$    // Sample from $F$ truncated to $[s, 1]$
2:      $u \sim \mathcal{U}(0, 1)$
3:      **return** $F^{-1}\left(u \cdot (1 - F(s)) + F(s)\right)$
4:
5: **while** not converged **do**
6:      Sample $z_1 \sim p_{\text{data}}$    // $z_1 \in \bar{\mathbb{Z}}$: padded with $[\text{D}]$ tokens
7:      Sample $t \sim \mathcal{U}(0, 1)$    // Diffusion time
8:      $(a_{\text{in}}^{\phi,i}, b_{\text{in}}^{\phi,i}, a_{\text{um}}^{\phi,i}, b_{\text{um}}^{\phi,i}) \leftarrow \phi(z_1)$    // Kumaraswamy params per position
9:      **for** $k = 1$ to $2$ **do**
10:        **for** $i = 1$ to $L$ **do**
11:          $T_{\text{in}}{}^{i,(k)} \sim F_{\text{in}}^{\phi,i}$    // Insertion time
12:          $T_{\text{um}}{}^{i,(k)} \leftarrow$ TRUNCSAMPLE$(T_{\text{in}}{}^{i,(k)}, F_{\text{um}}^{\phi,i})$    // Unmasking after insertion
13:          $z_t^{i,(k)} \leftarrow \begin{cases} [\text{D}] & T_{\text{in}}{}^{i,(k)} > t \\ [\text{M}] & T_{\text{in}}{}^{i,(k)} \le t < T_{\text{um}}{}^{i,(k)} \\ z_1^i & T_{\text{um}}{}^{i,(k)} \le t \end{cases}$
14:        **end for**
15:        $x_t^{(k)}, s_t^{(k)} \leftarrow f_{\text{contract}}(z_t^{(k)})$    // Use the bijection to get positions
16:        $\mathcal{L}_k \leftarrow \mathcal{L}_{\text{um}}^{\phi,\theta} + \mathcal{L}_{\text{in}}^{\phi,\theta}$    // Compute loss (Eq. (5))
17:      **end for**
18:      // Combine REINFORCE (with stop-grad) and pathwise gradients:
19:      $\mathcal{L} \leftarrow \frac{1}{2}\left[(\text{sg}(\mathcal{L}_1) - \text{sg}(\mathcal{L}_2)) \log \frac{p_{t|z_1}^{\phi}(z_t^{(1)})}{p_{t|z_1}^{\phi}(z_t^{(2)})} + \mathcal{L}_1 + \mathcal{L}_2\right]$
20:      $(\theta, \phi) \leftarrow (\theta, \phi) - \eta \nabla_{(\theta,\phi)} \mathcal{L}$    // Single backward pass
21: **end while**

---

### E.6. Sampling

The generator network predicts learnable hazard rates $\lambda_{\text{in},t}^{\theta,i}(x_t)$ for insertion at gap position $i$ and $\lambda_{\text{um},t}^{\theta,i}(x_t)$ for unmasking at masked position $i$, along with the token distribution $K^{\theta,i}(\cdot|x_t)$.

We use a simple $\tau$-leaping sampler that allows for larger time steps by sampling the number of events (insertions/unmaskings) in each interval $[t_n, t_{n+1})$ from a Poisson distribution. For small $\tau = t_{n+1} - t_n$, the first-order approximation gives Poisson intensities $\int_{t_n}^{t_{n+1}} \lambda_s ds \approx \lambda_{t_n} \cdot \tau$.

---

**Algorithm 4** Tau-Leaping Sampling for **LoFlexMDM**

---

1: **Input:** Time steps $0 = t_0 < t_1 < \cdots < t_N = 1$, nucleus $p$, confidence function $c$
2: Initialize $x_0 = \emptyset$ (empty sequence)
3: **for** $n = 0$ to $N - 1$ **do**
4: $\quad \tau \leftarrow t_{n+1} - t_n$
5: $\quad$ Predict $\lambda_{\text{in},t_n}^{\theta,i}(x_n)$, $\lambda_{\text{um},t_n}^{\theta,i}(x_n)$, and $K^{\theta,i}(\cdot|x_n)$ from network $\theta$
6: $\quad x_{n+1} \leftarrow x_n$
7: $\quad$ **for** each gap position $i = 0$ to $\text{len}(x_n)$ **do**
8: $\quad\quad N_{\text{in}}^i \sim \text{Poisson}(\lambda_{\text{in},t_n}^{\theta,i}(x_n) \cdot \tau)$
9: $\quad\quad$ **if** $N_{\text{in}}^i > 0$ **then**
10: $\quad\quad\quad$ Insert $N_{\text{in}}^i$ masks at gap position $i$ in $x_{n+1}$
11: $\quad\quad$ **end if**
12: $\quad$ **end for**
13: $\quad$ **for** each masked position $i$ in $x_{n+1}$ **do**
14: $\quad\quad$ **for** each token $y \in \mathbb{V}$ **do**
15: $\quad\quad\quad N_{\text{um}}^{i,y} \sim \text{Poisson}(\lambda_{\text{um},t_n}^{\theta,i}(x_n) \cdot K^{\theta,i}(y|x_n) \cdot \tau)$
16: $\quad\quad$ **end for**
17: $\quad\quad$ //candidate unmask locations
18: $\quad\quad u_i \leftarrow \delta(\sum_{y \in \mathbb{V}} N_{\text{um}}^{i,y} > 0)$
19: $\quad$ **end for**
20: $\quad$ **if** `confidence` enabled **then**
21: $\quad\quad$ // select $|\{i : u_i=1\}|$ locations with largest confidence $c_i$
22: $\quad\quad u \leftarrow \text{SELECT}(\sum_i u_i; c(K^\theta, \lambda^\theta))$
23: $\quad$ **end if**
24: $\quad$ **for** each masked position $i$ in $x_{n+1}$ with $u_i = 1$ **do**
25: $\quad\quad x_{n+1}^i \overset{\text{nucleus}(p)}{\sim} K^{\theta,i}(\cdot|x_n)$
26: $\quad$ **end for**
27: **end for**
28: **return** $x_N$

---

### E.7. Truncation *vs.* Rescaling and other choices for target schedule parameterization

We need to sample two event times $T_{\text{in}}$ and $T_{\text{um}}$ with a constraint $T_{\text{in}} < T_{\text{um}}$. For constructing a learnable schedule for this we need to parameterize

$$\mathbb{P}(T_{\text{in}} \leq t), \qquad \text{and} \qquad \mathbb{P}(T_{\text{um}} \leq t \mid T_{\text{in}} = s).$$

It is straightforward to parameterize $\mathbb{P}(T_{\text{in}} \leq t)$ using any monotonic function $F_{\text{in}}(t)$ that satisfies $F_{\text{in}}(0) = 0$ and $F_{\text{in}}(1) = 1$. However, for $\mathbb{P}(T_{\text{um}} \leq t \mid T_{\text{in}} = s)$, we have two easy choices. We can take a base monotonic function $F_{\text{um}}(t)$ and either **truncate** or **rescale** it to $[s, 1]$, where,

$$\text{truncate:} \qquad \mathbb{P}(T_{\text{um}} \leq t \mid T_{\text{in}} = s) = \frac{F_{\text{um}}(t) - F_{\text{um}}(s)}{1 - F_{\text{um}}(s)},$$

$$\text{rescale:} \qquad \mathbb{P}(T_{\text{um}} \leq t \mid T_{\text{in}} = s) = F_{\text{um}}\left(\frac{t - s}{1 - s}\right).$$

However, the unmasking hazard rate in Eq. (60) simplifies to a closed form expression only for the truncated case. Figure 9 shows a comparison of truncated and rescaled Kumaraswamy CDFs.

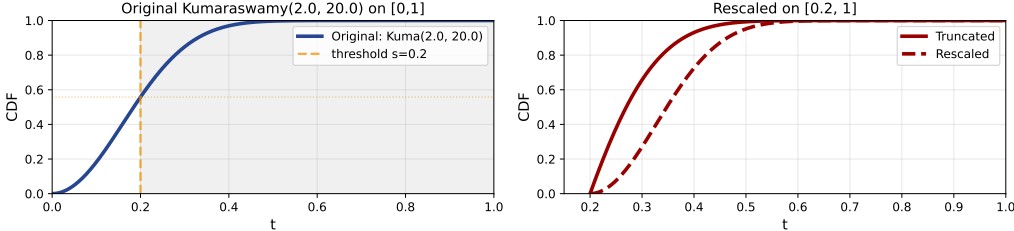

*Figure 9.* Comparison of truncated and rescaled Kumaraswamy CDFs for different parameter values.

# F. Details of the experiment setup

We use the xLM library (Patel et al., 2026a) to implement the training and evaluation harness for all our experiments. [5]

## F.1. Graph Traversal

**Dataset.** We use the star graphs dataset from Patel et al. (2025b) obtained from Hugging Face Hub.

**Model.** Following the setup in Kim et al. (2025a), we use a linear layer for modeling the insertion lengths in each gap for FlexMDM. For LMFlexMDM, we use two MLP layers for modeling the parameters of the learned target rates.

**Hyperparameters.** Table 6 shows the hyperparameters for the star graphs experiments.

| Model | lr | wt decay | train_steps | sampling_steps | confidence |
|---|---|---|---|---|---|
| FlexMDM | $10^{-4}$ | $10^{-2}$ | 80,000 | 500 | × |
| **Lo**FlexMDM | $10^{-4}$ | $10^{-2}$ | 80,000 | 500 | × |

*Table 5.* Hyperparameters for star graphs experiments.

## F.2. Molecule Generation

### F.2.1. DATASET

We use the bracketed version of the SAFE molecule strings provided by Noutahi et al. (2024).[6] The dataset contains around 1.17 billion unique molecules consisting of:

- ZINC20 (Irwin & Shoichet, 2005): $\sim 1$ billion commerically available compounds

- UniChem (Chambers et al., 2013b): $\sim 188$ million compounds from public databases

### F.2.2. TRAINING

**Hyperparameters.** Table 6 shows the hyperparameters for the molecule generation experiments.

---

[5]https://github.com/dhruvdcoder/xlm-core.
[6]https://huggingface.co/datasets/datamol-io/safe-gpt

| Model | lr | wt decay | train_steps | sampling_steps | confidence |
|---|---|---|---|---|---|
| FlexMDM | $10^{-4}$ | $10^{-2}$ | 50,000 | 1024 | ✗ |
| LoFlexMDM | $10^{-4}$ | $10^{-2}$ | 50,000 | 1024 | ✓ |

*Table 6.* Hyperparameters for star graphs experiments.

### F.2.3. EVALUATION METRICS

We use the same metrics as defined in Noutahi et al. (2024) and Lee et al. (2025). Each generated sample is a bracket SAFE string, which is then converted to standard SAFE string as done in `https://github.com/NVIDIA-Digital-Bio/genmol`. Next the standard SAFE string is decoded to canonical SMILES, optionally dropping fragments that fail to decode. Salts and counter-ions are also dropped. A sample counts as *invalid* if this pipeline yields no SMILES.

**Validity.**  Validity is the fraction of generated strings that decode to a valid molecule, i.e. the number of successful decodes divided by the total number of generations.

**Uniqueness.**  Among valid decodes, uniqueness is the fraction of distinct canonical SMILES: the number of unique SMILES divided by the number of valid molecules (with multiplicity).

**Diversity.**  Diversity is the average pairwise Tanimoto distance between Morgan fingerprints of the generated molecules, as in Noutahi et al. (2024); Lee et al. (2025). In code, we first deduplicate valid SMILES, require at least two unique structures, and then evaluate diversity with PyTDC's `Evaluator` for the `diversity` task, which implements this pairwise fingerprint-distance objective.

**Quality.**  Quality follows Lee et al. (2025): we count molecules that are drug-like and synthesizable under the thresholds from Jin et al. (2020)—quantitative estimate of drug-likeness (QED) (Bickerton et al., 2012) $\geq 0.6$ and synthetic accessibility (SA) (Ertl & Schuffenhauer, 2009) $\leq 4$. For each decoded SMILES, QED and SA are computed with PyTDC `Oracle` instances for the `qed` and `sa` tasks. Lee et al. (2025) define quality as the fraction of generated molecules that are simultaneously valid, unique, drug-like, and synthesizable. In our implementation, **quality** is the number of valid, unique generations satisfying both property thresholds, divided by the total number of generations.

**FCD.**  The Fréchet ChemNet Distance (FCD) (Preuer et al., 2018) measures distributional similarity between generated and reference molecules using penultimate-layer activations of ChemNet, which operates on canonical SMILES. However, the mapping from a molecule to its SAFE (or bracket SAFE) representation is a randomized one-to-many mapping: a single molecule is sliced into fragments via a stochastic fragmentation algorithm such as BRICS (Degen et al., 2008), and the resulting fragments can be ordered arbitrarily. The model therefore learns a distribution over the space of SAFE strings, not directly over canonical SMILES. Computing FCD requires mapping generated SAFE strings back to canonical SMILES, but this collapses the fragment-level structure that the model has learned to generate—the generalization patterns acquired in SAFE's fragment space need not align with proximity in ChemNet's SMILES-based feature space. As a result, naively converting generated SAFE strings to SMILES and computing FCD is not a faithful measure of distributional match for SAFE-based generative models. This is the main reason that previous works such as SAFE-GPT (Noutahi et al., 2024) and GenMol (Lee et al., 2025), which also operate on SAFE strings, do not compute FCD. We therefore omit FCD from our evaluation and focus on the metrics that are comparable across all methods that operate on SAFE strings.

### F.2.4. EVALUATION TASKS

***De novo* Generation**  For *de novo* generation, the model samples complete molecules unconditionally. FlexMDM and **Lo**FlexMDM start from an empty bracket SAFE sequence and grow it through alternating insertions and unmaskings, without conditioning on a fragment prefix. Following Lee et al. (2025), we generate 1000 molecules per run and report means and standard deviations over 5 independent eval runs (Table 2) with different random seeds. Metrics are computed as described in appendix F.2.3. We vary the sampling hyperparameters along two axes: budget and diversity. The budget is controlled by the maximum number of sampling steps, and the diversity is controlled by the nucleus sampling truncation threshold $p$, and confidence-based position selection (Holtzman et al., 2020). The Table 2 reports the results under maximum budget of 1024 steps and varying the diversity parameters. FlexMDM selects unmasking positions uniformly among eligible sites.

*Table 7.* **Fragment-constrained molecule generation results (full).** Means and standard deviations over 3 runs. The best results are highlighted in bold; the second-best mean in each column is underlined. († Noutahi et al. (2024) ‡; Lee et al. (2025))

| Model | Task | Validity (%) | Diversity | Uniqueness (%) | Quality (%) |
|---|---|---|---|---|---|
| ARM† | Linker design | 76.6 ±5.1 | 0.545 ±0.007 | 82.5 ±1.9 | 21.7 ±1.1 |
| | Motif extension | 96.1 ±1.9 | 0.562 ±0.003 | 66.8 ±1.2 | 18.6 ±2.1 |
| | Scaffold decoration | 97.7 ±0.3 | 0.575 ±0.008 | 74.7 ±2.5 | 10.0 ±1.4 |
| | Superstructure generation | 95.7 ±2.0 | 0.573 ±0.028 | 83.0 ±5.9 | 14.3 ±3.7 |
| MDM‡ | Linker design | **100.0** ±0.0 | 0.547 ±0.002 | **83.7** ±0.5 | 21.9 ±0.4 |
| | Motif extension | 82.9 ±0.1 | 0.617 ±0.002 | 77.5 ±0.1 | 30.1 ±0.4 |
| | Scaffold decoration | 96.6 ±0.8 | 0.591 ±0.001 | 82.7 ±1.8 | 31.8 ±0.5 |
| | Superstructure generation | 97.5 ±0.9 | 0.599 ±0.009 | **83.6** ±1.0 | 34.8 ±1.0 |
| FlexMDM | Linker design | 99.7 ±0.1 | **0.599** ±0.005 | 63.7 ±0.3 | 50.8 ±0.6 |
| | Motif extension | 99.7 ±0.1 | **0.623** ±0.001 | **79.9** ±0.4 | 46.9 ±0.9 |
| | Scaffold decoration | 99.6 ±0.1 | **0.615** ±0.002 | **84.3** ±0.5 | 39.0 ±0.7 |
| | Superstructure generation | 99.7 ±0.03 | **0.616** ±0.002 | 74.5 ±1.5 | 35.8 ±1.4 |
| **Lo**FlexMDM | Linker design | 99.6 ±0.1 | 0.576 ±0.003 | 64.4 ±0.6 | **51.7** ±0.9 |
| | Motif extension | **99.9** ±0.1 | 0.608 ±0.001 | 79.2 ±0.8 | **53.6** ±0.7 |
| | Scaffold decoration | **99.8** ±0.1 | 0.601 ±0.000 | 82.6 ±0.6 | **40.5** ±0.7 |
| | Superstructure generation | **100.0** ±0.03 | 0.593 ±0.002 | 72.6 ±0.4 | **37.0** ±0.6 |

**Lo**FlexMDM additionally supports confidence-based position selection during decoding and nucleus sampling (Holtzman et al., 2020) with truncation threshold $p$ on the token distribution. Table 2 reports both models under several decoding choices ($p \in \{0.5, 1.0\}$, with and without confidence-based position selection). The order-analysis in Section 4.2.1 uses **Lo**FlexMDM with trainable $b_{in}^{\phi}$, confidence-based position selection, and $p=0.5$.

SAFE-GPT (†) and MDM (‡) implemented following Noutahi et al. (2024) and Lee et al. (2025), respectively. For the fixed-length MDM baseline, Lee et al. (2025) sample mask-chunk lengths from the ZINC250k length distribution. Variable-length models do not use a fixed mask canvas and therefore do not need to sample masked chunk lengths from a predefined distribution.

**Constrained Generation** We use the fragment-constrained benchmark of Noutahi et al. (2024), with evaluation settings aligned to Lee et al. (2025). Conditional inputs come from ten reference drugs (Cyclothiazide, Maribavir, Spirapril, Baricitinib, Eliglustat, Erlotinib, Futibatinib, Lesinurad, Liothyronine, and Lovastatin) via the fragment decompositions in the SAFE-GPT data release. We report four completion tasks and omit scaffold morphing, which Lee et al. (2025) treat as equivalent to linker design.

*Linker design* asks the model to generate a fragment that connects two given side chains. *Motif extension* fixes a starting motif with attachment points and asks the model to generate the remaining fragment that completes the molecule. *Scaffold decoration* conditions on a scaffold with attachment sites and asks the model to fill in substituents. Following Noutahi et al. (2024), scaffolds are built by placing random attachment points on a core substructure before generation. *Superstructure generation* provides a substructure constraint and asks the model to sample a full molecule that contains it.

Each task is cast as prefix completion on bracket SAFE strings: the provided fragments are tokenized and held fixed, and the model generates the rest of the sequence. For both FlexMDM and **Lo**FlexMDM, we use the *de novo* checkpoints above without task-specific finetuning, with the same sampling budgets as in Section 4.2, i.e., 1024 steps and with respective best sampling parameters (confidence-based position selection for **Lo**FlexMDM, and uniform sampling for FlexMDM). Following Lee et al. (2025), we draw one completion per conditional input ($N=1$) and report means and standard deviations over three independent runs. ARM (†) and MDM (‡) rows in Table 3 are taken from Noutahi et al. (2024) and Lee et al. (2025). For the fixed-length MDM baseline, Lee et al. (2025) sample mask-chunk lengths from the ZINC250k distribution and tune task-specific decoding hyperparameters ($\tau=1.2$; $r=3$ for linker design, $r=1.2$ for motif extension, and $r=2$ for scaffold decoration and superstructure generation). Variable-length models instead grow the sequence from the fragment prefix rather than decoding within a fixed mask canvas.

# G. Additional Results and Analyses

## G.1. Star Graph Traversal

Table 8 shows the token accuracy along with exact match for the star graph traversal task.

### G.1.1. FLEXIBILITY *vs* STABILITY

When both insertion and unmasking target rates are trainable, optimization can become unstable. We do not attribute this failure mode to the precedence constraint $T_{in} < T_{um}$ itself, nor to the Kumaraswamy parameterization in particular. The main issue is the freedom to concentrate transition probability mass into very narrow time intervals. That concentration can sharpen the implied generation order, but it also makes training brittle. If the density becomes highly peaked around an incorrect time $t \in [0, 1]$, the resulting gradients can become extremely large.

We expect this behavior for any sufficiently expressive schedule family. Truncation together with fixing $b_{um} = 1$ provides enough regularization to prevent excessive concentration while still preserving a rich family of orderings. Figure 10 compares training loss on the hard star graph traversal task with trainable and frozen target unmasking rates. Fixing $b_{um} = 1$ stabilizes training significantly. As shown in Table 8, it also improves the overall performance of the final generator.

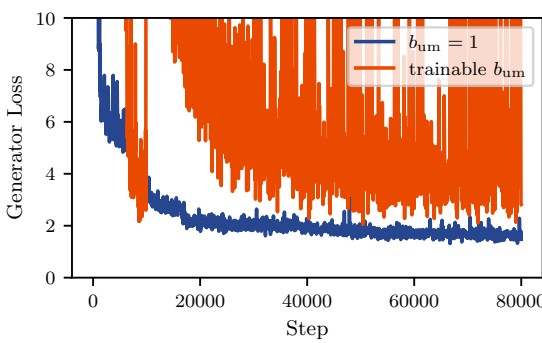

*Figure 10.* Loss comparison for the hard star graph traversal task with and without fixing target unmasking rate ($b_{um} = 1$). The erratic dynamics of a fully unconstrained set of target rates motivates freezing $b_{um}$.

### G.1.2. GENERATION ORDER CORRELATION

Figure 4b in the main text analyzes generation order using only trajectories that achieve 100% exact match. We reproduce that figure at two-column width in Fig. 11 for clarity and better visibility. Figure 12 repeats the analysis without the exact-match filter. The points spread more over generation order, but the pattern in correlation strength is unchanged: FlexMDM $(0.06, -0.21)$ is much weaker than **Lo**FlexMDM $(-0.07, -0.42)$. To compare the exact-match filter and the unfiltered analysis directly, Figure 13 plots filtered and unfiltered results in separate panels for each sampling setting. The facet layout presents the same comparison setting by setting, which makes it easier to see how much the unfiltered clouds broaden and shift right relative to the filtered case, without changing the relative ordering between models. This analysis establishes strong correlation rather than causation, but the rightward shift for incorrect predictions is consistent with less favorable order accompanying degraded prediction quality.

### G.1.3. EXAMPLES: GENERATION ORDER

Figures 14a and 14b show representative generation trajectories for **Lo**FlexMDM and FlexMDM on the hard star graph task. Each figure pairs the traversal on the query graph with the step-by-step unmasking sequence.

## G.2. Small Molecule Generation

Table 10 shows the results for *de novo* small molecule generation with additional nucleus-sampling $p$ values, as well as a shared-backbone implementation.

### G.2.1. GENERATION ORDER EXAMPLES

Here we continue the discussion of generation order from Section 4.2.1. In addition to the Fig. 6, we provide the complete table of per-category mean generation times in Table 9. All category-level differences are significant ($p < 10^{-13}$). **Lo**FlexMDM also separates the existence of an attachment point ($\mu \approx 0.61$) from its identity ($\mu \approx 0.80$),

*Table 9.* Summary of mean normalised generation time per token category ($\pm$ std). $\Delta\mu$ is the difference (FlexMDM − **Lo**FlexMDM), where positive values indicate FlexMDM generates that category later.

| Category | FlexMDM | **Lo**FlexMDM | $\Delta\mu$ |
|---|---|---|---|
| Ring closure | $.71 \pm .26$ | $.48 \pm .28$ | $+.23$ |
| Frag sep (.) | $.67 \pm .27$ | $.55 \pm .18$ | $+.12$ |
| Aromatic atom | $.65 \pm .26$ | $.55 \pm .24$ | $+.10$ |
| Attach bracket | $.63 \pm .27$ | $.61 \pm .19$ | $+.02$ |
| Attach label | $.68 \pm .27$ | $.80 \pm .24$ | $-.12$ |
| Aliphatic atom | $.68 \pm .27$ | $.80 \pm .16$ | $-.13$ |
| Bond / branch | $.70 \pm .26$ | $.84 \pm .15$ | $-.14$ |

| Task | Model | Exact Match (%) | Token Accuracy (%) |
|---|---|---|---|
| **star_medium** | ARM | 75.0 | 81.4 |
| | MDM | 36.5 | 90.6 |
| | FlexMDM | | |
| | (conf=null) | 89.6 | 95.7 |
| | (conf=top_prob) | 91.3 | 97.0 |
| | **Lo**FlexMDM (Aux. Size=small) | | |
| | (b_um=1, conf=null) | 93.2 | 97.8 |
| | (b_um=1, conf=top_prob) | 93.0 | 97.7 |
| | **Lo**FlexMDM (Shared=True) | | |
| | (b_um=1, conf=null) | 93.6 | 97.7 |
| | (b_um=1, conf=top_prob) | 93.2 | 97.6 |
| **star_hard** | ARM | 23.0 | 43.2 |
| | MDM | 21.0 | 54.9 |
| | FlexMDM | | |
| | (conf=null) | 6.0 | 48.5 |
| | (conf=top_prob) | 7.4 | 49.8 |
| | **Lo**FlexMDM (Aux. Size=small) | | |
| | (conf=null) | 38.0 | 64.5 |
| | (b_um=1, conf=null) | 87.9 | 96.3 |
| | (b_um=1, conf=top_prob) | 88.1 | 96.7 |
| | **Lo**FlexMDM (Shared=True) | | |
| | (b_um=1, conf=null) | 34.2 | 66.5 |
| | (b_um=1, conf=top_prob) | 69.7 | 86.9 |

*Table 8.* Raw results for star graph traversal tasks.

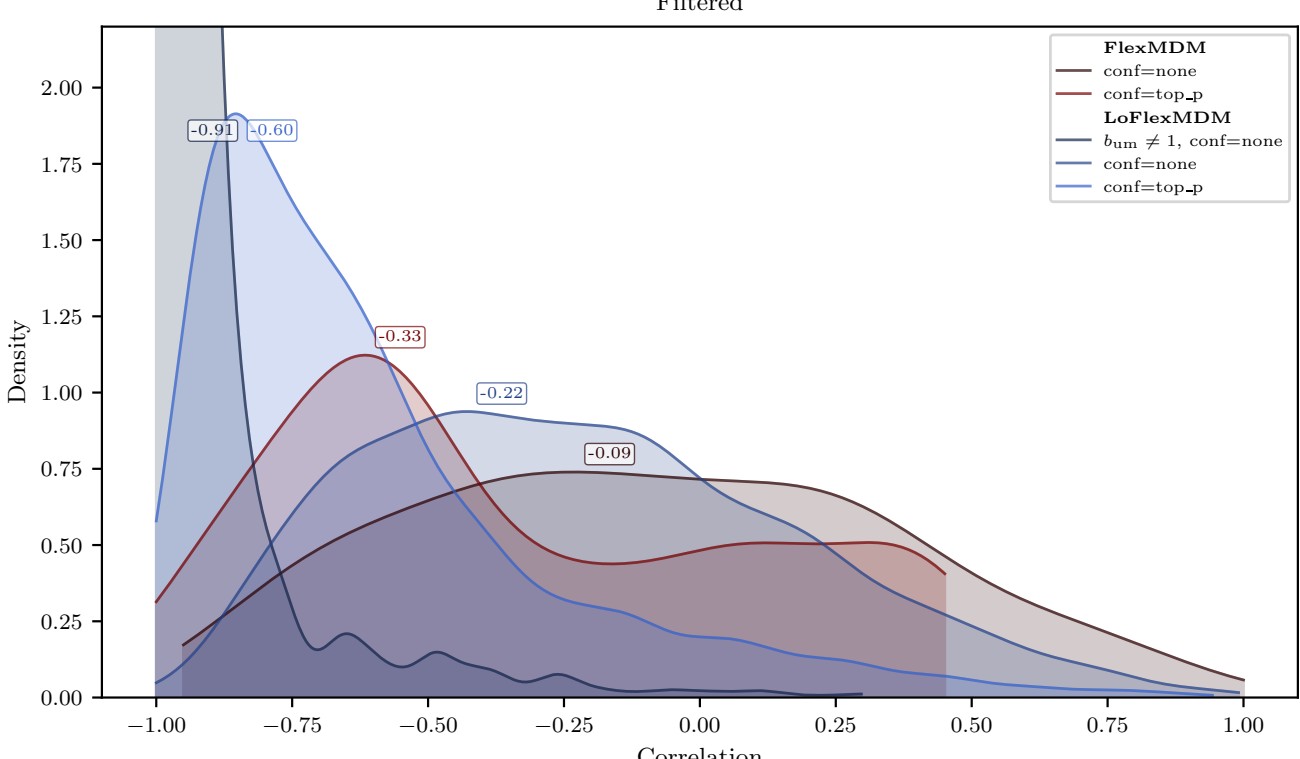

*Figure 11.* Correlation between normalized generation order and distance from the junction node (Fig. 4b), shown at two-column width for clarity. Only trajectories with 100% exact match are included.

committing to where fragments connect before deciding which fragments connect. Figure 15 shows another *de novo* generation trajectory for FlexMDM and **Lo**FlexMDM.

## H. Extended Related Work

**Discrete Flow Matching.** Flow matching (Lipman et al., 2022) and equivalently stochastic interpolants (Albergo et al., 2023) learn a probability path $p_t$ to transport samples from a source distribution $p_0$ to a target distribution $p_1$ in bounded continuous time. The probability path is represented by a time-dependent velocity vector field which together with the initial condition and the continuity equation uniquely determines the probability path. Discrete flow matching (Campbell et al., 2024; Gat et al., 2024) extends the generative framework of continuous flow matching to discrete probability spaces where the velocity vector field is replaced by a time-dependent rate matrix of a Continuous Time Markov Chain (CTMC), and the continuity equation is replaced by the Kolmogorov Forward Equation.

**Variable Length Discrete Flow Matching.** In domains like text, using fixed length model requires predicting padding tokens to generate variable length sequences. Since the pad tokens become extremely frequent in the dataset, the model typically predicts excessive pad tokens (Kim et al., 2026; Patel et al., 2025a). This has motivated several works to move away from fixed length models towards insertion-based models. The desiderata of *i)* modeling distributions over variable-length sequences and *ii)* the ability to insert tokens have inspired a number of recent works to move away from fixed-window text diffusion towards insertion- and substitution- based discrete flow models in which positions during the generation process acquire a relative rather than absolute interpretation. In particular, FlexMDM (Kim et al., 2025a) grows a sequence by parameterizing an action space which allows the insertion of a mask token and the unmasking of already-inserted masks. Patel et al. (2026b) formulate the generative process as a CTMC that inserts tokens at any available gaps between existing tokens, which allows the direct insertions of unmasked tokens, as opposed to the two-step insertion/unmasking approach of FlexMDM. EditFlows (Havasi et al., 2025) defines a discrete flow by means of insertion, deletion and substitution operations to support variable-length sequence generation. Our approach is general and can be applied to any of these models.

*Table 10.* Full set of results for *de novo* small molecule generation. For both FlexMDM and **Lo**FlexMDM, we use 1024 sampling steps. ❄ indicates frozen, 🔥 indicates trainable; ✓/✗ indicates whether confidence-based position selection for unmasking is enabled, and $p$ indicates the truncation threshold for nucleus sampling (Holtzman et al., 2020). † Noutahi et al. (2024) ‡ Lee et al. (2025)

| $b_{um}$ | $b_{in}$ | conf. | $p$ | Validity (%) | Diversity | Uniqueness (%) | Quality (%) |
|---|---|---|---|---|---|---|---|
| SAFE-GPT† | | | | 94.0 ±0.4 | 0.879 ±0.001 | **100.0** ±0.0 | 54.7 ±0.3 |
| MDM‡ | | | | 96.7 ±0.3 | 0.896 ±0.001 | 99.3 ±0.2 | 53.8 ±1.7 |
| **FlexMDM** | | | | | | | |
| ❄ | ❄ | ✓ | 1.0 | 67.8 ±0.3 | **0.940** ±0.000 | 61.7 ±0.7 | 5.5 ±0.4 |
| ❄ | ❄ | ✓ | 0.9 | 67.3 ±0.9 | 0.930 ±0.000 | 56.9 ±0.8 | 5.3 ±0.2 |
| ❄ | ❄ | ✓ | 0.5 | 71.4 ±0.9 | 0.920 ±0.000 | 41.4 ±0.7 | 3.1 ±0.2 |
| ❄ | ❄ | ✗ | 1.0 | 98.4 ±0.1 | 0.900 ±0.000 | 99.6 ±0.1 | 47.0 ±0.7 |
| ❄ | ❄ | ✗ | 0.9 | 98.9 ±0.1 | 0.890 ±0.000 | 99.6 ±0.1 | 62.0 ±0.7 |
| ❄ | ❄ | ✗ | 0.5 | 99.5 ±0.2 | 0.870 ±0.000 | 99.7 ±0.1 | 63.2 ±0.5 |
| **Lo**FlexMDM | | | (Aux. Size=medium) | | | | |
| ❄ | 🔥 | ✓ | 1.0 | 99.0 ±0.0 | 0.900 ±0.000 | 99.3 ±0.2 | 55.4 ±0.3 |
| ❄ | 🔥 | ✓ | 0.9 | 99.5 ±0.1 | 0.880 ±0.000 | 99.5 ±0.0 | 71.4 ±0.8 |
| ❄ | 🔥 | ✓ | 0.5 | **99.9** ±0.0 | 0.830 ±0.000 | 93.2 ±0.6 | 69.3 ±0.7 |
| ❄ | 🔥 | ✗ | 1.0 | 99.1 ±0.1 | 0.900 ±0.000 | 99.6 ±0.1 | 55.1 ±0.6 |
| ❄ | 🔥 | ✗ | 0.9 | 99.5 ±0.1 | 0.880 ±0.000 | 99.6 ±0.1 | 72.1 ±0.5 |
| ❄ | 🔥 | ✗ | 0.5 | 99.7 ±0.1 | 0.850 ±0.000 | 99.5 ±0.1 | **79.5** ±0.7 |
| **Lo**FlexMDM | | | (Aux. Size=small) | | | | |
| ❄ | 🔥 | ✓ | 1.0 | 99.1 ±0.1 | 0.900 ±0.000 | 99.4 ±0.1 | 55.5 ±0.7 |
| ❄ | 🔥 | ✓ | 0.9 | 99.7 ±0.1 | 0.880 ±0.000 | 99.3 ±0.1 | 72.1 ±1.1 |
| ❄ | 🔥 | ✓ | 0.5 | 99.7 ±0.1 | 0.820 ±0.000 | 93.6 ±0.5 | 73.3 ±0.6 |
| ❄ | 🔥 | ✗ | 1.0 | 99.2 ±0.1 | 0.900 ±0.000 | 99.6 ±0.2 | 55.4 ±0.7 |
| ❄ | 🔥 | ✗ | 0.9 | 99.4 ±0.1 | 0.880 ±0.000 | 99.3 ±0.1 | 71.0 ±0.5 |
| ❄ | 🔥 | ✗ | 0.5 | 99.7 ±0.1 | 0.850 ±0.000 | 99.4 ±0.1 | 79.3 ±0.5 |
| **Lo**FlexMDM | | | (Aux. Size=xtiny) | | | | |
| ❄ | 🔥 | ✓ | 1.0 | 99.0 ±0.1 | 0.900 ±0.000 | 99.7 ±0.0 | 55.8 ±0.7 |
| ❄ | 🔥 | ✓ | 0.9 | 99.6 ±0.1 | 0.880 ±0.000 | 99.6 ±0.1 | 71.1 ±0.5 |
| ❄ | 🔥 | ✓ | 0.5 | 99.7 ±0.1 | 0.830 ±0.000 | 93.4 ±0.3 | 72.2 ±0.6 |
| ❄ | 🔥 | ✗ | 1.0 | 99.0 ±0.1 | 0.900 ±0.000 | 99.8 ±0.1 | 55.1 ±0.7 |
| ❄ | 🔥 | ✗ | 0.9 | 99.4 ±0.1 | 0.880 ±0.000 | 99.6 ±0.1 | 70.7 ±0.6 |
| ❄ | 🔥 | ✗ | 0.5 | 99.7 ±0.1 | 0.850 ±0.000 | 99.6 ±0.2 | 79.4 ±0.6 |
| **Lo**FlexMDM | | | (Shared Backbone) | | | | |
| 🔥 | 🔥 | ✓ | 1.0 | 98.3 ±0.1 | 0.900 ±0.000 | 99.4 ±0.2 | 51.5 ±0.6 |
| 🔥 | 🔥 | ✓ | 0.9 | 99.2 ±0.1 | 0.890 ±0.000 | 99.6 ±0.1 | 69.2 ±1.1 |
| 🔥 | 🔥 | ✓ | 0.5 | 99.8 ±0.0 | 0.850 ±0.000 | 96.9 ±0.2 | 70.8 ±1.0 |
| 🔥 | 🔥 | ✗ | 1.0 | 98.4 ±0.1 | 0.900 ±0.000 | 99.2 ±0.1 | 52.8 ±1.1 |
| 🔥 | 🔥 | ✗ | 0.9 | 99.2 ±0.1 | 0.890 ±0.000 | 99.3 ±0.1 | 66.8 ±0.9 |
| 🔥 | 🔥 | ✗ | 0.5 | 99.5 ±0.0 | 0.860 ±0.000 | 99.5 ±0.1 | 71.2 ±0.6 |

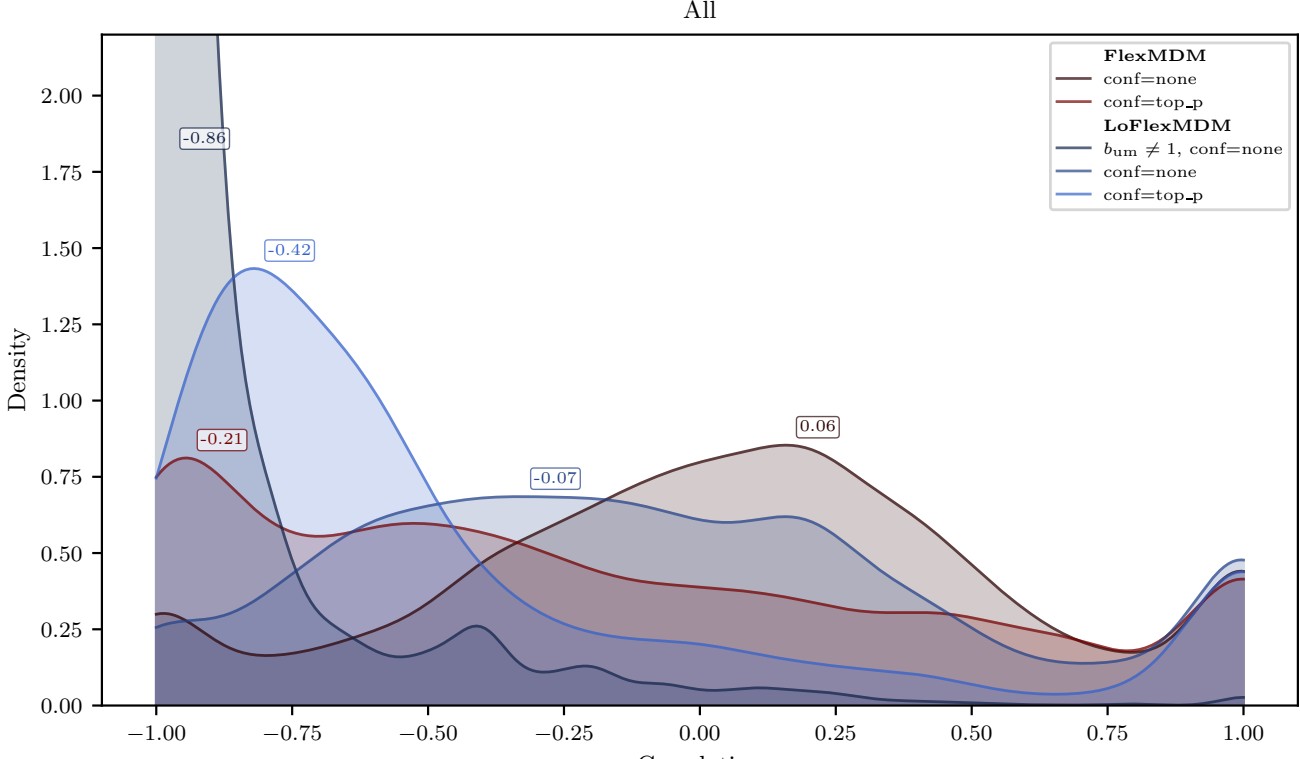

*Figure 12.* Correlation between normalized generation order and distance from the junction node, without filtering to 100% exact-match trajectories.

**Learnable generation order.** There has been some work highlighting the importance of generation order for fixed-length unmasking models. Kim et al. (2025b) show that even inference time heuristics can improve the generation quality of masked diffusion models significantly. However, the work on learning the generation order along with the generator parameters at training time has been limited. Wang et al. (2025) propose a method for learning the generation order for autoregressive models that work with absolute position information. For insertion models, most of the exiting works focus on models that insert one token at a time and require simulating the generation trajectory for learning the generation order (Gu et al., 2019; Brantley et al., 2019). Our approach is general and can be applied to all three settings: fixed-length unmasking, variable-length unmasking, and insertion only models, and combines well with multi-token prediction.

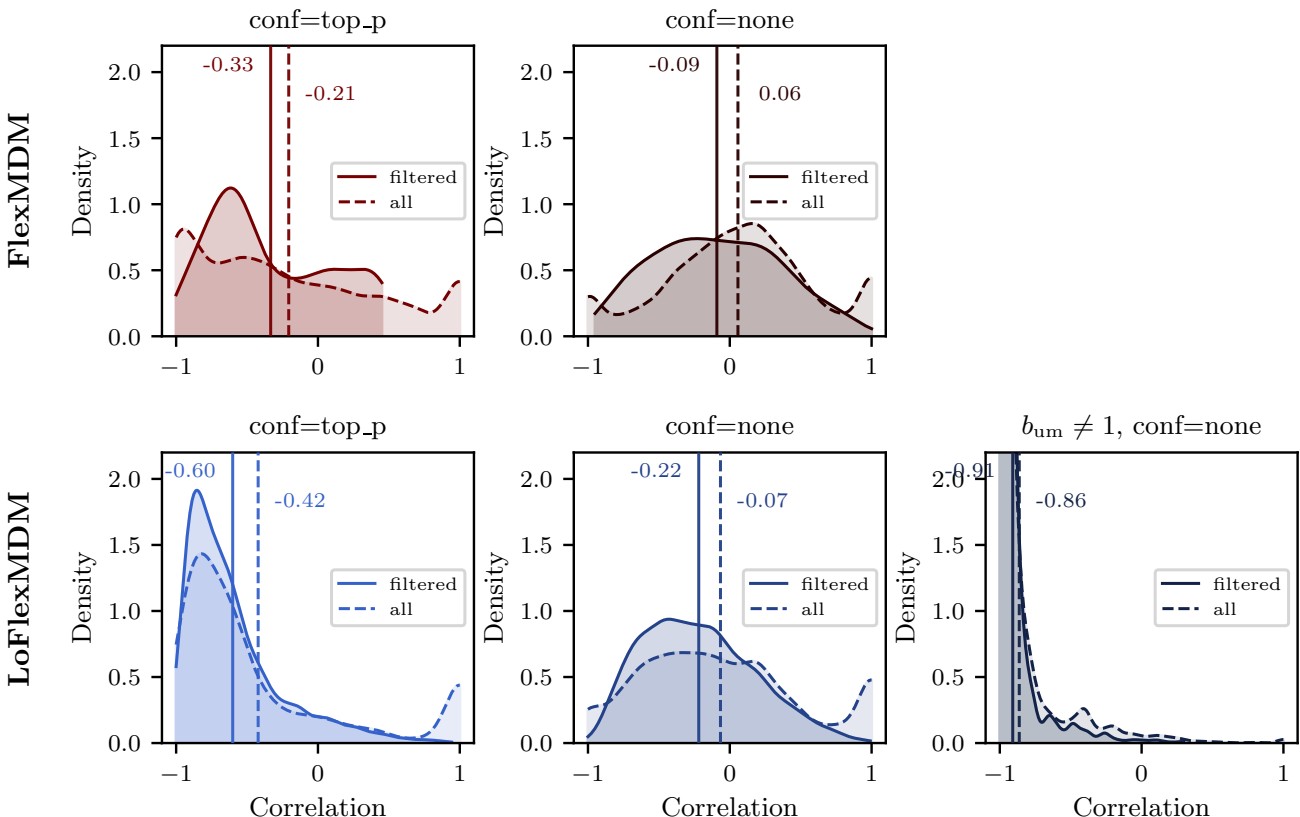

*Figure 13.* Filtered (100% exact match) vs. unfiltered generation-order correlation for each sampling setting.

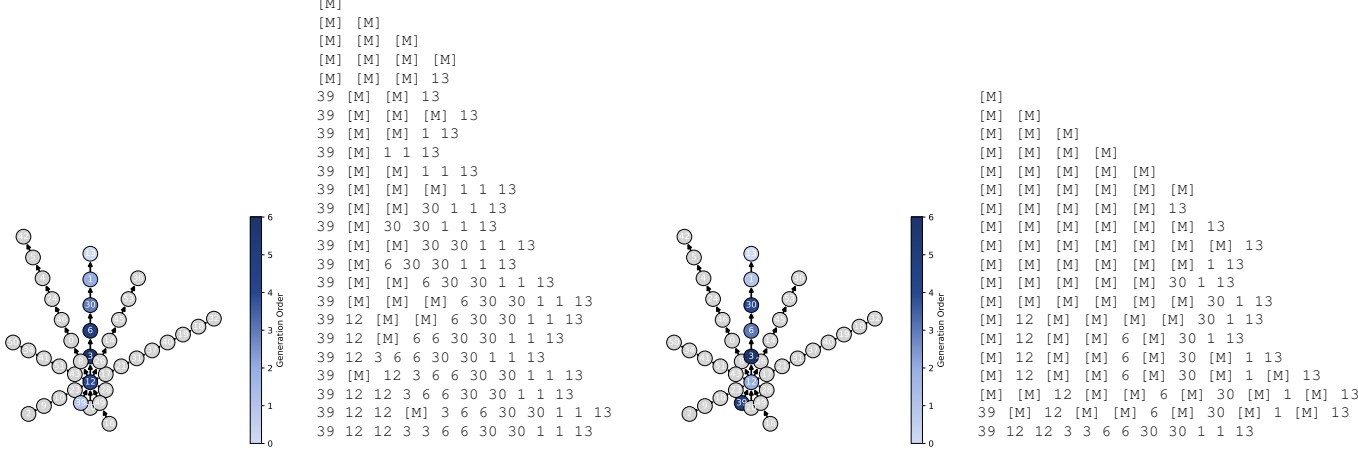

*(a)* Example of generation order for **Lo**FlexMDM.

*(b)* Example of generation order for FlexMDM.

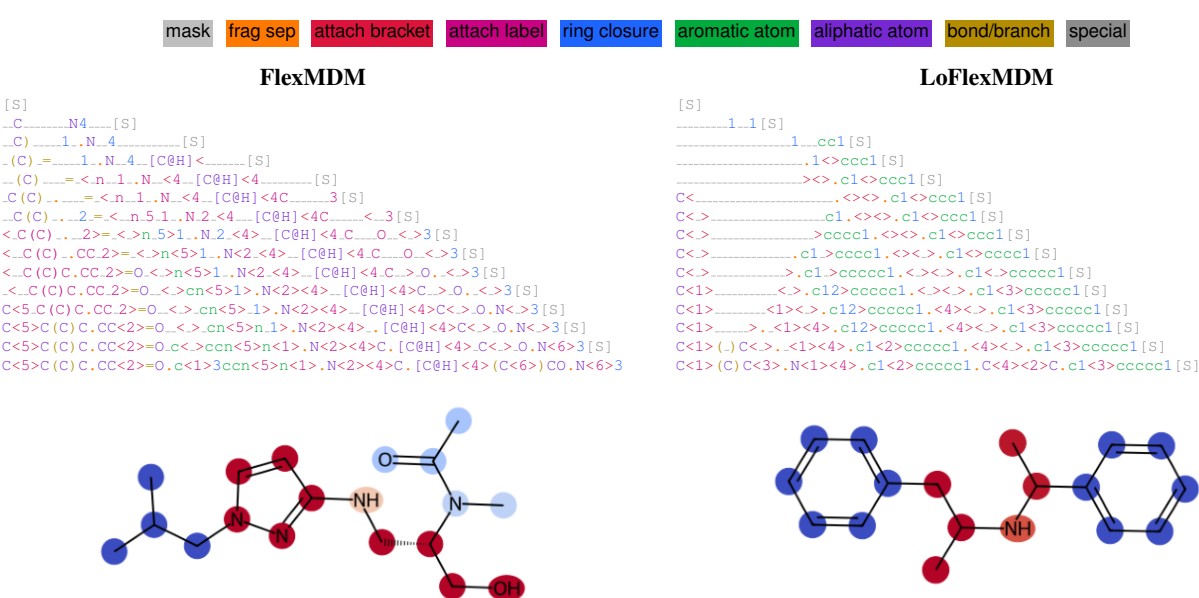

Figure 15. **Additional *de novo* generation trajectory.** FlexMDM (left) and **Lo**FlexMDM (right) on the same molecule. Every line is a subsampled generation step; grey _ marks a still-masked position. Tokens are coloured by semantic category (legend above). The molecule graph below each trajectory colours atoms by normalised generation time (blue = early, red = late). **Lo**FlexMDM first resolves ring closures and fragment separators, then places attachment brackets and aromatic atoms, and fills in aliphatic atoms, attachment labels, and bond/branch tokens last. FlexMDM shows no such systematic pattern.

