# OpenReview forum: "Insertion Based Sequence Generation with Learnable Order Dynamics"
_ICML.cc/2026/Conference — ICML 2026 regular_

### Official Review · Reviewer_xYTm · 2026-03-08

**Soundness:** 2
**Presentation:** 2
**Significance:** 3
**Originality:** 3
**Overall Recommendation:** 4
**Confidence:** 1

**Summary:**

This paper proposes LFlexMDM, a method for learning generation order in insertion-based sequence generation. Building on FlexMDM, the paper introduces learnable order dynamics into the insertion and unmasking target rates, so that the model learns not only what to generate, but also in what order to generate it. Experiments on graph traversal and molecule generation show that the learned order dynamics can improve generation quality over the FlexMDM baseline.

**Compliance With Llm Reviewing Policy:**

Affirmed.

**Final Justification:**

The added conditional generation experiments appear more meaningful, so I maintain my positive score.

**Key Questions For Authors:**

1. What is the model size? Could the model scale up like recent MDM work such as LLaDA [1]?
2. Is this order dynamics design tied specifically to insertion-based MDMs, or could it also improve standard MDMs?
3. Although the method is natural under the discrete flow matching framework, is learned order strongly tied to this framework? Could similar order learning also be introduced into a more general MDM framework? Are there any related works along this direction?

[1] Nie, Shen, et al. "Large language diffusion models." arXiv preprint arXiv:2502.09992 (2025).

**Limitations:**

This paper does not discuss limitations.

**Strengths And Weaknesses:**

*Disclaimer: I did not carefully check the theorems in this paper.*

Pros:
1. The motivation is interesting and reasonable. Addressing the generation order issue in MDM-style models is important and meaningful.
2. Incorporating the learning objective into the discrete flow matching framework is natural and well-motivated.

Cons:
1. The experiments are too toy-like. Graph Traversal does not look like a mainstream task, while unconditional generation on ZINC is also somewhat boring. Metrics that are already close to 100% do not seem very informative or discriminative. Could the authors also evaluate on more mainstream and meaningful text tasks, as well as conditional molecule generation tasks? Similarly, since Sudoku [1] is a common task used to highlight the importance of generation order, it would also be helpful to see results on such a benchmark.

[1] Kim, Jaeyeon, et al. "Train for the worst, plan for the best: Understanding token ordering in masked diffusions." arXiv preprint arXiv:2502.06768 (2025).

---

> ### Author Rebuttal · Authors · 2026-03-31
>
> Thank you appreciating the motivation and novelty of our work. Below we address each of your questions and suggestions.
>
> > Metrics that are already close to 100% do not seem very informative or discriminative.
>
> We agree that metrics near saturation can be less informative. In our molecule experiments, however, the most important metric is Quality, which measures the fraction of valid, unique, drug-like, and synthesizable molecules, and this metric is far from saturated. On this metric, LFlexMDM shows a substantial improvement in Table 2 (e.g. 39 vs 62).
> Similarly, the metrics are also not saturated for the hard variant of the star task, where there is high variability in sequence lengths and branching at the junction node. Here again, LFlexMDM shows a large improvement over FlexMDM (7% vs 88%).
>
> > evaluation on conditional molecule generation tasks
>
> We evaluated the models on constrained molecule generation tasks, which directly address your suggestion about conditional molecule generation (results at: https://anonymous.4open.science/api/repo/InsertionWithLearnabelOrderDynamics/file/constrained_gen_table.pdf).
> We observe strong improvement in Quality over FlexMDM for conditional generation tasks as well.
> We also perform order analysis for the molecular strings. As shown in the figure in the anonymous link: https://anonymous.4open.science/api/repo/InsertionWithLearnabelOrderDynamics/file/fig_molecule_order_analysis.pdf, LFlexMDM learns a clear structure-first, chemistry-later order, when compared to a more uniform order exhibited by FlexMDM.
> These results broaden the empirical evaluation beyond unconditional generation and provide additional evidence that the proposed order-learning mechanism is useful on constrained molecule-generation tasks as well.
>
> > What is the model size? Could the model scale up like recent MDM work such as LLaDA [1]?
>
> Our current model is a 12-layer transformer with around 96 million parameters. The shared backbone model can be scaled up to larger sizes like LLaDA but due to compute constraints, we focuse on molecule generation with smaller model.
>
> > Is this order dynamics design tied specifically to insertion-based MDMs, or could it also improve standard MDMs?
>
> The proposed framework is general and can also be applied to fixed-length MDMs; in that case, there is no insertion-rate component, and only the unmasking order needs to be learned. However, as discussed in the introduction, fixed-length MDMs are limited by their fixed-length assumption and use of absolute position information, which makes them less suited for dependencies that are better expressed through relative positions.
> For this reason, learning a good generation order has more limited benefits in the fixed-length setting, whereas it is especially valuable in the more expressive variable-length setting considered in our work. This is why we focus on variable-length MDMs in this paper.
>
> > evaluation on sudoku
>
> We agree that Sudoku is a relevant benchmark for studying order learning in fixed-length settings, and we did consider it. However, our goal in this work is to study learned order dynamics in the variable-length setting, where the benefits are more central to the modeling problem. In contrast, Sudoku is a fixed-length task, and strong inference-time heuristics are already known to perform well there, making it less diagnostic of the specific advantages of our framework.
> For this reason, we chose to focus our empirical budget on variable-length tasks that are more directly aligned with the main contribution of the paper.
>
> > Could similar order learning also be introduced into a more general MDM framework? Are there any related works along this direction?
>
> Yes, we believe the core idea of learning order dynamics is not tied specifically to insertion-based MDMs and could also be introduced into more general fixed-length MDM frameworks. In that case, the insertion-rate component would be absent, and the main object to learn would be the unmasking order. However, this would be strictly less general.
> Our focus here is on the variable-length setting, where the problem is more expressive and, in our view, more compelling.
>
> As for related work, there has been prior work on learning generation order in fixed-length settings like Wang et al. (2025). However, to the best of our knowledge, there are no prior works that learn order dynamics for variable-length unmasking models.
>
> **Thank you for reading our rebuttal. We are happy to answer any follow-up questions.**
>
> **References:**
> - Wang et. al. (2025). Learning-Order Autoregressive Models with Application to Molecular Graph Generation.

---

> > ### Author Rebuttal · Reviewer_xYTm · 2026-04-01
> >
> > The authors have addressed my concerns.

---

### Official Review · Reviewer_6NmF · 2026-03-12

**Soundness:** 3
**Presentation:** 3
**Significance:** 2
**Originality:** 3
**Overall Recommendation:** 4
**Confidence:** 4

**Summary:**

The paper proposes LFlexMDM, a framework that equips FlexMDM, a variable-length generative discrete diffusion model that inserts masks dynamically and then unmasks them, with learnable order dynamics. The original FlexMDM uses a fixed uniform schedule for insertion and unmasking; LFlexMDM reformulates this through a projected DFM framework and explicitly models the per-position transition rates via Kumaraswamy CDFs parameterized by an auxiliary network, thereby learning a data-dependent order of insertion and unmasking. FlexMDM becomes a special case where these rates are set to constants. The authors demonstrate LFlexMDM on graph traversal tasks and de novo molecular design.

**Compliance With Llm Reviewing Policy:**

Affirmed.

**Final Justification:**

All concerns addressed and I think the paper has the merit to be accepted. Despite the authors' positioning of this paper against FlexMDM, I still believe it inherits too much from the original work which to some extent shadows its own novelty. Therefore, my recommendation is 4: weak accept.

**Key Questions For Authors:**

Some questions related to "learnable order dynamics":
1. The correlation analysis in Figure 5 only includes samples that achieve 100% exact match. Does this introduce a selection bias because generating the exact correct traversal plausibly requires a reasonable ordering in the first place? How does the model learn the order in general?
2. Also, Figure 5 establishes correlation between generation order and distance from the junction node, but not causation. Is the model producing correct results because it learned the right order, or does it simply happen that successful generations (selected post-hoc) exhibit structured ordering?
3. In Sec 3.2, it seems the $\phi$ network is optimized such that $\theta$ network produces smaller losses (please correct me if I'm wrong). How does this provide a signal for discovering an optimal order?

**Limitations:**

Yes

**Strengths And Weaknesses:**

## Strengths:

1. The paper is clearly motivated and well presented. Recasting FlexMDM as a projected DFM special case is insightful, and the Kumaraswamy parameterization naturally connects learnable rates to generation order while keeping FlexMDM as a recoverable special case.
2. Extending discrete diffusion models to variable-length generation is an active research topic and yet learning the generation order is underdeveloped and hence this work provides meaningful insights to this direction.

## Weaknesses
1. It seems most part of the framework is inherited from FlexMDM and only the $\theta$ and $\phi$ networks to learn the order dynamics are novel. Maybe more clarification on the positioning of this framework from FlexMDM is needed.
2. Empirical experiments are quite limited. Only a synthetic graph traversal experiment and a molecular design task are included, both relatively small-scale. More standard benchmark with MDMs and FlexMDMs like natural language is missing.
3. The $\theta$ and $\phi$ networks design are only discussed in the caption of Figure 1 but not in the main text and such design choice and training strategy is not clearly motivated.
4. What actual "order dynamics" is learned seems a bit unclear. (I will elaborate in the Questions section so that the authors can reply directly).

---

> ### Author Rebuttal · Authors · 2026-03-31
>
> Thank you for the thoughtful review.
>
> > correlation analysis in Fig 5 only includes samples that achieve 100% exact match.
>
> This is a great point. We agree that filtering to exact-match samples can weaken the evidence. We initially applied this filter because generation order is less informative when the generation trajectory has already deviated from the correct solution. However, to address your concern, we repeated the analysis without the exact-match filter; the resulting plot is available at this anonymous link: https://anonymous.4open.science/api/repo/InsertionWithLearnabelOrderDynamics/file/fig5.pdf
> As noted in the captions for the middle and last figures in the link, the correlations become slightly weaker for all (model, confidence) combinations, but their relative ordering remains unchanged relative to the original Fig. 5. This unfiltered analysis provides stronger evidence that our conclusions about learned order are not an artifact of the exact-match filter, and we will incorporate it in the paper.
>
> > Fig 5 establishes correlation between generation order and distance from the junction node, but not causation
>
> You are correct that Fig. 5 alone does not establish causality quantitatively. However, it is consistent with the causal interpretation supported by our broader empirical evidence. In particular, when incorrectly predicted sequences are included (cf. Figs. 1 and 2 in the anonymous link), the histogram shifts slightly to the right, suggesting that degraded prediction quality is accompanied by less favorable order behavior.
>
> > regarding experiments on natural language tasks
>
> We agree that evaluation on natural language tasks would further strengthen the paper. In this work, we focused on settings where we can isolate the effect of learning the generation order under our compute constraints, which is why we chose tasks where that factor can be studied cleanly.
> To broaden the empirical picture, we also provide order analysis for molecular strings in our response to Reviewer GaC5 (anonymous link: https://anonymous.4open.science/api/repo/InsertionWithLearnabelOrderDynamics/file/fig_molecule_order_analysis.pdf). Additionally, we ran inference on constrained molecule generation tasks (results at: https://anonymous.4open.science/api/repo/InsertionWithLearnabelOrderDynamics/file/constrained_gen_table.pdf).
> While these experiments do not fully replace large-scale natural language evaluation, they provide additional evidence that the proposed order-learning mechanism is useful beyond the graph setting.
>
> > ... networks design are only discussed in the caption of Figure 1
>
> We use a standard DDiT Transformer backbone for both networks, as discussed in Section 5. The token-selection head for the generator $K^\theta$ is the standard softmax head, while the rate heads $\lambda^{\theta}$ are 2-layer MLPs that produce a single non-negative scalar using $\exp()$; $\lambda^\phi$ uses the same architecture.
> We will state this more explicitly in the main text, and accompany it with a link to the open-source implementation of the model.
>
>
> > ... more clarification on the positioning of this framework from FlexMDM is needed.
>
> We agree that the positioning relative to FlexMDM should be as clear as possible. We currently discuss this relation in the following places:
> 1. Introduction paragraphs 2 and 3.
> 2. Line 135 in Section 3.
> 3. Remark 1, section 3.2.
> 4. Last paragraph of the related work Section 4.
>
> In the revision, we will make this contrast more even more explicit, and are happy to incorporate detailed suggestions.
>
> > $\phi$ network is optimized to minimize the loss of the $\theta$ network. How does this provide a signal for discovering an optimal order?
>
> This is a great question.
> As shown in Fig. 1, the target rates $\phi$ only control the timing of events, while the token identities are fixed by the ground truth. In this way, $\phi$ determines the generation order implicitly through event timing. Therefore, for two intermediate states $z_t^{(1)}$ and $z_t^{(2)}$ corresponding to the same ground-truth sequence $z_1$, increasing $p^\phi(z_t^{(1)})$ relative to $p^\phi(z_t^{(2)})$ encourages the model to prefer order in which $z_t^{(1)}$ appears before $z_t^{(2)}$. As shown in Algo. 1, lines 8-10, the first term in the RLOO objective does exactly this:
> $$
> (sg(L_1) - sg(L_2)) \log \frac{p^\phi(z_t^{(1)})}{p^\phi(z_t^{(2)})},
> $$
>
> The difference $L_1 - L_2$ acts like a reward or advantage signal indicating which partial state is better according to the generator. The objective then increases the probability of the better state under $\phi$, thereby encouraging the corresponding generation order.
>
>
> **We hope these clarifications address your concerns, and we are happy to answer any follow-up questions.**

---

> > ### Author Rebuttal · Reviewer_6NmF · 2026-04-03
> >
> > Thanks for the rebuttal. All concerns addressed and I will raise my score.

---

### Official Review · Reviewer_Tekf · 2026-03-12

**Soundness:** 3
**Presentation:** 3
**Significance:** 2
**Originality:** 2
**Overall Recommendation:** 4
**Confidence:** 4

**Summary:**

This paper presents LFlexMDM, an extension of the FlexMDM framework designed to support variable-length sequence generation with learnable ordering dynamics, Instead of relying on fixed or uniform generation schedules, the approach models insertion and unmasking rates using Kumaraswamy distributions, enabling the generation order to adapt to the data. The framework is trained using Discrete Flow Matching (DFM) together with a score-based gradient estimator based on REINFORCE, allowing the generator and target rate networks to be optimized jointly without simulating complete trajectories. Experiments demonstrate the method on tasks including graph traversal and de novo small-molecule generation.

**Compliance With Llm Reviewing Policy:**

Affirmed.

**Final Justification:**

I have increased my recommendation from Weak Reject to Weak Accept. The authors provided a rebuttal that clarifies the technical trade-offs inherent in the LFlexMDM framework. Specifically, the justifications for the choice of the Kumaraswamy distribution over the Beta distribution for closed-form tractability, and the explanation of the "stochastic fragmentation" in the SAFE representation, have resolved most of my concerns regarding the soundness of the methodology and evaluation.

**Key Questions For Authors:**

- Why does the unmasking rate ($b_{um}$) specifically destabilize the model? Is this a byproduct of the precedence constraint ($T_{in} < T_{um}$), or an issue with the Kumaraswamy parameterization?
- The authors added schedule regularization to prevent mass concentration at endpoints. Could a more robust prior or a different distribution family (e.g., Beta) mitigate the need to freeze $b_{um}$?
- How does LFlexMDM compare to more recent non-insertion discrete flow models that use simple heuristics for importance sampling of tokens?

**Strengths And Weaknesses:**

**Strengths**

- The integration of learnable hazard rates into the DFM framework for variable-length sequences is a principled approach to overcoming the limitations of fixed-order autoregressive models.
- The use of Kumaraswamy schedules provides a neat analytical pathway to closed-form hazard rates, which is crucial for efficient training.
- The qualitative analysis on star graphs demonstrates that the model successfully learns the "optimal" order (endpoints to junction) for specific structural tasks.

**Weaknesses**

- A significant concern is the "numerical degeneracy" and instability reported when both insertion and unmasking rates are trainable. The paper found that a fully unconstrained set of target rates led to erratic dynamics, forcing them to freeze the unmasking rate ($b_{um}=1$) to achieve stable results. This significantly limits the claimed flexibility of the "learnable dynamics."
- The "shared" backbone setting—which is more computationally efficient—results in a "considerable drop in performance" on harder tasks like the star graph traversal. This suggests the model's complexity might outpace its practical efficiency.
- While LFlexMDM shows gains in validity and quality, it suffers a decrease in diversity compared to FlexMDM.
- The experimental results for small molecule generation are limited by the absence of FCD scores or similar metrics that quantify how well the generated samples fit the underlying data distribution.

---

> ### Author Rebuttal · Authors · 2026-03-30
>
> Thank you for the thoughtful review and for appreciating the novelty and the neatness of our idea and analysis.
>
> Below we address each of your main concerns and questions.
>
> > LFlexMDM improves validity and quality, but diversity decreases compared to FlexMDM.
>
> A flat or slightly lower diversity is expected here because learning the order dynamics narrows the space of generation trajectories by discouraging less useful orders that can inflate the diversity.
> Importantly, Fig. 7 shows that simply reducing entropy in FlexMDM does not further improve Quality, whereas LFlexMDM continues to improve on Quality consistently. This suggests the gain is not coming from being more conservative in a generic sense, but from learning a better generation order, which is the main contribution of our work.
>
> > The experimental results for molecule generation would be stronger with FCD or similar distribution-matching metrics.
>
> We agree that a distribution-matching metric would be informative.
> However, FCD operates on canonical SMILES.
> Mapping from a SMILES molecule to its SAFE representation is a randomized one-to-many mapping: a single molecule is sliced into fragments via a stochastic fragmentation algorithm (BRICS).
> The model learns a distribution over the space of SAFE strings, and mapping generated SAFE strings back to canonical SMILES collapses the fragment-level structure that the model has learned to generate---the generalization patterns acquired in SAFE's fragment space need not align with proximity in ChemNet's SMILES-based feature space.
> For this reason, prior SAFE-based works such as Noutahi et al. (2024) and Lee et al. (2025) primarily report validity, uniqueness, diversity, and quality rather than FCD, and we follow the same evaluation protocol.
>
> *If you have a suggestion and a reference for a faithful distribution matching metric for SAFE strings, we'd be happy to include it to improve the evaluation.*
>
> > Could a stronger prior or a different distribution family (e.g. Beta) mitigate the need to freeze $b_{um}$?
>
> This is a great point. We initially formulated the idea with a Beta distribution because it admits a location-scale style parameterization, which makes it convenient to regularize the parameters directly with a prior.
> However, once we impose the precedence constraint, we can no longer obtain the log-likelihood of an intermediate state $z_t$ under $\phi$ in closed form for Beta.
> In contrast, the truncated construction with the Kumaraswamy distribution yields a closed-form expression (Appendix D.2). Since Kumaraswamy parameters do not separate cleanly into location and scale, we instead regularize in time space.
>
> > ... freezing the unmasking rate to 1 ... limits the claimed flexibility of the learnable dynamics.
>
> > Why does the unmasking rate ($b_{um}$) specifically destabilize the model? Is this due to the precedence constraint ($T_{in} < T_{um}$), or to the Kumaraswamy parameterization?
>
> We believe the main issue is neither the precedence constraint itself nor the Kumaraswamy family specifically. The core source of instability is the freedom to concentrate transition probability mass into very narrow time intervals. While this can induce sharp orders, it also makes optimization brittle: if the density becomes highly peaked around an incorrect time $t \in [0,1]$, the resulting gradients can become extremely large. We expect this failure mode to arise with any sufficiently expressive family.
> Interestingly, truncation together with fixing $b_{um}$ appears to provide enough regularization to prevent excessive concentration, while still preserving a rich enough family of orderings. Empirically, this is already sufficient to learn substantially better orders than FlexMDM on star graphs (Section 5.1.1, Fig. 5), and the same trend carries over to molecule generation, as discussed in our response to Reviewer GaC5 (see the order analysis plot for molecules at this anonymous link: https://anonymous.4open.science/api/repo/InsertionWithLearnabelOrderDynamics/file/fig_molecule_order_analysis.pdf)
>
> *We will add a more detailed discussion of these two points in the appendix.*
>
> > How does LFlexMDM compare to more recent non-insertion discrete flow models that use simple heuristics for importance sampling of tokens?
>
> By this if you mean methods that use heuristics like entropy or top-probability for selecting a position to unmask, then we use this at inference time for both FlexMDM and LFlexMDM. Please correct us if we misunderstood your question.
>
> **We hope these clarifications address your concerns, and we are happy to answer any follow-up questions.**

---

> > ### Author Rebuttal · Reviewer_Tekf · 2026-04-03
> >
> > Thank you for your response. I will update my score accordingly.

---

### Official Review · Reviewer_GaC5 · 2026-03-13

**Soundness:** 4
**Presentation:** 3
**Significance:** 4
**Originality:** 3
**Overall Recommendation:** 5
**Confidence:** 3

**Summary:**

The paper proposes a way to learn the insertion order in an Unmasking Discrete Flow Matching (UDFM) model. The theory developed allows the parameterization of the arrival rate of changes over tokens, which controls when (in time) a token should be inserted and/or unmasked. This enables to learn the ordering, which in turn is approximated by the actual UDFM model. The model is evaluated on a graph traversal dataset, and molecular strings dataset.

**Compliance With Llm Reviewing Policy:**

Affirmed.

**Final Justification:**

The authors have answered all my questions. I found the paper particularly interesting and useful for developing future generative models that handle discrete data and arbitrary sizes. For these reasons, I have raised my score to accept.

**Key Questions For Authors:**

1. Why not merge the insertion and unmasking events? Shouldn't it be possible to choose a token class as soon as it is inserted? Or do you wish to create dependencies among tokens which are unmasked together?
2. Because you set $z_1$ in line 158 as $x_1$ padded with [D] tokens, do you expect generated samples to behave similarly, meaning that the model generates sequences of tokens, followed by sequences of [D]? Or do you also allow sequences with [D] in-between the tokens? Do you handle it with $f_\text{rm-drop}$?
3. How are metrics computed in the molecular dataset? Could you include the computation in the paper?

**Limitations:**

Yes

**Strengths And Weaknesses:**

### Strengths:
1. The findings of the paper are very interesting, and the paper is a step forward in filling the gap between UDFMs and autoregressive models, which allow the generation of arbitrary length sequences.
2. The paper is dense, but is not difficult to follow. The inclusion of the Algorithms in the main body of the paper is surely welcome.
3. Experimental results are interesting and high in performance. The analysis of the learned ordering in the graph traversal

### Weaknesses:
1. Figure 1 (right) is not very clear. The used visual elements might make it overly complex and difficult to understand. Please, either explain it more in the caption, or simplify the figure.
2. Although samples are variable length, the representation in memory during generation still uses the same footprint as the full sequence with length $L$, where $L$ is the maximum length.
3. In my opinion, an ordering analysis for molecular strings could strengthen the paper even further.

I'm willing to raise my score if points regarding presentation (e.g., figure, analyses) and questions are addressed.

### Typos:
- 041: is are -> are
- 154: is $(a,b,c)$ right? Shouldn't it be the sequence of mask tokens and $c,f$?

---

> ### Author Rebuttal · Authors · 2026-03-30
>
> Thank you for the constructive suggestions. Below we address each question and suggestion.
>
> > Figure 1 (right) is not very clear.
>
> We simplified the figure to only focus on how the loss is computed at a single input position.
> The updated figure and caption are at the anonymous link: https://anonymous.4open.science/api/repo/InsertionWithLearnabelOrderDynamics/file/fig1.pdf
>
> > Because you set $z_{1}$ in line 158 as $x_{1}$ padded with [D] tokens, do you expect generated samples to behave similarly...
>
>  The [D] tokens are only a theoretical device in $\bar{\mathbb Z}\coloneqq (\mathbb V\cup \{\text{[D]}\})^L$ (line 143), introduced for cleaner exposition and proofs. The generator operates in the variable-length space $\mathbb X \coloneqq \bigcup_{k=1}^L \mathbb V^k$, so it never receives or outputs [D] tokens; it always works on $f_{\text{rm-drop}}$-ed sequences.
>  We agree this is a subtle point and will clarify it earlier in Section 3.
>
> > .. samples are variable length, but ... memory during generation uses same footprint ...
>
> The memory footprint is not the same as in fixed-length models because the generator works directly with variable-length sequences in $\mathbb X$ (without [D] tokens), sequence length grows during generation as [M] tokens are inserted. For batched inference, we pad to the maximum current length in the batch and use an attention mask to ignore the padded positions.
>
> > metrics computation
>
> We follow the evaluation protocol of Noutahi et al. (2024). Generated bracket SAFE strings are decoded to canonical SMILES with the SAFE library. Decoding failures are consided invalid molecules.
>
> - *Validity* is the fraction of valid decodes;
> - *Uniqueness* is the fraction of distinct canonical SMILES among valid samples;
> - *Diversity* is the average pairwise Tanimoto distance between Morgan fingerprints on deduplicated valid SMILES, computed with PyTDC `Evaluator("diversity")`;
> - *Quality* (Lee et al. (2025)) is the fraction of molecules that are valid, unique, drug-like, and synthesizable, where drug-like means QED  $\ge 0.6$ (Bickerton et al (2012)) and synthesizable means SA  $\le 4$ (Ertl et. al. (2009), following Jin et. al. (2020), with QED and SA computed per molecule via PyTDC `Oracle`. We will add this to the main text, move lower-level details to the appendix, and include the code link.
>
> > Why not merge the insertion and unmasking events? Shouldn't it be possible to choose a token class as soon as it is inserted? Or do you wish to create dependencies among tokens which are unmasked together?
>
> Yes, our framework can be instantiated as an insertion-only model that chooses a token immediately upon insertion. However, such a model can place only one token in a gap at a time. This is theoretically simpler, but it ignores correlations among tokens within the same gap. By inserting [M] tokens first and unmasking later, we can explicitly model correlations over larger contiguous chunks, which has been a key advantage of discrete diffusion models. A main benefit of our framework is precisely that it supports this more expressive variable-length unmasking model.
>
> > an ordering analysis for molecular strings could strengthen the paper even further.
>
> Unlike star graph traversal, molecular strings do not have a single correct context-dependent generation order making a similar position-level analysis difficult.
> However, following your suggestion we performed a coarser global analysis: we assign each token in the final SAFE string to one of seven semantic categories and measure its normalized generation time. The anonymous link to the figure is here: https://anonymous.4open.science/api/repo/InsertionWithLearnabelOrderDynamics/file/fig_molecule_order_analysis.pdf
>
> As seen, FlexMDM is roughly uniform across categories (means $\approx 0.65-0.71$), whereas LFlexMDM learns a clear structure-first, chemistry-later order: ring closures ($\mu{=}0.48$) and fragment separators ($\mu{=}0.55$) are generated first, then attachment brackets ($\mu{=}0.61$), and finally aliphatic atoms ($\mu{=}0.80$), attachment labels ($\mu{=}0.80$), and bond/branch tokens ($\mu{=}0.84$).  Table 1 in that figure summarizes the violin plots (the differeces in the means are statistically significant with $p < 10^{-13}$ for Mann-Whitney $U$ test).
>  We also observe that LFlexMDM separates the existence of an attachment point ($\mu{=}0.61$) from its identity ($\mu{=}0.80$), deciding where fragments connect before deciding which fragments connect. Qualitative trajectories (Figure 2 at the same anonymous link) show the same pattern. Overall, this suggests that LFlexMDM discovers a chemically meaningful ordering by resolving topological constraints before chemical details.
>
>  **We believe we have addressed all your questions and concerns. We are happy to answer follow-up questions.**
>
> # References
>
> - Jin, W. et al. (2020). Multi-objective molecule generation using interpretable substructures. In *ICML*.

---

> > ### Author Rebuttal · Reviewer_GaC5 · 2026-04-03
> >
> > Dear authors,
> >
> > Thank you for your response, which has addressed all my concerns. I found the ordering analysis of molecules to be very interesting.

---

> > > ### Author Response · Authors · 2026-04-06
> > >
> > > Thank you for your thoughtful follow-up, for your encouraging comments on our analysis, and for confirming that our response addressed your concerns. We also appreciate your suggestions, which helped improve the paper. Since you had mentioned that you might reconsider your score if these points were addressed, we wanted to gently ask whether you might still be open to revisiting it, if you feel the paper now merits it.

---

### Decision · Program_Chairs · 2026-04-30

**Decision:**

Accept (regular)

**Comment:**

This paper presents a method to learn the insertion order in a discrete flow matching model. This extends the FlexMDM framework to Learnable FlexMDM.

The reviewers agree that the experiments are compelling and well presented. While the novelty over FlexMDM is quite marginal, reviewers leaned towards acceptance. I agree with reviewers on this point and recommend acceptance at this time.